



**Real-Time Snow Depth Estimation and Historical Data**
**Reconstruction Over China Based on a Random Forest**
**Machine Learning Approach**
Jianwei Yang[1], Lingmei Jiang[1], Kari Luojus[2], Jinmei Pan[3], Juha Lemmetyinen[2],
Matias Takala[2], Shengli Wu[4]
[1]State Key Laboratory of Remote Sensing Science, Jointly Sponsored by Beijing Normal University and
the Institute of Remote Sensing and Digital Earth of Chinese Academy of Sciences, Beijing Engineering
Research Center for Global Land Remote Sensing Products, Faculty of Geographical Science, Beijing
Normal University, Beijing 100875, China
[2]Finnish Meteorological Institute, Helsinki Fi00101, Finland
[3]State Key Laboratory of Remote Sensing Science, Institute of Remote Sensing and Digital Earth,
Chinese Academy of Sciences, Beijing 100101, China
[4]National Satellite Meteorological Center, China Meteorological Administration, Beijing 100081, China
*Corresponding Author:* Lingmei Jiang (jiang@bnu.edu.cn)
**Abstract.** Snow depth data time series are valuable for climatological and hydrological applications.
Passive microwave (PMW) sensors are advantageous for estimating spatially and temporally continuous
snow depth. However, PMW estimate accuracy has several problems, which results in poor performances
of traditional snow depth estimation algorithms. Machine learning (ML) is a common method used in
many research fields, and its early application in remote sensing is promising. In this study, we propose
a new and accurate approach based on the ML technique to estimate real-time snow depth and reconstruct
historical snow depth from 1987-2018. First, we trained the random forest (RF) model with advanced
microwave scanning radiometer 2 (AMSR2) brightness temperatures ($T_B$) at 10.65, 18.7, 36.5 and 89
GHz, land cover fraction (forest, shrub, grass, farm and barren), geolocation (latitude and longitude) and
station observation from 2014-2015. Then, the trained RF model was used to retrieve a reference dataset
with 2012-2018 AMSR2 $T_B$ data as the accurate snow depth. With this reference snow depth dataset, we
developed the pixel-based algorithm for the Special Sensor Microwave/Imager (SSM/I) and Special
Sensor Microwave Imager Sounder (SSMI/S). Finally, the pixel-based method was used to reconstruct a
consistent 31-year daily snow depth dataset for 1987-2018. We validated the trained RF model using the
weather station observations and AMSR2 $T_B$ during 2012-2013. The results showed that the RF model
root mean square error (RMSE) and bias were 4.5 cm and 0.04 cm, respectively. The pixel-based

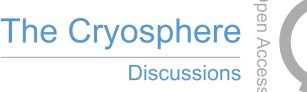

algorithm's accuracy was evaluated against the field sampling experiments dataset (January-March, 2018)
and station observations in 2017-2018, and the RMSEs were 2.0 cm and 5.1 cm, respectively. The pixel-
based method performs better than the previous regression method fitted in China (RMSEs are 4.7 cm
and 8.4 cm, respectively). The high accuracy of the pixel-based method can be attributed to the spatial
dynamic retrieval coefficients and accurate snow depth estimates of the RF model. Additionally, the
1987-2018 long-term snow depth dataset was analyzed in terms of temporal and spatial variations. On
the spatial scale, daily maximum snow depth tends to occur in Xinjiang and the Himalayas during 1992-
2018. However, the daily mean snow depth in Northeast China is the largest. For the temporal
characteristics, the February mean snow depth is the thickest during snowy winter seasons. Interestingly,
the January mean snow depth represents the annual mean snow depth, which plays an important role in
snow depth prediction and hydrological management. In conclusion, through step-by-step validation
using in situ observations, our pixel-based approach is available in real-time snow depth retrievals and
historical data reconstruction.
**1 Introduction**
Seasonal snow cover is an important parameter in the context of the Earth's hydrological cycle, the global
radiation balance, and climate system (Fernandes et al., 2009; Hernández-Henríquez et al., 2015; Derksen
et al., 2012; Kevin et al., 2017; Huss et al., 2017; Dorji et al., 2018). The latest Intergovernmental Panel
on Climate Change (IPCC) special report of 2018 stated that the cryosphere is very sensitive to climatic
changes, and extreme snow cover changes and melting caused by global warming were threatening
natural and human systems (Hoegh-Guldberg et al., 2018). Long-term snow cover records are crucial for
climate studies, hydrological applications and weather forecasts over the Northern Hemisphere (Gong et
al., 2007; Derksen et al., 2012; Safavi et al., 2017; Tedesco et al., 2016; Huang et al., 2017; Zhong et al.,
2018). A key parameter is the snow water equivalent (SWE), which describes the amount of water stored
in the snowpack as a product of snow depth and mean snow density (Dressler et al., 2006; Kelly et al.,
2009; Foster et al., 2011; Xiao et al., 2018; Takala et al., 2017; Tedesco et al., 2016). Fortunately, passive
microwave (PMW) signals can penetrate snow cover and provide snow depth estimates through volume
scattering of snow particles in dry snow conditions. PMW remote sensing also has the advantage of
sensing without the dependency of solar illumination and weather conditions (Chang et al., 1987; Foster



et al., 2011; Larue et al., 2017). In addition, there exists a long achieved historical spaceborne PMW data
dating back to 1978, allowing us to study seasonal snow climatological changes (Takala et al., 2011;
Takala et al., 2017; Santi et al., 2012). These superiorities make snow depth estimation from satellite
PMW remote sensing an attractive option.
However, there are two challenges when generating long-term snow depth data. The first challenge
is choosing the most suitable algorithm. The most widely used inversion algorithms are based on
empirical relationships between spaceborne satellite brightness temperature ($T_B$) differences (high
frequency sensitive to snow volume scattering ~37 GHz and low frequency insensitive to snow ~19 GHz)
and snow depth (Chang et al., 1987; Foster et al., 1997; Derksen et al., 2005; Che et al., 2008; Kelly et
al., 2003; Kelly et al., 2009; Chang et al., 2009; Jiang et al., 2014; Yang et al., 2019). However, these
algorithms are always not reliable in all regions using the fitted empirical constants (Davenport et al.,
2011; Derksen et al., 2010; Che et al., 2016; Takala et al., 2017; Yang et al., 2019). Subsequently, more
advanced algorithms that use theoretical or semiempirical radiative transfer models were developed
(Durand et al., 2006; Jiang et al., 2007; Tedesco et al., 2010; Takala et al., 2011; Picard et al., 2012;
Luojus et al., 2013; Che et al., 2014; Lemmetyinen et al., 2015; Metsämäki et al., 2015; Tedesco et al.,
2016; Huang et al., 2017; Larue et al., 2017; Pan et al., 2016; Pan et al., 2017; Saberi et al., 2017),
however, these algorithms were computationally expensive and required complex ancillary data or prior
knowledge to provide accurate predictions. These factors restrict the applications of these algorithms on
a global scale. Improving the performance of PMW retrieval algorithms by means of data assimilation
has also been investigated (Durand et al., 2006; Tedesco et al., 2010; Che et al., 2014; Huang et al., 2017).
Currently, the most representative operational assimilation system is the European Space Agency (ESA)
Global Snow Monitoring for Climate Research (GlobSnow) SWE product, which combines synoptic
weather station data with satellite PMW radiometer measurements through the snow forward model
(Helsinki University of Technology snow emission model, HUT) (Pulliainen et al., 1999; Pulliainen.,
2006; Takala et al., 2011; Luojus et al., 2013; Metsämäki et al., 2015; Takala et al., 2017). To avoid
spurious or erroneous deep snow observations, a mask is used in mountainous areas (Takala et al., 2011;
Luojus et al., 2013). Moreover, the product is generated in the Northern Hemisphere (> 35º N), which
excludes most parts of Qinghai-Tibetan Plateau (QTP). The algorithm may not be as feasible as empirical
algorithms in terms of real-time operation because of its sophisticated procedure and diverse inputs.
Currently, machine learning (ML) is being utilized in many different research areas, and its early



application in remote sensing fields is promising (Liang et al., 2015; Bair et al., 2018; Xiao et al., 2018;
Xiao et al., 2019). ML techniques can reproduce the nonlinear effects and interactions between variables
without assumptions of a functional form. The widely known ML algorithms include support vector
machine (SVM), artificial neural network (ANN) and random forest (RF). Among these methods, RF is
an ensemble method whereby multiple trees are grown from random subsets of predictors, producing a
weighted ensemble of trees (Breiman, 2001; Liang et al., 2015; Bair et al., 2018). RF is also robust against
overfitting in the presence of large datasets and increases predictive accuracies over single trees. The
method has been used in classification and prediction due to its proven accuracy, stability, and ease of
use (Bair et al., 2018; Belgiu et al., 2016; Rodriguez-Galiano et al., 2012; Qu et al., 2019).
The second challenge is how to take full advantage of the data from different sensors and rebuild a
long time series dataset. On the one hand, global snow estimates from PMW measurements are among
the longest satellite-derived climate records in existence, from the Scanning Multichannel Microwave
Radiometer (SMMR, 1978-1987), Special Sensor Microwave/Imager (SSM/I, 1987-2008) and Special
Sensor Microwave Imager/Sounder (SSMI/S, 2006-present) to NASA's Advanced Microwave Scanning
Radiometer for the Earth Observing System (AMSR-E, 2002-2011) and AMSR2 (2012-present)
(Knowles et al., 2000; Armstrong et al., 1994; Kawanishi et al., 2003; Imaoka et al., 2012). The
Microwave Radiation Imager (MWRI) onboard the Chinese FengYun-3 (FY-3) series of satellites (FY-
3A, 2008; FY-3B, 2010-the present; FY-3C, 2013-present; FY-3D, 2017-present) was designed for broad
meteorological and environmental applications (Yang et al., 2011). Subsequent satellites, FY-3E, 3F and
3G, are expected to be launched in the future until 2025. However, several consecutive generations have
different sensor calibration and design characteristics, which tend to result in uncertainties and
inconsistencies (Armstrong et al., 1994; Derksen et al., 2003; Cavalieri et al., 2012; Meier et al., 2011;
Okuyama et al., 2015). For example, the footprint size of AMSR2 has been improved compared to its
predecessors, and the grid $T_B$ is more representative for pixels ($25 \times 25$ km$^2$). The 10.65 GHz included
in the AMSR2 and MWRIs is more suitable for the estimation of deep snow cover (Derksen et al., 2008;
Kelly et al., 2009; Jiang et al., 2014). This frequency has been missed since the SSM/I substituted for the
SMMR and was not available until the Global Change Observation Mission (GCOM-W) AMSR-E was
operational. The SSMI(S) sensors, including SSM/I and SSMI/S, on the U.S. Defense Meteorological
Satellite Program (DMSP) satellites (F08, F11, F13, and F17) collect data at four frequencies (19, 22, 37,
85 or 91 GHz) from 1987 to the present. Although there is no 10.65 GHz frequency, the satellite sensors





and platforms possess similar configurations. Moreover, the latest dataset was reprocessed to complete
intersensor calibrations by Remote Sensing Systems Version (RSS V7), providing interconsistency of $T_B$
from the sensors (Armstrong et al., 1994). Thus, balancing the data consistency (SSM/I and SSMI/S) and
the advanced PMW instruments (AMSR2 and MWRI) is still an issue. To make use of the advantages of
both aspects, we propose a pixel-based method of snow depth reconstruction and real-time estimation
based on the RF model, where the RF model was trained using the 10.65-89 GHz satellite observations
(AMSR2) and other ancillary data. The estimated snow depth from the RF was used to develop a pixel-
based algorithm using 19.35 and 37 GHz for the SSMI(S).

9       The primary objective of this study is to test the RF model feasibility in estimating snow depth,

establish a pixel-based method to retrieve real-time snow depth and reconstruct historical snow depth
data (~31 years, from 1987-2018). The paper is organized as follows. The data and methodology are
presented in Section 2. In Section 3, the results are described, including the RF model test, RF model
training, development of a pixel-based model and long-term snow depth reconstruction. The discussion
is provided in Section 4, and in Section 5, we present our conclusions.
**2 Data and Methodology**
**2.1 Data**
(1) Satellite passive microwave measurements
There is a relatively long time series of remotely sensed PMW measurements (from 1978-present). Table
1 shows the characteristics of PMW remote sensing sensors. Among these sensors, AMSR2 has three
major advantages compared with other instruments: (a) $T_B$s from 10.65 GHz-89 GHz are available
compared to the SMMR, SSM/I and SSMI/S sensors; (b) it contains a newly added 7.3 GHz channel at
the C-band compared to the previous AMSR-E; and (c) the antenna is enhanced with a smaller footprint
size. Thus, the overall reliability has been improved to a certain extent. Therefore, in the first step, the
RF model was trained using the AMSR2 measurements to generate the reference snow depth. The
AMSR2 data are provided in the EASE-Grid projection with an equidistant latitude-longitude at a quarter
degree resolution since 3 July 2012 (http://gportal.jaxa.jp/gpr/). To avoid the influence of wet snow on
snow depth estimation, only the $T_B$ observations from nighttime overpasses (Descending, 1:30 a.m.) were
used in this paper (Chang et al., 1987; Derksen et al., 2010; Tedesco et al., 2016).



The SSMI(S) sensors provide $T_B$ data at 19.35, 23.235, 37, 85.5 or 91.655 GHz from 1987-present.
The data are available from the National Snow and Ice Center
(https://daacdata.apps.nsidc.org/pub/DATASETS). Both the vertical and horizontal polarizations are
measured, except for 23.235 GHz, where only the vertical polarization is measured. The satellite sensors
and platforms with similar configurations can reduce system errors, which is suitable for producing a
long-term consistent snow depth dataset. We used the dataset reprocessed by RSS, in which the
intersensor calibrations were completed. To avoid the influence of wet snow, only cold overpass data
were used. Notably, in this study, the difference between 19.35 (36.5) GHz and 18.7 (37) GHz was
ignored.
(2) In situ measurements
The weather station data were acquired from the National Meteorological Information Centre, China
Meteorology Administration (CMA). The snow depth measurement dataset used in this paper is from
689 stations throughout China (Fig. 1, left) from 2012-2018. The recorded variables include the site name,
observation time, geolocation (latitude and longitude), elevation (m), near surface soil temperature
(measured at a 5-cm depth, °C), and snow depth (cm). Notably, because of the harsh climate and complex
terrain, meteorological stations are few in the QTP, especially in the western part.
Quality control was conducted prior to using the data for developing the retrieval algorithm. The first
step was to select the records where the near surface soil temperature was lower than 0 °C. The second
step was to remove the sites if the areal fraction of the open water exceeded 30 % within a satellite pixel.
Finally, only ground-measured snow depths greater than 3 cm were used because the microwave response
to thinner snow cover at 37 GHz is basically negligible (Derksen et al., 2010; Tedesco et al., 2016). A
small number of points with extremely high snow depth values (greater than 70 cm) were also removed.
The snow depth distribution in the filtered subset is from 3-70 cm.
In addition, the field campaign supported by the Chinese snow survey (CSS) project was conducted
from January-March in 2018 to measure the snow depth transects in two satellite pixels in Xinjiang and
Northeast China (Wang et al., 2018). Figure 1 shows the two field sampling pixels in Xinjiang and
Northeast China. Table 2 shows the details of the snow field sampling work, including longitude, latitude,
altitude (m) and land cover types. The lack of canopy cover makes it an ideal study area for PMW remote
sensing. There are 26 and 21 sampling measurements within a coarse 25 km pixel in Xinjiang and
Northeast China, respectively. There were four days of snow depth transect measurements on January



21, 23, February 1 and March 9, 2018. For field sampling, measurements within each grid are averaged
to represent the ground truth snow depth.
(3) Land cover fraction
A 1-km land use/land cover (LULC) map derived from the 30-m Thematic Mapper (TM) imagery
classification was provided by the Data Center for Resources and Environmental Sciences, Chinese
Academy of Sciences (http://www.resdc.cn/). Because the 1-km LULC map was derived from 30-m TM
imagery, the map can be recalculated as the areal percentages of each land cover type in the 25-km grid
cells. In this study, the fractions of grass, barren, farm, forest, and shrub were calculated as inputs of the
RF model. The dataset is not described here; see Jiang et al. (2014) for more details. To avoid the
influence of water bodies and construction, the record was used only if the total fraction, including grass,
barren, farm, forest, and shrub, was greater than 60 %.
**2.2 Methodology**
(1) Stability test of RF
RF is an ensemble algorithm that was developed by Breiman in 2001. RF runs by constructing many
single decision trees to improve performance, which is much more efficient than traditional ML
techniques. The RF algorithm generally only requires two user-defined parameters, the number of trees
in the ensemble, and the number of random variables at each node. A particular advantage of RF is that
because of the presence of multiple trees, the individual trees need not be pruned, avoiding overfitting
(Breiman, 2001). In this paper, the RF method is trained to retrieve the reference snow depth dataset,
which is necessary to build the pixel-based model.
In general, the quality of the reference snow depth is determined by the RF model performance. In
this study, the number of variables selected at each node (split) is set to 4 (usually the square root of the
number of input variables) based on the number of input variables (Gislason et al., 2006; Belgiu et al.,
2016). The number of trees is set to 500 according to the out-of-bag (OOB) test because the errors are
stable when the number of decision trees is adequate. This finding agrees with previous studies
suggesting that a tree number of approximately 500 is generally sufficient (Belgiu et al., 2016; Cánovas-
García et al., 2015; Cánovas-García et al., 2017; Tsai et al., 2019). However, how many samples should
be inputted to the RF model? Specifically, is the performance of the RF model related to the training



samples? Thus, the RF's performance is tested in terms of different training datasets. The flowchart of
the test process is shown in Fig. 2.

3       There are 80000 pairs of samples from 1987-2004 (including PWM $T_B$ from SSM/I, land cover

fraction and in situ snow depth). Notably, the SSM/I $T_B$ pairs here are only used to test the number of
samples required, not the ultimate training data of the RF model. During this process, the number of
samples selected randomly is from 5000 to 80000 (step, 5000). A unified dataset from 2005-2006 is used
to evaluate the performance of the RF model. We consider three evaluating indicators (the root mean
square error (RMSE), bias and correlation coefficient) to illustrate the stability of the RF model.
(2) RF model training, reference snow depth and the pixel-based model
The main processing steps are described in detail in Fig. 3. To build the RF model, as shown in Table 3,
the training dataset is composed of fifteen predictors including land cover fraction (5), latitude (1),
longitude (1), AMSR2 $T_B$ (8) and one target - station snow depth (1) from 2014 to 2015 (45000 samples).
The data were used to validate the trained model in the period from 2012 to 2013. The PMW
measurements contain dual-polarized (H & V) $T_B$s in four channels: 10.65 GHz, 18.7 GHz, 36.5 GHz
and 89 GHz. All available channels on the AMSR2 are listed in Table 1. Specifically, the 6.925 GHz and
7.3 GHz channels are contaminated by radio frequency interference (RFI) and are not sensitive to
snowpack (Kelly et al., 2009; Rodríguez-Fernández et al., 2015). The 23.8 GHz channel is sensitive to
water vapor and not surface scattering, which introduces uncertainty to the estimation process. Typically,
the lower frequency (18.7 GHz) is used to provide a background $T_B$ against which the higher frequency
(36.5 GHz) scattering-sensitive channels are used to retrieve snow depth. However, the possibility that
deep snow can scatter 18.7 GHz radiation suggests that a lower frequency (10.65 GHz) is more suitable
to provide background information (Kelly et al., 2009, Derksen et al., 2008; Tedesco et al., 2016). The
89 GHz channel was added because of its penetrability of shallow snow. For shallow snow or fresh snow,
it is probably transparent for 36.5 GHz. Thus, the use of 89 GHz channels can greatly improve depth
retrieval for barren land (Jiang et al., 2014). The mixed-pixel problem is the dominant limitation on snow
depth estimation accuracy (Derksen et al., 2005; Kelly et al., 2009; Jiang et al., 2014; Roy et al., 2014;
Cai et al., 2017; Li et al., 2017; Li et al., 2019). Satellite $T_B$ usually represents several land cover types
due to coarse footprints (tens of km). Thus, we added the main land cover fraction as part of the training
dataset. Some previous studies have shown that latitude and longitude contribute to improving RF model
performance and present the spatial distribution of snow depth (Bair et al., 2018; Qu et al., 2019).





After the RF model was trained, it was validated with the AMSR2 $T_B$ and station snow depth of 2012-
2013. Then, the trained RF model was used to generate a relatively accurate snow depth dataset (hereafter
referred to as the reference dataset) with AMSR2 observations from 2012 to 2018 (Fig. 3, step 1). We
hypothesize that the snow depth estimates with the RF model are the most accurate ground truth available.
Then, the reference snow depth was used to establish a pixel-based algorithm using the $T_B$ gradient (19.35
GHz-37 GHz):
$$SD = Slope \times (T_{B19.35H} - T_{B37H}) + Intercept \qquad (1)$$
where the $Slope$ and $Intercept$ are dynamic coefficients for each grid. $T_{B19.35H}$ and $T_{B37H}$ are PMW
brightness temperatures in Kelvin (K) at horizontal polarization. SD is snow depth in centimeters from
the reference data. The development of a pixel-based retrieval method makes it possible to estimate real-
time snow depth without relying on the use of multiple sources of information.
The performance of the pixel-based method was also compared with the static linear-fitting algorithm
developed by fitting 19.35 and 37 GHz with the snow depth measurements with a constant empirical
coefficient over China (Fig. 3, step 2). The linear-fitting method is the modified Chang equation, which
was developed based on Chinese weather station observations and SSM/I $T_B$ for China (Che et al., 2008).
The equation is as follows:
$$SD = 0.66 \times (T_{B19.35H} - T_{B37H}) \qquad (2)$$
where the $T_{B19.35H}$ and $T_{B37H}$ are brightness temperatures for 19.35 GHz and 37 GHz at horizontal
polarization, respectively and 0.66 is the static fitting coefficient.
(3) The reconstructed snow depth product and validation
The reconstructed snow depth dataset from 1987 to 2018 with the pixel-based method was evaluated by
the in situ measurements from the weather stations (2017-2018) and the field snow transects from the
CSS (January to March 2018). Then, the spatiotemporal distribution of snow depth was analyzed (Fig. 3,
step 3). To ensure the possible dry snow cover, the reconstruction periods are the main snow winter
seasons (January, February, March, November, and December).
**3 Results**
**3.1 RF stability test**





Although RF has many advantages over other ML techniques, the performance is related to the number
of training samples. Moreover, the quality of the reference snow depth is determined by the performance
of the RF model. To conduct a complete test with enough samples, 80000 pairs of records from 1987 to
2004 were used to test the required size of the training samples. The results are shown in Fig. 4 after
several test runs. Figure 4a represents the RMSEs range from 5.1 cm to 5.4 cm with increasing samples.
Figure 4b shows slight fluctuations of bias between -0.2 and 0.2 cm. Figure 4c shows that the correlation
coefficient is as high as 0.79 and seems to be stable when the samples are up to 50000. In any case, the
figure shows that the RF model performs robustly in terms of the training sample subset. In other words,
the number of training samples has less influence on the prediction accuracy because of the sufficient
number (500) of single decision trees (Belgiu et al., 2016; Cánovas-García et al., 2015; Cánovas-García
et al., 2017; Bair et al., 2018). The test is very helpful for us to determine the number of training samples
because of limited training samples from AMSR2.
**3.2 RF model training and validation**
To obtain a spatially continuous and accurate reference snow depth dataset, the RF model was used to
find the nonlinear relationship linking the input data to the target. The input data are composed of the
AMSR2 $T_B$, land cover fraction and geolocation (Table 3). The target dataset used to train the RF is from
weather station observations in 2014 and 2015. The performance of the trained RF model was evaluated
by the weather station snow depth in 2012 and 2013. Figure 5a shows that the RMSE is 4.5 cm. The
determination coefficient is as high as 0.77. Figure 5b shows the spatial distribution of RMSE. The
pattern of the high RMSE is consistent with the mountains (Xinjiang: Altai Mountains and Tianshan
Mountains; Northeast China: Changbai Mountains and Xiao Hinggan Mountains), which means that the
accuracy is low in that location. Additionally, the large uncertainties in snow depth retrieval are
associated with forest cover in Northeast China, which agrees with the studies by (Cai et al., 2017; Li et
al., 2017; Roy et al., 2014; Liu et al., 2018). The RMSE in the QTP and South China is also large due to
patchy, shallow and wet snow (Dai et al., 2017, 2018; Yang et al., 2015). Figure 5c shows that it tends
to overestimate snow depth over shallow snow areas, especially in the QTP and South China. In these
areas, weather stations are sparsely distributed, and snowfall is ephemeral. The snow cover is as thin as
1~5 cm, which challenges the ability of PMW remote sensing. Figure 5d shows the spatial distribution
of relative errors (RMSE is divided by mean snow depth). The error in the shallow snow cover is higher



than that in the thick snow areas. This pattern is caused by the low mean snow depth. Similarly,
occasionally, a high RMSE does not mean a poor performance because the relative error is less than 20 %,
for example, for the sites in northern Xinjiang and the Heilongjiang Province.
Long-term snow depth datasets retrieved from the RF model and linear-fitting model are compared
with the station observations in three regions of China (Fig. 6). There are sixteen pixels, three pixels and
one pixel in Northeast China, Xinjiang and the QTP, respectively. The land cover types are mainly
farmland in Northeast China, grassland in Xinjiang and grassland in the QTP. In situ measurements of
mean snow depth are obtained for the sites within each region. The results show that the linear-fitting
method performance is unstable. It tends to underestimate snow depth at the beginning of the snow season
but overestimate the snow depth in the late winter. This is because the grain size and density of fresh
snow are very small, so the scattering effect is nearly negligible. Along with the seasonal evolution, the
snow particle grows (~2 mm), and the snowpack becomes denser (200~400 kg m$^{-3}$), which causes
stronger scattering effects. In situ measurements show that the snow cover is shallow in the QTP, even
less than 5 cm, which results in patchy snow cover (Dai et al., 2017). However, the snow depth was
overestimated, which may be due to the following reasons. First, the data with a depth thinner than 3 cm
were excluded from the training dataset. Second, a distinct meteorological characteristic of the QTP is
the large diurnal temperature range, which causes snow to undergo frequent freeze-thaw cycles and leads
to rapid snow grain growth and consequently a high $T_B$ difference (Durand et al., 2008; Yang et al., 2015;
Dai et al., 2017). Third, frozen soil is also a factor that reduces the accuracy of estimates in the QTP.
Both snow and frozen ground are volume scattering materials, and they have similar microwave radiation
characteristics, making them difficult to distinguish (Chang et al., 1987; Grody and Basist., 1996).
Figure 7 shows the spatial distribution of the monthly average snow depth (winter season, 2016). The
left figure is the station observation; the middle figure is the RF estimation; and the right figure is the
linear-fitting model estimation. The five rows present mean snow depths in January, February, March,
November and December. The patterns between the RF estimations and station measurements are similar,
especially in Northeast China and Xinjiang. In November, December and January, serious
underestimation occurs for the linear-fitting model. This is because fresh snow has little scattering effect,
and the forest canopy attenuates the ground signals (Che et al., 2016; Li et al., 2019). Moreover,
overestimation occurs in February and March due to strong scattering caused by snow microphysical
properties, such as snow grain size and density (Che et al., 2016; Dai et al., 2017; Yang et al., 2019). In



November and December, sites recording snow cover are very sparsely distributed in Tibet, Qinghai and
western Inner Mongolia. Thus, it is difficult to assess the performances of the two methods. Although
the station sites show snow cover in southern China, the snowpack identification method does not classify
snow as snow (Li et al., 2007; Liu et al., 2018). In February, there are many site records in central China,
including Gansu, Ningxia and Shanxi. The comparison demonstrates that the RF model tends to
overestimate snow depth in these areas. This is related to the sparse sites and ephemeral snowfall events,
which result in poor representativeness. The snow cover is as thin as 1~5 cm in these areas, which makes
PMW remote sensing weak for estimating snow depth. Another reason is that the sample record is
removed if the in situ snow depth is below 3 cm. Thus, training samples of the RF model also give
estimates higher than 3 cm. Additionally, snow depth estimation in the mountains remains a challenge
(Lettenmaier et al., 2015; Dozier et al., 2016). The RF model and linear-fitting method have sharply
different performances in the Himalayan range. Numerous studies have been conducted on the snow
cover over the QTP and have indicated that the snow cover frequency in the Himalayas is higher than
elsewhere, ranging from 80 % to 100 % during the winter seasons (Basang et al., 2017; Hao et al., 2018).
Additionally, Dai et al. (2018) showed that deep snow (greater than 20 cm) was mainly distributed in the
Himalaya, Pamir, and Southeastern Mountains. The spatial distribution of snow depth in spring (March,
April and May) and winter (December, January and February) showed that the annual mean snow depth
is greater than 20 cm in the Himalayas (Dai et al., 2018). The pattern based on reference Dai et al. (2018)
is similar to the results of the RF model in this study. Obviously, the linear-fitting method does not
capture the deep snow cover in the Himalayas.
**3.3 Pixel-based model and validation**
Based on the reference snow depth retrieved with the RF model (in Sect. 3.2) and $T_B$ gradient between
19.35 GHz and 37 GHz at horizontal polarization (Eq. (1)), the *Slope* and *Intercept* of the pixel-based
model are determined in Fig. 8a and 8b. The *Slope* and *Intercept* are set to 0 when there are no samples
for some pixels where it is impossible for snow to fall. The interpolation method (3×3 sliding window,
average value) is used to determine the *Slope* and *Intercept* in which the number of samples is between
3 and 10. The *Slope* is high in Northeast China and Northern Xinjiang. It is also high in the Himalayas
and the Pamir, where the snow cover is thick. The *Intercept* is low in unstable snow-covered areas,
including Inner Mongolia and central and South China. The RMSE between the reference data and



estimates is shown in Fig. 8c. The mean RMSE is approximately 3.2 cm. In most areas, the RMSE is less
than 5 cm. However, the RMSE is very high in South China, where snowfall is highly unlikely to occur.
From 2012 to 2018, there are no more than 3 snowfall events in South China. Thus, the *Slope* and
*Intercept* are directly set to 0.66 and 0, respectively. In northern Xinjiang and Northeast China, a high
RMSE occurs over the Tianshan and Altai Mountains, Changbai Mountains and Xiao Hinggan
Mountains. These areas not only have varied topography but are also covered with forest or shrub. The
correlation between the reference snow depth and estimated snow depth with the pixel-based model is
shown in Fig. 8d. Obviously, the pattern of correlation is in accordance with snow cover types. Stable
snow cover areas present high correlations (Xinjiang and Northeast China) due to dry and nearly full
coverage snow cover (Yang et al., 2019). The correlation is very low and even negative in most areas of
South China, which are shown in white in Fig. 8d.
In this study, a long-term snow depth dataset (1987 to 2018) was reconstructed with a pixel-based
model in Sect. 3.2. To evaluate the snow depth product, we use ground-based truth snow depth
measurements from two sources: weather station and field sampling. Weather station snow depth is
retrieved during the winter season from 2017 to 2018, independent of training samples of the pixel-based
model. Field measurements are taken from CSS, providing records of dense snow depth sampling within
a coarse pixel across Xinjiang and Northeast China in 2018 (Fig. 1). As shown in Fig. 9a, there is good
agreement between the snow depth estimated with the pixel-based model and the measured snow depth.
The RMSE is 2.0 cm, and the determination coefficient reaches as high as 0.91, which is much better
than the linear-fitting method coefficients of 4.7 cm and 0.52. The station data validation is shown in Fig.
9b. The error bar shows that the linear-fitting method tends to seriously underestimate (bias is -2.6 cm)
when the snow depth is over 10 cm. The pixel-based model overestimates the shallow snow cover (less
than 5 cm), but the overall accuracy is higher than the linear-fitting method.
The time series of snow depth retrieved from the pixel-based model and linear-fitting method are
compared with the station observations in three regions of China (Fig. 10). The results show that the
pixel-based model performs better than the linear-fitting method in Northeast China and Xinjiang. The
linear-fitting method tends to underestimate snow depth at the beginning of the snow season (November
and December) but overestimates the snow depth in the late winter (February and March). However, the
snow depth was seriously overestimated for the pixel-based method in the QTP. The reasons were shown
in Sect. 3.2. Most parts of the QTP are covered with shallow snow. Deep snow is distributed in the




Himalaya, Pamir, and Southeastern mountainous areas. However, there are no in situ observations in
these areas due to complex terrain and atmospheric conditions, resulting in validation failure.
**3.4 Spatial-temporal analysis of the reconstructed snow depth**
The spatial-temporal distribution of snow depth over China is analyzed based on a reconstructed snow
dataset (1987-2018). The time series of snow depth in different regions over China is shown in Fig. 11.
The black, green, blue and magenta lines represent daily mean or maximum snow depth in China,
Northeast China, Xinjiang and the QTP, respectively. Figure 11a shows that the daily mean snow depth
in Northeast China is larger than that in Xinjiang and the QTP for most years. The mean snow depth in
the QTP is the smallest (< 12 cm). Please note that the mean snow depth over the QTP is the highest in
1998, which aggravated major flooding in the area of the middle and lower reaches of the Yangtze River
(Dorji et al., 2018). Figure 11b shows the time series of daily maximum snow depth. The maximum snow
depth is most likely to occur in Xinjiang and the QTP, although the mean snow depth is large in Northeast
China. The maximum is usually distributed in the QTP Himalayas during these years, such as 1996, 1998,
1999, 2009, 2010 and 2015.
To show the monthly snow depth difference for every year, the time series of the yearly snow depth
for winter seasons is shown in Fig. 12. Because there are only two months (November and December)
and three months (January, February, March) of snow depth records in 1987 and 2018, respectively, the
period is from 1988 to 2017. The results show that the mean snow depths in February and March are
higher than the yearly average snow depth. The mean in November is smallest and below 10 cm during
the winter seasons. The mean snow depth in January is basically on behalf of the annual mean snow
depth but for individual years, such as 1988 and 1994. This is highly important for predicting snow depth
in hydrologic studies.
On the spatial scale, the time series of snow depth in different subregions is analyzed. Figure 13
shows that the annual mean snow depth in Xinjiang and Northeast China is above average over China.
The mean in Northeast China is the largest among the three subregions. However, the maximum snow
depth has a tendency not to occur in Northeast China. The yearly mean snow depth in the QTP is the
smallest among the three subregions. However, the maximum sometimes occurs in the QTP (Fig. 11).
Thus, the spatial pattern of snow depth in the QTP exhibits great heterogeneity (Fig. 7).





**4 Discussion**
**4.1 Spatial correlation and bias between the RF model and pixel-based method**
To obtain further insight into the ability of the pixel-based method to capture the temporal and spatial
variability in snow depth, it is essential to compare the pixel-based retrievals with respect to the reference
snow depth dataset retrieved with the RF model. Figure 14a shows a scatter plot of snow depth retrieved
by the RF model vs. the pixel-based method. The coefficient of determination is very high ($R^2$=0.83).
The pixel-based product displays a very strong correlation with the reference snow depth dataset. A
histogram of the bias (RF minus pixel-based method) distribution is shown in Fig. 14b and suggests that
the mean bias is very small (0.47 cm), and most biases are between -2 cm and 2 cm. Figure 14c shows
the time series of the spatial correlation ($R$) of retrieval RF with respect to the pixel-based method. The
mean value of $R$ is 0.91, which is a strong correlation between RF and the pixel-based method. The time
series of correlation show a seasonal oscillation, with slightly lower values for months during late autumn
(November) and early spring (March). This is because the snow cover is patchy and shallow in November,
challenging the relationship between satellite $T_B$ and snow depth (Dai et al., 2017; Yang et al., 2019). In
addition, snowfall is also ephemeral and occurs in the mountains. The results may be affected by
variations in the number of samples and the station representativeness. Thus, the reference snow depth
retrieved with RF may still be inaccurate. Another limiting factor in estimating snow depth from PMW
data is the presence of liquid water because of the relatively high air temperature in these months,
resulting in higher absorption and poor penetration depth. Consequently, the satellite observation is
mainly associated with the emissions from the wet surface of the snowpack. Therefore, in wet snow
conditions, snow depth retrieval is not possible (Chang et al., 1987; Foster et al., 1997; Derksen et al.,
2010; Tedesco et al., 2016). The time series of mean biases in Fig. 14c shows that bias is within ±1 cm.
In any case, the pixel-based method, which uses only satellite data as input, shows the robustness as its
performances are comparable to the performances of RF over the training period.
**4.2 Disadvantages and potential errors of the reconstrued snow depth**
There are no available in situ measurements over all of China to ensure that the training dataset is
statistically significant to perform spatial inversions once the RF is trained. Thus, the accuracy of the
pixel-based algorithm is uncertain in the mountains or high-altitude areas where few stations are
distributed. In addition, the problem of training the RF with in situ measurements is that the





measurements are point measurements while the satellite grids have a spatial resolution of $25 \times 25$ km$^2$.
Moreover, only the 19.35 and 37 GHz are H-pol. $T_{BS}$ values were used to yield the long-term
reconstructed snow depth through the pixel-based method. Comparing Fig. 5 and Fig. 9b, the diminished
underestimation of snow depth by the RF model for the 20-60 cm thick snow appeared again in the pixel-
based regression model. Therefore, some snow depth underestimation is still possible in the reconstructed
snow depth dataset.
**4.3 Variable Importance in RF Model**
RF can examine the predictor importance as an increased mean squared error which is calculated by
summing changes using every split for a predictor, then dividing by the total number of splits (Breiman,
2001; Bair et al., 2018). The larger this value, the greater the importance of the variable. Figure 15 shows
the importance of all the input independent variables in the RF model. The results indicate that $T_B$ at 36.5
GHz is by far the most important predictor, with values of 44 % and 43 % for horizontal and vertical
polarizations, respectively, showing that the PMW snow depth retrievals have significant predictive
power for dry snow cover. The third most important predictor is longitude, followed by latitude, which
makes the RF model more dependent on station data.
Figure 16 shows the spatial patterns of the reconstructed snow depth over China for 1992-2017 at
intervals of five years. The deep snow cover is mainly distributed in Xinjiang, Northeast China and the
QTP (Himalayas). Moreover, the distribution of snow depth is affected by topography (the digital
elevation model, DEM). For example, the elevations of the west and south QTP are higher than that of
the east QTP, so that snow cover is relatively thick there (Figure 16). This phenomenon could be ascribed
to two reasons: the sparsity of the sites and the significant geolocation (latitude and longitude). Figure 1
shows that the stations are sparsely and unevenly distributed in the QTP. Moreover, since most of the
stations are located in inhabited valleys, the representativeness of these in situ data is questionable
(Orsolini et al., 2019). Another reason is that inputs of the RF model include longitude and latitude,
which should contribute to the present spatial patterns of snow depth according to previous studies
(Belgiu et al., 2016; Qu et al., 2019, Xiao et al., 2018, Wang et al., 2019). In fact, the longitude and
latitude reflect the DEM information, which greatly affects the Plateau's vegetation, precipitation and
snowfall (Qu et al., 2019, Wang et al., 2019).
**4.4 Influence of land cover types on product accuracy**





The evaluation of the pixel-based method performance with station observations from 2017 to 2018
revealed that the snow depth product accuracy varies significantly between land cover classes (Table 4).
The grids are viewed as pure pixels where the land cover fraction is greater than 85 % (Jiang et al., 2014).
Densely forested regions tend to yield a higher RMSE (6.2 cm) and lower determination coefficient (0.43)
when compared to grassland and farmland (Table 4). RMSEs in open areas, such as grassland (5.5 cm),
farmland (4.2 cm) and barren (4.6 cm), are low due to no canopy influence on the satellite observations
(Derksen et al., 2005; Cai et al., 2017; Che et al., 2016; Li et al., 2017). The determination coefficient for
grassland is as high as 0.74, which shows that the snow cover is homogeneous and that the station snow
depth is representative of satellite pixels (Yang et al., 2019). The determination coefficient of barren is
0.35 because of shallow, patchy snow cover and poor station representativeness (Dai et al., 2018; Yang
et al., 2019). This study demonstrates that the underlying surface condition influences the snow depth
estimation with a pixel-based approach. One of the future developments to improve the product accuracy
will be training the RF model separately for each land cover class.
**4.5 RF model trained by snow emission model simulations**
In this study, the RF model's performance determines the accuracy of the reconstructed snow depth. The
input variable describing the snow cover is only snow depth. The more prior information there is on snow
cover, the better the performance of the RF model will be. To determine the ability of the RF model, the
microwave emission model of layered snowpack (MEMLS) is applied to simulate the $T_B$ with varying
snow parameters (Mätzler et al., 1999; Löwe et al., 2015; Pan et al., 2015). Table 5 shows the ranges of
variable parameters and constants. The snowpack is set as one layer. Then, 10000 combinations of
parameters are randomly chosen in the range by the computer, and these combinations are inputted to
MEMLS to simulate the multifrequency brightness temperatures (10, 18.7, 37 and 89 GHz at H and V
polarizations). The training dataset of the RF model is composed of $T_B$, snow depth, snow density and
correlation length. Finally, two-thirds of the samples are inputted to the RF to train the model. One-third
of the samples are used to test the performance in estimating snow depth.
To illustrate that more snow cover information can improve the accuracy of the RF model, two sets
of samples are inputted to the RF model. One set includes the 10-89 GHz observations, snow depth, snow
density and grain correlation length. Another set consists of 10-89 GHz brightness temperatures and
snow depth only. The measured snow depth is the initial input of the MEMLS. The estimated snow depth





is retrieved with the RF model. Figure 17a shows that more snow parameter inputs (snow density and
snow grain size) describing snow cover characteristics can improve the accuracy of snow depth
estimation. Otherwise, the scatter plots are dispersed, namely, there is a large RMSE between the truth
measurement and estimated snow depth (Figure 17b). Thus, the snow parameters retrieved with snow
models and measured in the field work are significant for improving the RF model. How to combine the
snow forward model with the ML method will be the focus of future work.
**5 Conclusions**
In this study, a novel approach called a pixel-based algorithm based on the RF model was proposed to
reconstruct snow depth using legacy PMW remote sensing satellite data. The RF model was trained using
AMSR2 $T_B$ and other auxiliary data. The validation showed that the RF model performs well in snow
depth estimations. The RMSE is 4.5 cm. The determination coefficient was as high as 0.77. Then, a pixel-
based model was built based on the reference snow depth that was retrieved with the RF model. The aim
was to reconstruct the long-term snow depth datasets from 1987 to 2018. Validation results with field
sampling data (weather station observations) show that the RMSE was 2.0 cm (5.1 cm), much better than
the linear-fitting method value of 4.7 cm (8.4 cm). Finally, a spatial-temporal analysis based on a long-
term snow depth dataset was conducted. On the spatial scale, daily maximum snow depth tended to occur
in Xinjiang and the QTP, while the mean snow depth in Northeast China was the highest. On a temporal
scale, the annual mean snow depth varied in February and March, and snow cover was the deepest among
winter seasons. Interestingly, the mean snow depth in January was basically on behalf of the yearly mean
snow depth, which is significant for predicting snow depth in hydrologic studies. In conclusion, a
spatiotemporally continuous snow depth product with a 31-year time series over China was obtained
from the pixel-based method. As discussed in Sect. 4, our reconstructed snow depth estimates are not
perfect. However, the reconstructed long-term product maintains high accuracy relative to others. In
addition to the historical data reconstruction, another merit of the presented approach is the ability to
provide real-time snow depth from satellite-based measurements, while the RF model that operates on a
daily basis is difficult and relies on the use of multiple sources of auxiliary data. We also realize that
efforts should still be made to solve the underestimation of deep snow cover and overestimation of
shallow snow cover areas. On the one hand, more prior knowledge of snow cover, such as snow cover



fraction, snow density, and snow grain size, is necessary to improve the RF model by means of the snow
forward model. In terms of the pixel-based method, two different $T_B$ differences ($T_{B37GHz}$-$T_{B89GHz}$ and
$T_{B19GHz}$-$T_{B37GHz}$) will be used to account for shallow and deep snow. On the other hand, a snow depletion
curve based on the relationship between snow depth and snow cover fraction will be used to improve the
snow depth retrievals in the QTP.
*Author contributions*. L. Jiang conceived and designed the study; J. Yang produced the first draft of the
manuscript, which was subsequently edited by L. Jiang, K. Luojus, J. Pan and J. Lemmetyinen; and M.
Takala, S. Wu, J. Pan and J. Yang contributed to the analytical tools and methods.
*Competing interests.* The authors declare that they have no conflicts of interest.
*Acknowledgments.* This study was supported by the Science and Technology Basic Resources
Investigation Program of China (2017FY100502) and the National Natural Science Foundation of China
(41671334). The authors would like to thank the China Meteorological Administration, National
Geomatics Center of China, National Snow and Ice Data Center and NASA's Earth Observing System
Data and Information System for providing the meteorological station measurements, land cover
products and satellite datasets.
*Data availability.* Satellite passive microwave measurements are available for download from
http://gportal.jaxa.jp/gpr/ and https://nsidc.org/. The in-situ measurements provided by China
Meteorology Administration (CMA) and Chinese snow survey (CSS) project are not available to the
public due to legal constraints on the data's availability. The land use/land cover (LULC) map is provided
by the Data Center for Resources and Environmental Sciences, Chinese Academy of Sciences
(http://www.resdc.cn/). The Shuttle Radar Topography Mission (SRTM) version 004 digital elevation
model (DEM) data with 90m resolution was obtained from http://srtm.csi.cgiar.org.

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

**List of Tables and Figures**
Table 1. Summary of the main passive microwave remote sensing sensors.

| Sensor | SMMR | SSM/I | | SSMI/S | AMSR-E | AMSR2 | MWRI | | |
|---|---|---|---|---|---|---|---|---|---|
| Satellite | Nimbus-7 | DMSP-F08 | DMSP-F11 | DMSP-F13 | DMSP-F17 | EOS Aqua | GCOM-W | FY-3B | FY-3C | FY-3D |
| On Orbit time | 1978-1987 | 1987-1991 | 1991-1995 | 1995-2008 | 2006-present | 2002-2011 | 2012-present | 2010-present | 2013-present | 2017-present |
| Passing Time | A: 12:00 D: 24:00 | A: 06:20 D: 18:20 | A: 17:17 D: 05:17 | A: 17:58 D: 05:58 | A: 17:31 D: 05:31 | A: 01:30 D: 13:30 | A: 01:30 D: 13:30 | A: 13:30 D: 01:30 | A: 22:00 D: 10:00 | A: 14:00 D: 02:00 |
| Frequency: Footprint (GHz): (km × km) | 6.6: 95×160 10.7: 60×100 18: 35×60 21: 30×50 37: 17×29 | | 19.35: 45×68 23.235: 40×60 37: 24×36 85.5: 11×16 | | 19.35: 42×70 23.235: 42×70 37: 28×44 91.655: 13×15 | 6.925: 43×75 10.65: 29×51 18.7: 16× 27 23.8: 18×32 36.5: 8×14 89: 4×6 | 6.925: 35×62 7.3: 35×62 10.65: 24×42 18.7: 14× 22 23.8: 15×26 36.5: 7×12 89: 3×5 | 10.65: 51 × 85 18.7: 30 × 50 23.8: 27 × 45 36.5: 18 × 30 89: 9 × 15 | | |

Table 2. Details of the snow field sampling data (location, attitude, altitude and land cover type).

| Snow sampling pixels in Xinjiang | | | | | Snow sampling pixels in Northeast China | | | | |
|---|---|---|---|---|---|---|---|---|---|
| No | longitude | latitude | altitude (m) | land cover | No | longitude | latitude | altitude (m) | land cover |
| 1 | 84.026 | 42.973 | 2400 | grass | 1 | 125.514 | 44.765 | 186 | farm |
| 2 | 84.047 | 42.977 | 2431 | grass | 2 | 125.434 | 44.762 | 195 | farm |
| 3 | 84.069 | 42.983 | 2436 | grass | 3 | 125.434 | 44.717 | 179 | farm |
| 4 | 84.094 | 42.988 | 2444 | grass | 4 | 125.512 | 44.722 | 181 | farm |
| 5 | 84.117 | 42.994 | 2389 | grass | 5 | 125.480 | 44.680 | 154 | farm |
| 6 | 84.128 | 43.003 | 2408 | grass | 6 | 125.509 | 44.678 | 164 | farm |
| 7 | 84.127 | 43.014 | 2415 | grass | 7 | 125.435 | 44.673 | 178 | farm |
| 8 | 84.134 | 43.021 | 2412 | grass | 8 | 125.442 | 44.634 | 160 | farm |
| 9 | 84.172 | 43.036 | 2415 | grass | 9 | 125.506 | 44.632 | 159 | farm |
| 10 | 84.201 | 43.047 | 2432 | grass | 10 | 125.373 | 44.768 | 196 | farm |
| 11 | 84.229 | 43.055 | 2408 | grass | 11 | 125.300 | 44.766 | 194 | farm |
| 12 | 84.263 | 43.058 | 2425 | grass | 12 | 125.299 | 44.727 | 207 | farm |





| 13 | 84.286 | 43.061 | 2431 | grass | 13 | 125.375 | 44.724 | 176 | farm |
| 14 | 84.131 | 43.000 | 2412 | grass | 14 | 125.365 | 44.681 | 192 | farm |
| 15 | 84.129 | 42.988 | 2430 | grass | 15 | 125.344 | 44.680 | 195 | farm |
| 16 | 84.142 | 42.973 | 2470 | grass | 16 | 125.313 | 44.650 | 206 | farm |
| 17 | 84.142 | 42.989 | 2450 | grass | 17 | 125.291 | 44.586 | 188 | farm |
| 18 | 84.170 | 42.933 | 2520 | grass | 18 | 125.361 | 44.588 | 165 | farm |
| 19 | 84.189 | 42.912 | 2510 | grass | 19 | 125.450 | 44.584 | 158 | farm |
| 20 | 84.188 | 42.914 | 2510 | grass | 20 | 125.517 | 44.585 | 162 | farm |
| 21 | 84.217 | 42.887 | 2470 | grass | 21 | 125.370 | 44.625 | 185 | farm |
| 22 | 84.218 | 42.887 | 2500 | grass | | | | | |
| 23 | 84.234 | 42.872 | 2450 | grass | | | | | |
| 24 | 84.250 | 42.851 | 2420 | grass | | | | | |
| 25 | 84.266 | 42.858 | 2480 | grass | | | | | |
| 26 | 84.266 | 42.858 | 2440 | grass | | | | | |

2    Table 3. Predictor and target variables of the RF model

| Name | Description | Number |
|------|-------------|--------|
| **Predictors** | | |
| 10.65 GHz | | |
| 18.7 GHz | AMSR2 brightness temperature at V and H | |
| 36.5 GHz | polarizations | 8 |
| 89 GHz | | |
| Grass | | |
| Farm | | |
| Forest | Land cover fraction ranging from 0 to 100 | |
| Shrub | retrieved from 1-km LULC map | 5 |
| Barren | | |
| Latitude (°) | Geolocation of weather station | 2 |
| Longitude (°) | | |
| **Target** | | |
| SD (cm) | Station snow depth | 1 |

4    Table 4. Summary of land cover effects on the reconstructed product accuracy.

| Land Cover Types | RMSE (cm) | Bias (cm) | Correlation ($R^2$) | Samples |
|------------------|-----------|-----------|---------------------|---------|
| Forest | 6.2 | -0.81 | 0.43 | 2680 |
| Grass | 5.5 | -0.18 | 0.74 | 2763 |
| Farm | 4.2 | 0.51 | 0.58 | 5255 |
| Barren | 4.6 | 0.71 | 0.35 | 553 |



Table 5. The main input parameters of MEMLS.

| Ranges and steps of variables used for the simulation of TB by MEMLS | | | | |
|---|---|---|---|---|
| Items | Snow Depth | Snow Density | Correlation Length | Snow-ground Reflectivity |
| Range | 1~100 cm | 50~400 kg m$^{-3}$ | 0.01~0.5 mm | H-polarization: 0.2 |
| Interval | 1 cm | 10 kg m$^{-3}$ | 0.05 mm | V-polarization: 0.07 |
| The constant values of snow parameters used for the simulation of TB by MEMLS | | | | |
| Items | Liquid Water Content | Salinity | Incidence Angle | Ground & Snow Temperature |
| Value | 0 | 0 | 55° | 270 K |

Figure 1. Spatial distribution of the weather stations and land cover types in study area. Black points are
station sites where snow depth recorded is greater than 0 cm. Right attached figures describe the
configuration of the snow sampling experiment. The red marker represents the snow sampling points
within one pixel ($25 \times 25$ km$^2$). There are 26 and 21 snow measurements in Xinjiang and Northeast China,
respectively.

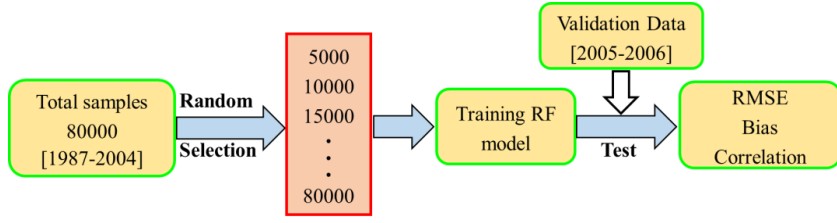

Figure 2. The test process flowchart for the RF model.



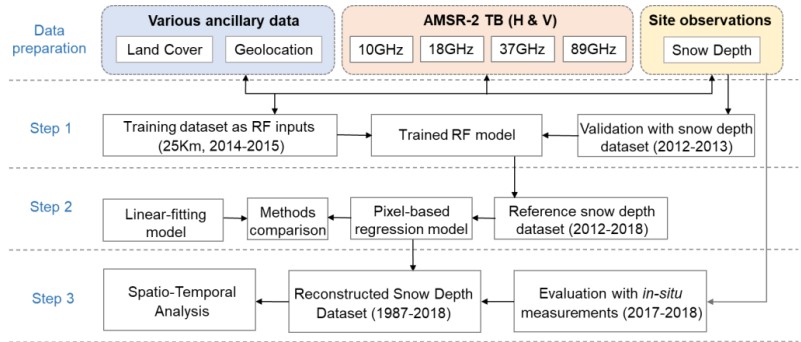

Figure 3. Flowchart of the snow depth reconstruction in this study.

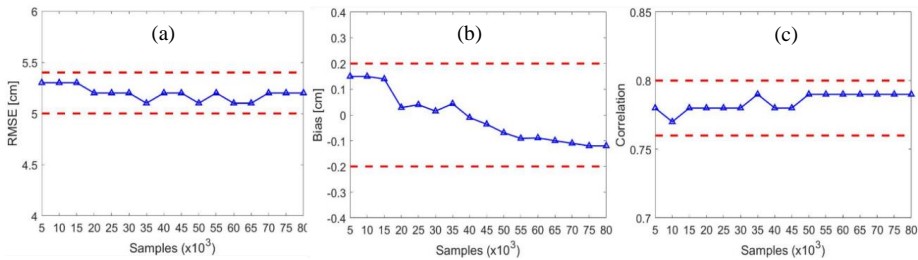

Figure 4. RF stability for the numbers of training samples: (a) RMSE; (b) bias and (c) correlation
coefficient.

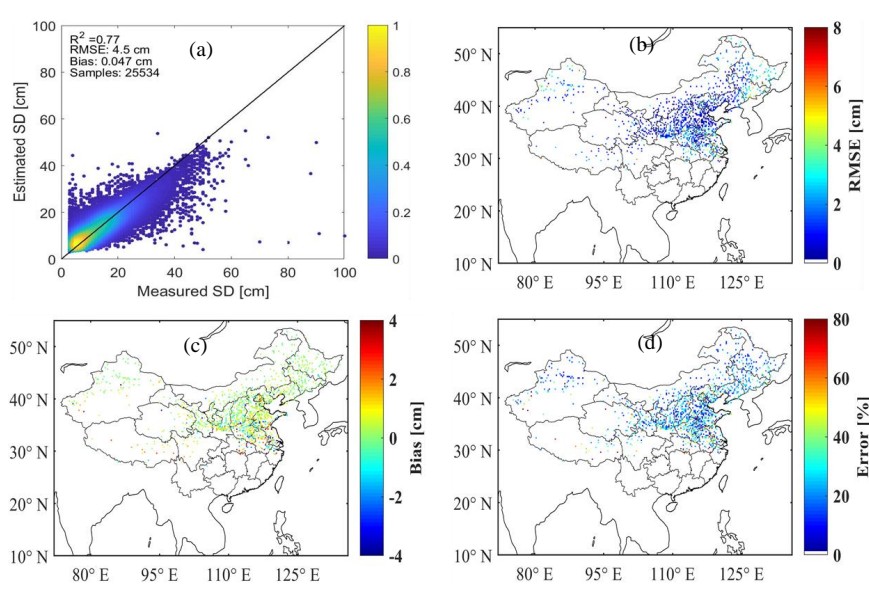

Figure 5. Validation of trained RF model: (a) scatterplots, (b) RMSE [cm], (c) bias [cm], and (d) relative
Error [%].

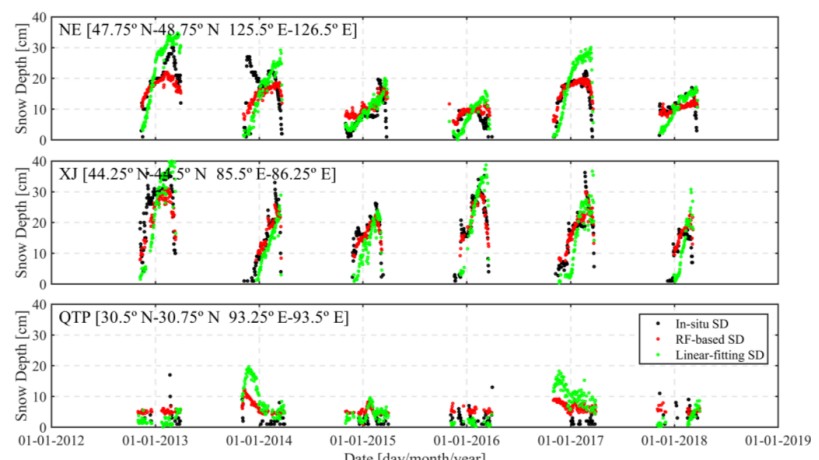

3     Figure 6. Snow depth comparison during winter seasons for the 2012-2018 period. Green dot: RF model;

4     red dot: linear-fitting model; black dot: station observations. There are sixteen pixels (Northeast China,

5     NE), three pixels (Xinjiang, XJ) and one pixel (QTP) in the three regions.
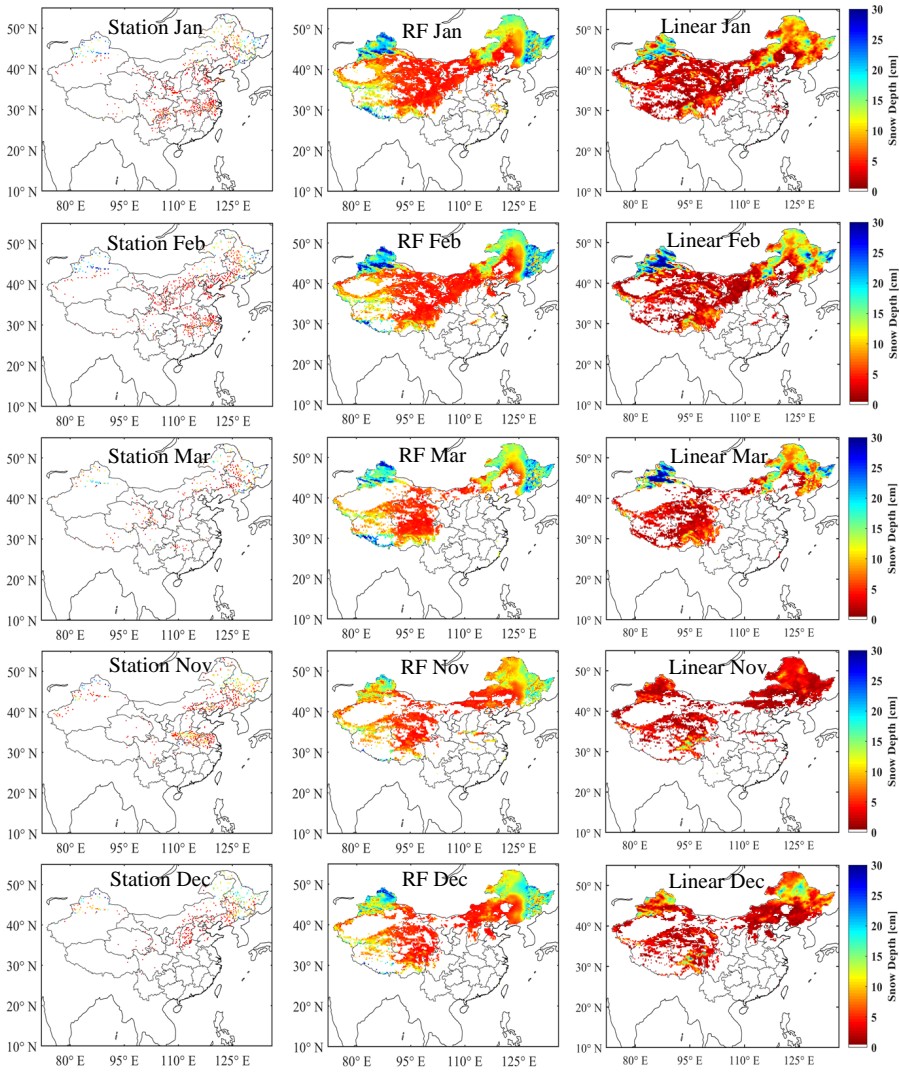

Figure 7. Spatial distribution of the monthly mean snow depth for January (Jan, first row), February (Feb, second row), March (Mar, third row), November (Nov, fourth row), and December (Dec, fifth row), 2016. Left: Station observations; Middle: RF estimations; Right: linear-fitting retrievals. The color scale denotes snow depth in centimeters, which ranges from 0 to 30 cm.


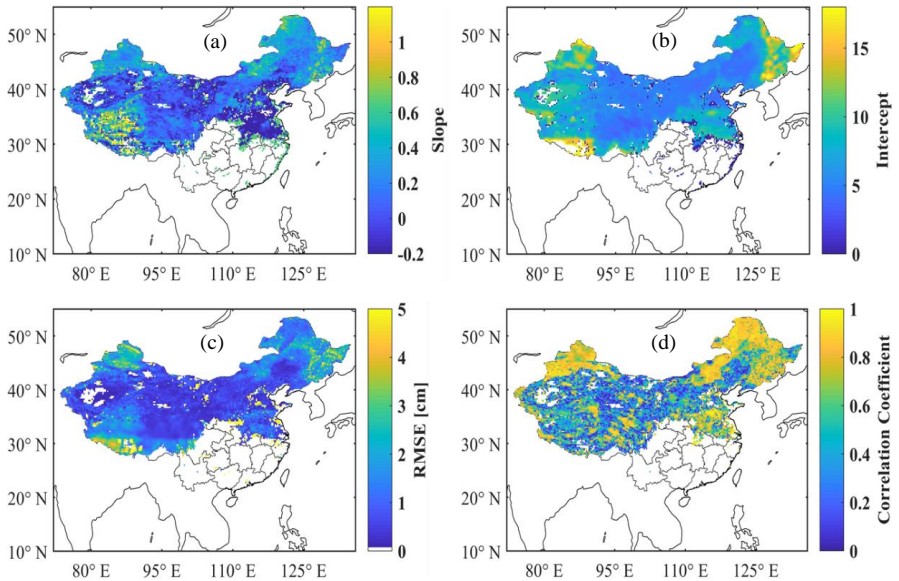

Figure 8. Spatial distribution of pixel-based coefficients: (a) *Slope*; (b) *Intercept*; (c) RMSE; and (d)
correlation coefficient. The RMSE and correlation coefficient are calculated with the reference snow
depths and retrievals of the pixel-based method.

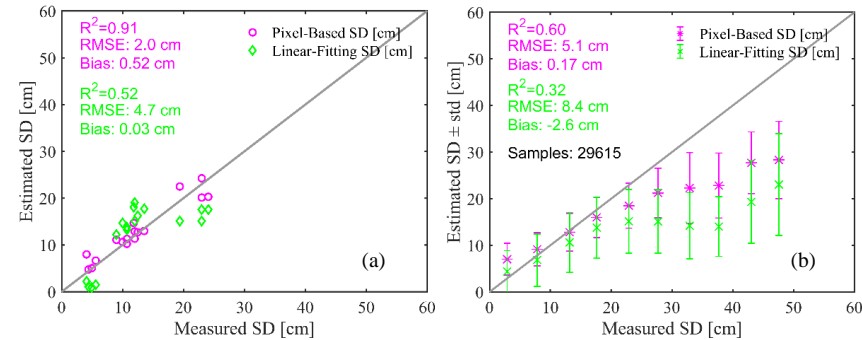

Figure 9. Pixel-based method validation with two sources: (a) field sampling and (b) weather stations.
Green and magenta markers denote linear-fitting method and pixel-based model, respectively.

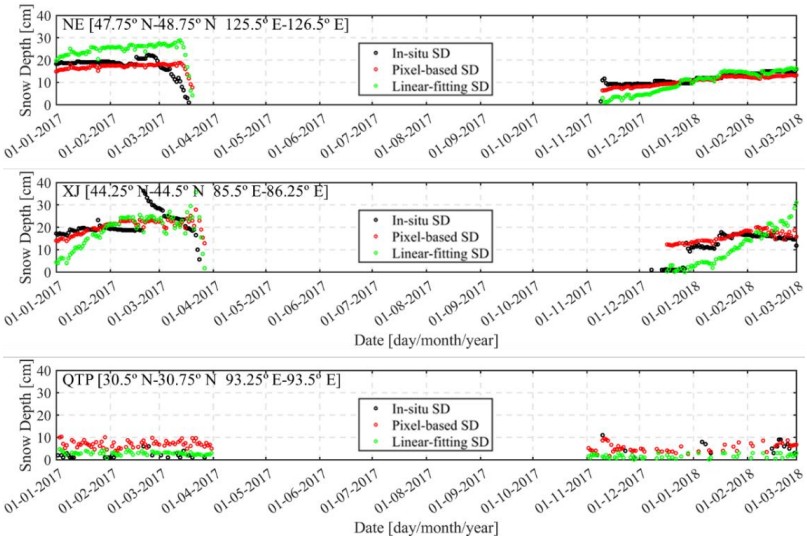

Figure 10. Snow depth validation during the winter seasons for the 2017-2018 period. Black dots: station
observations; red dots: pixel-based model; green dots: linear-fitting model.

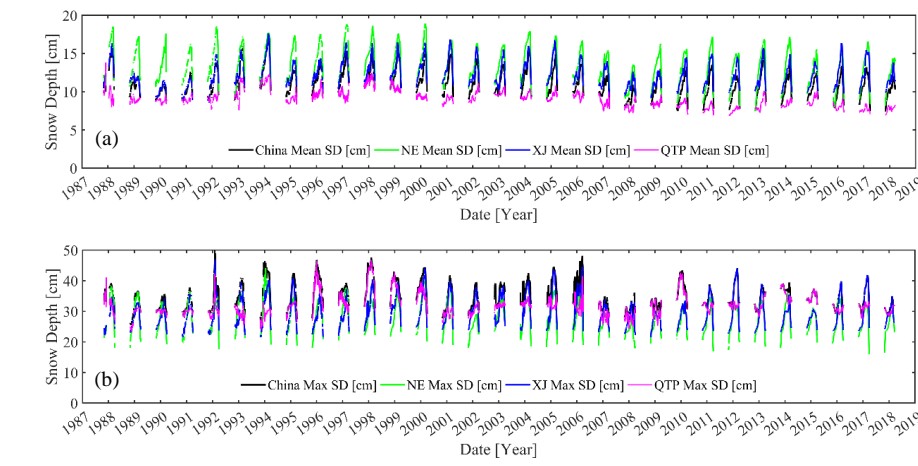

Figure 11. The time series of snow depth over China during winter seasons: (a) mean snow depth and (b)
max snow depth. NE: Northeast China; XJ: Xinjiang; QTP: Qinghai-Tibetan Plateau.

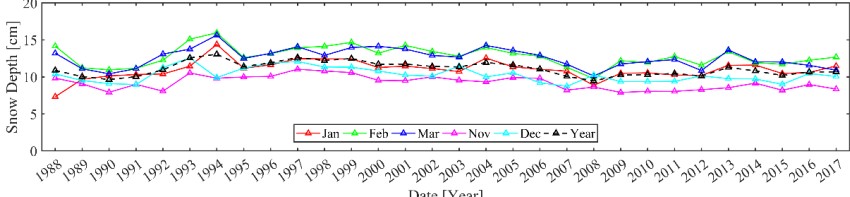



Figure 12. Time series of annual mean snow depth over China for the winter seasons. Black dashed line:
yearly; Red line: January (Jan); Green line: February (Feb); Blue line: March (Mar); Magenta line:
November (Nov); and Cyan line: December (Dec).

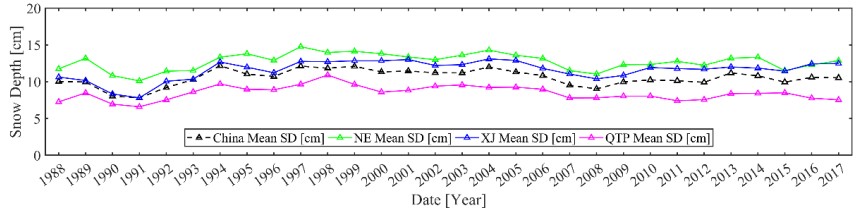

Figure 13. Long-term annual mean snow depth over different subregions. Black dashed line: China;
Green line: Northeast China, NE; Blue line: Xinjiang, XJ; and Magenta line: QTP.

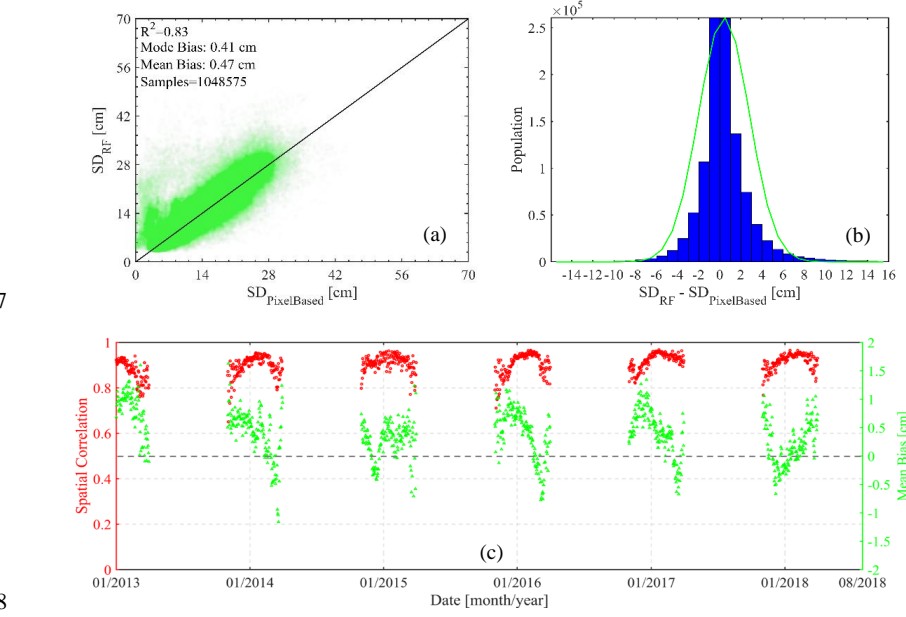

Figure 14. Comparison of retrievals between the RF model and pixel-based method: (a) scatter plot of
snow depth; (b) histogram of bias, $SD_{RF}$ - $SD_{PixelBased}$; and (c) temporal series of spatial correlation and
mean bias. The histogram bin width is 1 cm.


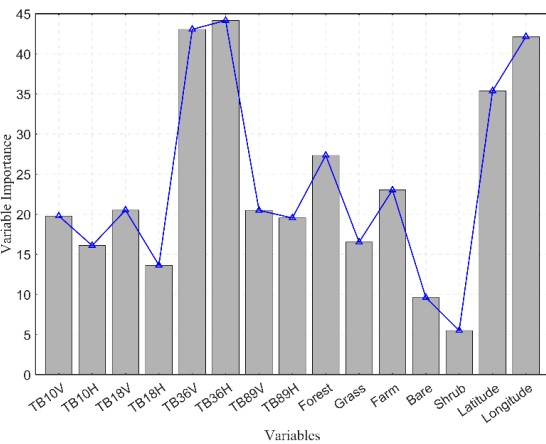

Figure 15. Predictor importance of the RF model based on increased mean square error. The variable
importance is based on the concept that a variable associated with a considerable reduction in prediction
accuracy is excluded. The larger the MSE, the greater the importance of the variable is.

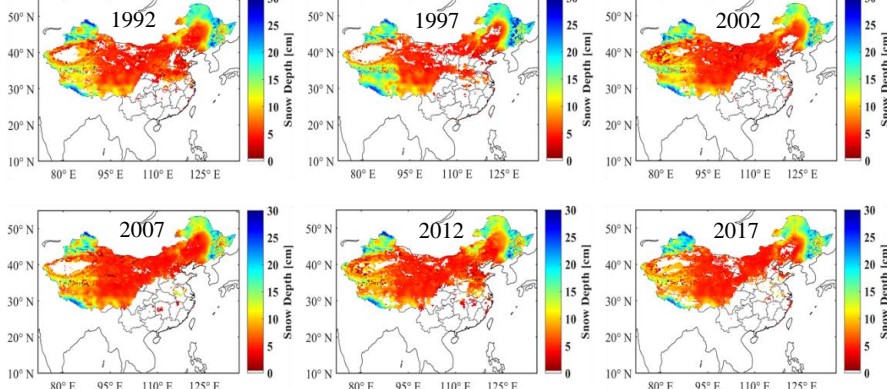

Figure 16. Reconstructed annual mean snow depth for 1992-2017 at five-year intervals.

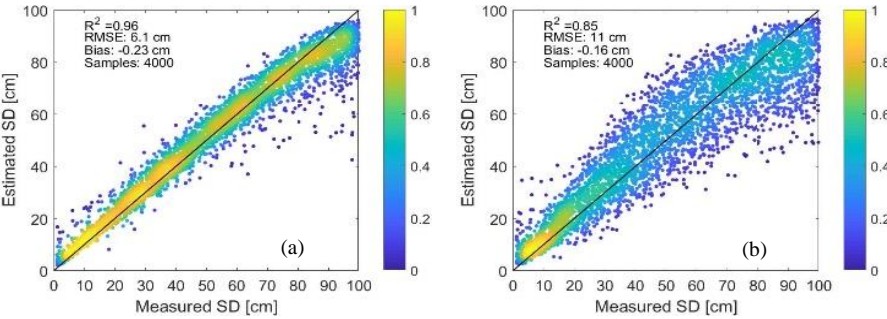

Figure 17. The performance of RF with two validation datasets from the MEMLS model: (a) 10-89 GHz
brightness temperatures, snow depth, snow density and correlation length; and (b) 10-89 GHz brightness
temperatures and snow depth. The color scale represents the data density of scattered points, which range
from 0 to 1.