# Peer review of "Snow Depth Estimation and Historical Data Reconstruction Over"

_The Cryosphere, 2019_

## Referee Comment (RC1) · Divyesh Varade (Referee) · 19 Nov 2019

Snow depth estimates are significant for the assessment of the hydrological potential of the snowpack. The application of machine learning tools provides us with a means to derive new depth estimates from a trained model. The methods for the modeling of snow depth using remote sensing data are predominantly based on passive microwave data with much higher repeatability and spatial coverage than InSAR data, rendering such analysis suitable for the monitoring of the snow accumulation. I thus, consider this work to be significant.

Overall, the manuscript is organized and written neatly and represented in a well-

structured manner. The language is mostly appropriate except for a few sentences which are not easily understandable. There are some claims and statements made by the authors that lack references or evidence.

This work is appreciable in the extent of the analysis performed by the authors, in particular for the time series evolution of the snow depth in some of the major provinces in China. However, the manuscript also presents some weaknesses in the methodology, experiments, and particularly the validation. Specific comments are as follows.

Major issues

1. The authors have not clearly stated the novelty of their proposed method. In my opinion, the novelty of the proposed method is in the design of the regression model using the Random Forests i.e. the step -1 in Figure 3 and its application for the modeling of snow depth. The other steps are similar to the methodology proposed in – Jiang, L., Wang, P., Zhang, L. et al. Sci. China Earth Sci. (2014) 57: 1278. https://doi.org/10.1007/s11430-013-4798-8

2. Why the Random Forest is used, in contrast to better alternatives such as deep neural networks? The authors claim that RF is superior to SVM and ANN, is there any documented evidence regarding RF to be superior to SVM or ANN in link with modeling of geophysical parameters similar to snow depth? Deep learning for classification and regression has been found very useful in recent literature. What is the reason that the authors use RF instead of deep neural networks? Please provide evidence for this or perform additional experiments to prove that RF-based estimates are superior to SVM, ANN, and deep NN based estimates.

3. In both cases, steps 1 and 3, the authors use only a single year data for validation. This neither provides enough points for validation nor any comprehensive inferences from the validation results.

4. The datasets used for training and testing have some issues. The authors have

shown how the actual depth has varied through the years 1987-2019. But for training only data till 2004 was used. The trends from Figures 10 and 11 show a marginal decrease in the mean snow depth. Would it not be better to use data from every two year or alternate year for training the RF. Similarly for testing, the authors use data from the only year 2012-13 for model testing and 2017-18 for testing the final results. This is not sufficient to develop a comprehensive interpretation of the results.

5. In section 3.2, the correlation coefficient is 0.77. Is this satisfactory enough to be used to generate the reference dataset from the RF model? A majority of data are below 10 cm snow depth, then an error of 4.5 cm is significantly high. To have a better understanding of the modeled results, it is vital that we observe the accuracy for the points of higher snow depth also. Particularly, when there is a very high snow depth different for the regions QTP and the others. The validation should be carried out for these regions separately. I suggest the authors show a histogram of the data and also carry out a separate fit for points of snow depth >10cm or perform a case by case fit with respect to the study area. A significant concern is that in the case of shallow snow (<10cm), is the brightness temperature actually representative of the contributions from the shallow snowpack or the underlying ground. This requires further investigations. This is important since the bulk of the data is within the 0-10 cm range. Another concern is that there are very few points with snow depth >40cm. In several locations in the Himalayas, the peak snow depth is usually around 1m or more. Thus, the applicability of the proposed method or the transferability of the proposed method to other areas, in these cases, is in question.

6. The authors observed higher errors for shallow snow depth, but the manuscript lacks any discussion on the contributions from the underlying ground layer to the passive microwave brightness temperature in case of shallow snow depth. The authors have simply added some references. A discussion is required in the manuscript on the sensitivity of snowpack thickness and stratigraphy towards the passive microwave brightness temperature.

7. Page 12, L25-27: Does this mean 3-10 samples in (3x25)x(3x25) sq. km area? This is not clear to me. I think the authors are referring to measurements from field campaigns or weather stations as samples. In this case, the number of samples is very small per the averaging window. Please provide references for this.

8. Figures 9a and 9b. There are very few samples used for validation in these figures. Further, these samples are discontinuous (Figure 9a) and therefore, this should not be used as the basis for ascertaining the performance of the proposed method, since due to the distribution of the points, it is expected that the fit will provide better results. The authors may perform other significance tests such as Nemar's test, but the fact remains that the validation data is not really comprehensive. The data shown in Figure 9b is much better for assessment, as it is continuous. But why only 10 points? Earlier it was shown that several ground stations exist in the area. I suggest the authors also use data from other years in their validation scheme, as the results shown at present are not convincing. Why is the modeled snow depth showing very less sensitivity between 20-40cm (nearly constant) and again afterward? This is an issue that requires investigation.

9. In section 4.5, the selection of sample size for training and testing is reversed. Since the MEMLS requires auxiliary information, which is seldom available, the training samples should be much less than the validation samples. This validation strategy is not convincing. From the discrepancy in the training and testing samples, it is already expected that the model accuracy would be high. Minor issues

Finally, some minor corrections in language are also required. Some specific comments are as follows. The authors should check the manuscript for similar errors.

Page 02, L7: " the Himalayas during….". The Himalayan ranges are very long and are shared by several countries. Please specify which Himalayan ranges the authors are referring to here. I do not agree with the statement that mean snow depth is maximum in Xinjiang for the entire Himalayan range. Please provide references for this.

Page 02: L8-11: These are documented facts in literature for several other locations, however. Thus, the authors should strictly restrict their inferences to their own findings and not speculate. Thus, here the sentence should be specific to the study area in the manuscript.

Page 02, L11-13: The sentence "In conclusion. . .." is not clear. Please rephrase.

Page 02, L24: "mean snow density". I believe the authors are here referring to mean stratigraphic snow density". Please correct this.

Page 03, L17-18: "however, these. . .". Is there any evidence that the RTM based methods are computationally more expensive than machine learning-based methods. In my opinion, both depend on the selection of the parameters. For example, an RF with substantial input and a high number of trees may be as expensive computationally. If there is no documented evidence on this, please remove this statement.

Page 06, L 28-29: "The lack of ..". This is only true when observed from the perspective of the spatial resolution of 25x25km. However, the topography of these pixels is not shown to the readers. The authors should show a high-resolution map of these pixels in Figure 1 also.

Page 11, L 11-13: Please correct the range as 200-~350 kg/m3 and provide a reference, for example- Meløysund, Vivian, Bernt Leira, Karl V. Høiseth, and Kim R. Lisø. 2007. "Predicting snow density using meteorological data." Meteorological Applications 14 (4): 413–23. doi:10.1002/met.40.

Page 17, L20: "The snowpack is set ..". This should be the snowpack is assumed to comprise a single layer indicating a semi-infinite medium. This is a common assumption in electromagnetic modeling of the snowpack. Please add references to this.

Figure 1: This needs to be revised. Firstly, the authors use 3 areas for their study which have not been shown on the large map. Secondly, the two pixels mentioned previously should be shown at a higher resolution. Third, write in captions what the color bar

[Figure]

represents, is it elevation? Finally, the pixels shown should also have a lat-long grid and scale bar.

Figure 7: Why is the number of points and their locations changing in the maps showing stations. I believe this should remain fixed irrespective of the month. If there is no snow at some of the stations which have been omitted, these should be shown with either a different symbol or a color.

Figure 8: The images are distorted. It appears as if they were stretched manually to fit some size.

Figure 9/Table 4 and several other instances: The R2 and R, i.e. the determination coefficient and the correlation coefficient, respectively, are two different parameters and have been used interchangeably with similar symbols in the manuscript, which makes it difficult to judge the accuracy of the results.

---

## Referee Comment (RC2) · Anonymous Referee #2 · 25 Nov 2019

**Review Report**

Journal:            The Cryosphere

Journal's Ref.:     tc-2019-161

Title:              Real-time snow depth estimation and historical data reconstruction over China based on a random forest machine learning approach

Authors:            Jianwei Yang, Lingmei Jiang, Kari Luojus, Jinmei Pan, Juha Lemmetyinen, Matias Takala, Shengli Wu

Date:               2019-11-25

Recommendation: Major revisions are needed

**Aim of the manuscript**

**[1]**    The aim of the manuscript is (a) to test random forests in estimating snow depth in a remote sensing application and (b) to reconstruct historical snow depth in China in the period 1987–2018 (see page 5, lines 10–14).

**[2]**    The procedure of the manuscript is presented in Figure 3.

**General evaluation**

**[3]**    The aim of the manuscript (in particular the reconstruction of historical snow depth) as well as the use of a big dataset justifies its publication.

**[4]**    The procedure followed in the manuscript is complicated, while I think that some steps are unnecessary and a more straightforward approach to the problem would achieve comparable (or even better results).

**[5]**    Regarding the algorithmic part of the manuscript, I have some recommendations to justify certain choices of the manuscript and highlight some advantages and drawbacks of random forests (regarding most minor comments on the algorithmic part, e.g. parameters of random forests, variable importance, number of predictor variables and more, as well as why one should use random forests instead of another algorithm, please consider reading the random forests review by Tyralis et al. 2019a for more details).

**[6]**    Furthermore, I think that the manuscript is wordy at some Sections, for instance explanation of Figures.

**[7]**    Perhaps the reconstructed dataset could be made available online increasing the value of the manuscript.

**[8]**    Some comments which should be discussed / addressed in the manuscript follow.

**Major comments**

**[9]**   Page 8, line 10 – page 9, line 25: In general, I think that the procedure described here is complicated, while some steps may be unnecessary. In particular:

a.   Random forests are fitted using 15 predictor variables in the period 2014–2015 (page 8, lines 11, 12) and then they are validated in the period 2012–2013. I do not understand the scope of this validation, considering that parameters of the algorithm have been defined earlier.

b.   Random forests are used to predict snow depth in the period 2012–2018. Then a linear model is trained in the predictions of the period 2012-2018 using two predictor variables. The trained linear model is used to predict snow depth in the period 1987-2018.

In my opinion it would be more straightforward to train random forests in the period 2014-2015 using two predictor variables and then predict in the period 1987-2018. Another straightforward option would be to train a linear model in the period 2014-2015 and then predict in the period 1987-2018.

Instead, following the two-stage procedure of the manuscript, a dataset, obtained by some predictions, is used to train a new model. In these procedures uncertainties are introduced (since the dataset obtained by random forests is an approximation of the true snow depth) which are transferred to the second stage prediction. I understand that this approach gives a rich dataset to do the second stage training, however I think that the induced uncertainties are not compensated by the bigger dataset. Perhaps the manuscript could justify this approach by performing some comparisons between the one and the two-stage approaches in the period 2012-2018 or just completely use the straightforward approach.

c.   Perhaps the approximation of equation (2) is suboptimal because it is based on data before 2008, while it does not include the intercept parameter. Given the big magnitude of the dataset, it is surprising that a one-parameter linear model (equation 2) would be preferable to the two-parameter model of equation (1).

**Minor comments**

**[10]**   Page 2, lines 15 – 20: A proper assumption for applying random forests is stationarity. Furthermore random forests do not predict outside the range of the training sample. Therefore, the assumption of global warming is not compatible with random forests.

**[11]**   Page 6, line 1: SSMI/S provides data in the period 2006-present according to Table 1.

**[12]**   Page 7, lines 16 – 17: Random forests parameters are more than two.

**[13]** Page 7, lines 21 – 27: In general the default values (in the software implementation) of random forests parameters are good.

**[14]** Page 7, lines 21 – 27: In general it is suggested to use as high number of trees as computationally feasible. However, indeed the number of 500 trees is high enough in most applications.

**[15]** Page 7, line 27 – page 8, line 2: In general the larger the dataset, the better the predictive ability of a regression algorithm.

**[16]** Page 10, lines 8–12: By increasing the size of training sample one would expect that the performance of predictive algorithm would increase.

**[17]** Page 11, lines 4, 5: Which linear model?

**[18]** Page 11, lines 22–24: The comparison between random forests and the linear model is unfair considering that the latter uses less predictor variables.

**[19]** Page 12, lines 25–27: This procedure is not clear.

**[20]** Page 13, lines 3, 4: I do not understand why assigning values to the slope and intercept.

**[21]** Page 16, lines 8–11: It is not clear which period was used to compute variable importance.

**[22]** Page 16, lines 24–28: Perhaps the information added by the longitude and latitude predictor variables is already included in the remaining predictor variables (see e.g. a similar application in Tyralis et al. 2019b). In the latter study, the predictive performance was examined by comparing models with and without longitude and latitude, and the effect of coordinates was found insignificant. Perhaps, computing variable importance and predicting performance would give some explanations on the value of the remaining predictor variables and make the model less dependent on the proximity of nearby stations.

**[23]** Page 18, lines 1–3: In general one would expect that using more predictor variables related to the dependent variable of interest would improve the trained model. Furthermore, redundant predictor variables slightly affect random forests.

**[24]** Figure 6: Figures should be numbered and respective explanations should be added in the caption.

**[25]** Regarding the implementation of random forests, some of their disadvantages and their impact in the results of the study can be discussed (see a list of disadvantages in Tyralis et al. 2019a), e.g. they do not extrapolate outside the training range, variable importance metrics are not always reliable, as they are affected by high correlations and interactions, and more.

**[26]** Implemented software, software packages, libraries etc used in the study for computations

and visualizations should be cited in the references list to credit software developers.

**Language**

**[27]**  Page 4, line 8: Perhaps regression instead of prediction would be more accurate.

**References**

Tyralis H, Papacharalampous G, Langousis A (2019a) A brief review of random forests for water scientists and practitioners and their recent history in water resources. Water 11(5):910. https://doi.org/10.3390/w11050910.

Tyralis H, Papacharalampous G, Tantanee S (2019b) How to explain and predict the shape parameter of the generalized extreme value distribution of streamflow extremes using a big dataset. Journal of Hydrology 574:628–645. https://doi.org/10.1016/j.jhydrol.2019.04.070.

---

## Referee Comment (RC3) · Tomasz Berezowski (Referee) · 27 Nov 2019

The manuscripts aims to reconstruct the historical snow data set and to develop a real time snow depth estimation. I qualify this manuscript somewhere between major revision and rejection. The major revision is because the MS has some serious issues in methods, validation and some of the statements are not supported by the result. On the other hand the historical snow data set is an interesting product (if properly validated). The rejection is due to lack of novelty in this study: Authors use well established methods in a standard way and what they obtain is a product that has a similar RMSE as a former product available for China.

[Figure]

Major issues:

I agree with the Anonymous Reviewer (point 9b), who pointed that this complicated methodology of using RF to produce more data is probably unnecessary and that it should be tested whether this step is necessary and whether it does increase uncertainty to the product or not.

The most important issue is that the validation of the RF and the pixel based snow depths is not fair. This is because stations used for validation are only a temporal subsample of the training station set. The spatial sub-sampling was not conducted, i.e., stations for all geographic locations were used for training and validation. This is a very important problem, because latitude and longitude are the third and fourth important predictors in the model, nearly as important as the Tb. The RF model cannot know values of this predictors already during training, because the validation does not make sense. Therefore, the errors reported in this study are very optimistic (underestimated) and should be recalculated using 50

The pixel based SD product effectively fails to model snow above 20cm depth (Figure 9). This is a serious limitation and it should be explained very deeply in the discussion: (1) why this happens, (2) what is the true applicability of the product given the RMSE is 5cm, what is 25

The methods are very difficult to follow, I noticed that the other Reviewers managed to understand them better than me, but still, I am not completely sure how the study was conducted. This entire chapter should be rewritten, simplified and better structured. Often different words are used in the same context to name the same things, what makes understanding of this paper even more difficult (see attachment for some examples).

The results and discussion sections are very poorly written: methods, results and discussion are mixed in each of these sections (see attachment for some examples).

Authors should also justify better why this is a real-time approach. Is there an operational implementation of this algorithm?

Eventually, Authors claim that ML in RS is a very novel research problem, e.g. "Machine learning (ML) is a common method used in many research fields, and its **early** application in remote sensing is promising". The applications of ML in RS are not early, they are in RS since decades, either for regression (as in this study) or classification. The use of RF for regression, cannot be understand as a novelty, because it simply is not. Authors should better explain in which aspects the MS is novel.

Minor issues:

The manuscript is very poorly written with many inconsistencies. I had so many comments that I was not able to put them back from paper to the online reviewing system. Therefore I attach a scan of the paper with hand-written comments. I attach only those pages which had some comments on them.

Please also note the supplement to this comment:
https://www.the-cryosphere-discuss.net/tc-2019-161/tc-2019-161-RC3-supplement.pdf

**Supplement:**

[revised manuscript text omitted]

---

## Referee Comment (RC4) · Nir Krakauer (Referee) · 27 Nov 2019

The basic theme of this manuscript, the application of random forest (RF) to provide an empirical transfer model from remotely sensed radiances to snow depth, has merit, given that physically based transfer models are subject to limitations. However, some of the modeling choices appear questionable and should be better justified or simplified.

The RF modeling described in Section 2.3 has the following main components: (1) Using SSMI data from 1987-2004 for training and from 2005-2006 for validation, in order to evaluate the number of training samples required for good accuracy. (2) Using AMSR2 data from 2014-2015 for training and from 2012-2013 for validation. Snow

depth estimated by this model is then used to generate an approximate spatially vary-
ing relationship between 2 SSMI channel radiances and snow depth. The resulting
simple SSMI-based formula is used to reconstruct estimated snow depth for 1987-
2018, which is validated for 2017-2018.

Approach (2) appears unnecessarily complicated. If the goal is to establish a product
for 1987-2018, where only SSMI inputs are available for the entire period, it is more
logical to train an RF model directly with SSMI inputs (as done in (1) – not with AMSR2
inputs) fitted to station data (not reconstructed data). If the authors want to retain their
more complicated approach, they should compare it to the simpler one to demonstrate
that it actually has superior accuracy.

There is another way to tackle the problem of different microwave satellite sensors be-
ing available over different portions of the 1987-2018 period, which the authors may
also want to consider. This would involve combining estimates from multiple fitted
RF models, one for each satellite sensor available for part of the time period, which
would potentially more fully use the partly-independent information from multiple satel-
lite sources, which may each have different wavelength ranges, overpass times, and
other sensor characteristics.

Another issue is the training/validation station data split. As one of the other reviewers
points out, in order to better estimate the error at ungauged sites, it makes more sense
to not use some stations at all for training and retain them for validation, instead of
validating with data for the same stations but different years.

There is no comparison presented between the RF method and physically based trans-
fer models or existing satellite or reanalysis snow products over China. This work would
be stronger if the authors can conduct such a comparison and show whether RF in fact
leads to improvements in snow estimation beyond existing approaches.

Section 4.5 discusses the performance of an RF model under an ensemble of simu-
lated weather conditions and microwave radiances. It is not clear what this section adds

to the stronger results of the earlier section, which are based on real satellite and snow data. The authors should consider omitting it, and returning to these considerations in a future publication.

Also, the authors should discuss the difference between snow depth and snow water equivalent (SWE). To my understanding, SWE is more relevant for hydrologic applications, and may be more directly measured by the microwave retrievals.

On a related note, the authors note that snow measurements in high mountain areas are sparse, so that remote sensing based snow estimates cannot be validated. This could be partly overcome using a mass balance approach based on, for example, spring and summer streamflow measurements, which would give SWE (and hence, making assumptions about density, also snow depth) on a watershed scale (which in some cases might even be comparable with the satellite spatial resolution scale). See, e.g., Dahri et al. (2018) "Adjustment of measurement errors to reconcile precipitation distribution in the high-altitude Indus basin" and related work.

---

## Referee Comment (RC5) · Tomasz Berezowski (Referee) · 28 Nov 2019

Dear Authors, I found that the second point of my review did not paste fully into the online form. Here is the complete point. Sorry for the inconvenience:

The most important issue is that the validation of the RF and the pixel based snow depths is not fair. This is because stations used for validation are only a temporal subsample of the training station set. The spatial sub-sampling was not conducted, i.e., stations for all geographic locations were used for training and validation. This is a very important problem, because latitude and longitude are the third and fourth important predictors in the model, nearly as important as the Tb. The RF model cannot know

values of this predictors already during training, because the validation does not make sense. Therefore, the errors reported in this study are very optimistic (underestimated) and should be recalculated using 50% of the stations (not the data, i.e. spatial not temporal subset) which were not used to train the RF model. The same 50% subset should be used to validate the pixel-based method,

---

## Editor Comment (EC1) · Florent Dominé (Editor) · 9 Dec 2019

Dear Authors,

Your work has benefited from four useful and constructive reviews. All reviewers have issues with the choice of your machine learning methods and with your training strategy. For example, reviewer 4 wonders why you did not save some stations for validation rather than just save some years for that purpose? This is just one example, and there are many other objections. I feel that your work is overall potentially valuable and may deserve being published. However, my feeling is that you may need to demonstrate convincingly your machine learning choice of methods and perhaps rerun your training

using different schemes to test whether you have indeed made the best choices. I suspect that significant work is required to adequately address the reviewers' reasonable objections, but it is certainly worth the effort as I feel your paper is potentially of significant interest.

Also, in agreement with our data accessibility guidelines, please make sure the data you use are made accessible to other investigators on a data repository. https://www.the-cryosphere.net/about/data_policy.html

Not being myself an expert in machine learning method, I will most likely send your revised version for further review.

I look forward to reading your responses to the constructive reviews.

Best regards

Florent Domine Editor

---

## Author Comment (AC1) · 5 Jan 2020

**Response to Reviewer Comments by Divyesh Varade on "Real-Time Snow Depth Estimation and Historical Data Reconstruction Over China Based on a Random Forest Machine Learning Approach" by Jianwei Yang et al.**

Thank you for your letter and the comments concerning our manuscript. Those comments have all been very helpful for revising and improving our paper as well as providing important guidance for our research. We have studied the comments carefully and have made corrections, which we hope meet with approval. The detailed corrections and the responses to your comments are listed below point by point:

Review #1

**General Comments:** Snow depth estimates are significant for the assessment of the hydrological potential of the snowpack. The application of machine learning tools provides us with a means to derive new depth estimates from a trained model. The methods for the modeling of snow depth using remote sensing data are predominantly based on passive microwave data with much higher repeatability and spatial coverage than InSAR data, rendering such analysis suitable for the monitoring of the snow accumulation. I thus, consider this work to be significant.

Overall, the manuscript is organized and written neatly and represented in a well-structured manner. The language is mostly appropriate except for a few sentences which are not easily understandable. There are some claims and statements made by the authors that lack references or evidence. This work is appreciable in the extent of the analysis performed by the authors, in particular for the time series evolution of the snow depth in some of the major provinces in China. However, the manuscript also presents some weaknesses in the methodology, experiments, and particularly the validation.

**Specific comments:**

**1.** The authors have not clearly stated the novelty of their proposed method. In my opinion, the novelty of the proposed method is in the design of the regression model using the Random Forests i.e. the step -1 in Figure 3 and its application for the modeling of snow depth. The other steps are similar to the methodology proposed in – Jiang, L., Wang, P., Zhang, L. et al. Sci. China Earth Sci. (2014) 57: 1278. https://doi.org/10.1007/s11430-013-4798-8.

**Response 1:** Thank you for your comments, we agree on your original assessment of novelty, and this point was indeed weakly represented in the original manuscript. However, we have now redesigned the methodology in order to further increase the novelty with respect to previous studies. Specifically, there are now four RF algorithms trained with different predictive variables. Temporally and spatially independent datasets were used to validate the fitted RF algorithms. The aims were to

(1) test whether certain choices of predictive variables are necessary and whether they improve the RF algorithm;

(2) demonstrate the transferability in spatial and temporal scales.

We rewrote the part of the introduction concerning novelty, and it now reads as follows: "The primary objectives of this study are to assess the feasibility of the RF model in estimating snow depth, to determine whether the inclusion of auxiliary information (geolocation, elevation and land cover fraction) contributes to the improvement of RF, and eventually to develop a time series (1987 to 2018) of snow depth data in China and analyze the trends in annual mean snow depth. To complete the feasibility study of the RF model, we designed four RF algorithms trained with different combinations of predictor variables and validated them using temporally and spatially independent reference data. To our knowledge, this type of assessment of RF algorithm performance has not been made to date over China" (Page 3, Line 7-11, in the revised manuscript).

**2.** Why the Random Forest is used, in contrast to better alternatives such as deep neural networks? The authors claim that RF is superior to SVM and ANN, is there any documented evidence regarding RF to be superior to SVM or ANN in link with modelling of geophysical parameters similar to snow depth? Deep learning for classification and regression has been found very useful in recent literature. What is the reason that the authors use RF instead of deep neural networks? Please provide evidence for this or perform additional experiments to prove that RF-based estimates are superior to SVM, ANN, and deep NN based estimates.

**Response 2:** Thank you for your comments. In our view, any machine learning model has both advantages and disadvantages. Over the last two decades, RF has been one of the most successful machine learning algorithms for practical applications due to its proven accuracy, stability, speed of processing and ease of use (Reichstein et al., 2019). Thus, we studied whether the RF model could be used to retrieve snow depth in this study.

We also conducted a comparison between RF and ANN. The training data were from the training stations during the period 2012-2014 (Fig. 2). The predictor variables included brightness temperatures (19 GHz and 37 GHz at vertical polarization), latitude, longitude, elevation and land cover fraction. We used spatially independent data from validation stations (2015-2018) to verify the fitted ANN and RF algorithms. The results showed that the RF model was superior to ANN with respect to snow depth estimation in China (Fig. 1).

[Figure]

Figure 1. Comparison between (a) ANN and (b) RF with respect to snow depth estimation in China.

As you pointed out, there are a few pitfalls such as the risk of naive extrapolation and poor transferability in spatially limiting the applications in spatio-temporal dynamics. It is in this

realm that the techniques of deep learning promise breakthroughs. We are attempting to operate the Deep Neural Networks (DNN) model to overcome the limitations of traditional machine learning approaches.

[1] Reichstein, M., Camps-Valls, G., Stevens, B., Jung, M., Denzler, J., Carvalhais, N., Prabhat.: Deep learning and process understanding for data-driven Earth system science, Nature 566, 195–204, 2019.

We also rewrote the sentence, and now it reads as follows: "Over the last two decades, RF has been one of the most successful ML algorithms for practical applications due to its proven accuracy, stability, speed of processing and ease of use (Rodriguez-Galiano et al., 2012; Belgiu et al., 2016; Maxwell et al., 2018; Bair et al., 2018; Qu et al., 2019; Reichstein et al., 2019, Tyralis et sl., 2019a)" (Page 3, Line 2-5, in the revised manuscript).

**3.** In both cases, steps 1 and 3, the authors use only a single year data for validation. This neither provides enough points for validation nor any comprehensive inferences from the validation results.

**Response 3:** We are sorry for the confusion. The term (2012-2013) refers to two years of data, not single year. However, it does not matter because we have redesigned the methodology and added more validation data. Available stations were randomly divided into two roughly equal-sized parts by Matlab software (Fig. 2). The data from training stations (Fig. 2) during the period 2012-2014 were used to train the RF model. The dataset from validation stations during the period 2015-2018 was used to assess the accuracy of the fitted RF algorithm.

[Figure]

Figure 2. Spatial distribution of the weather stations and land cover types in the study area. There are three stable snow cover areas in China: Northeast China (NE), northern Xinjiang (XJ) and the Qinghai-Tibetan Plateau (QTP).

In this study, we used the fitted algorithm to reconstruct a long-term snow depth dataset (1987 to 2018) directly. Then, this product was evaluated by the independent ground truth

measurements over the period 1987-2018 from the validation stations (Fig. 3) and was also compared with the former snow depth data (WESTDC) in China (Fig. 4).

[Figure]

Figure 3. Scatterplots of the estimated snow depth and the ground truth observation for (a) RF and (b) WESTDC products.

[Figure]

Figure 4. Time series of (a) unbiased RMSE (unRMSE), (b) correlation coefficient (corr.coe) and (c) bias for RF and WESTDC products. The colorful dashed lines represent mean values of assessment indexes.

**4.** The datasets used for training and testing have some issues. The authors have shown how the actual depth has varied through the years 1987-2019. But for training only data till 2004 was used. The trends from Figures 10 and 11 show a marginal decrease in the mean snow depth. Would it not be better to use data from every two year or alternate year for training the RF. Similarly for testing, the authors use data from the only year 2012-13 for model testing and 2017-18 for testing the final results. This is not sufficient to develop a comprehensive interpretation of the results.

**Response 4:** Thank you for your comments. We indeed have collected ground truth snow depth observations from 1987-2018. To determine the appropriate number of training samples, a test was conducted to analyze the sensitivity of the RF model to training sample size. To ensure there were enough samples, we selected 80,000 samples from 1987 to 2004 as available training data, and a two-year dataset from 2005 to 2006 was applied to assess the performance.

We agree with your opinion regarding the validation using much more data, and these comments are very constructive. Thus, we have added more data to validate the fitted RF algorithms and the reconstructed snow depth product. Please refer to the response to "Specific comment 4" above.

**5.** In section 3.2, the correlation coefficient is 0.77. Is this satisfactory enough to be used to generate the reference dataset from the RF model? A majority of data are below 10 cm snow depth, then an error of 4.5 cm is significantly high. To have a better understanding of the modeled results, it is vital that we observe the accuracy for the points of higher snow depth also. Particularly, when there is a very high snow depth different for the regions QTP and the others. The validation should be carried out for these regions separately. I suggest the authors show a histogram of the data and also carry out a separate fit for points of snow depth >10cm or perform a case by case fit with respect to the study area. A significant concern is that in the case of shallow snow (<10cm), is the brightness temperature actually representative of the contributions from the shallow snowpack or the underlying ground. This requires further investigations. This is important since the bulk of the data is within the 0-10 cm range. Another concern is that there are very few points with snow depth >40cm. In several locations in the Himalayas, the peak snow depth is usually around 1m or more. Thus, the applicability of the proposed method or the transferability of the proposed method to other areas, in these cases, is in question.

**Response 5:** Thank you for your comments. Other reviewers gave similar comments. Since the dataset obtained by RF is an approximation of the true snow depth, the uncertainties are transferred to the second stage of prediction. Other reviewers suggested that we directly use the fitted RF algorithm to produce the long-term snow depth data in the period 1987-2018.

Figure 5 shows the histograms of observations from training and validation stations during the period 2012-2018. Ninety percent of the samples range from 1 cm to 25 cm. The maximum values of the snow depth extend to approximately 50 cm. However, the number of such cases is small and is therefore not evident in Fig. 5.

[Figure]

Figure 5. Histograms of snow depth observations from (a) training and (b) validation stations. The average values (black dashed lines) are equal to 10.5 cm and 9.8 cm, respectively.

The idea to carry out a separate fit for points of snow depth > 10 cm is good, but it cannot be used to estimate snow depth in space and time. This is because passive microwave observations cannot distinguish deep and shallow snow cover so that the background of snow depth is unknown. Thus, for a snow cover satellite pixel, we don't know which fitted RF algorithm should be used to retrieve snow depth.

We agree with your comments about underestimations for deep snow. The validation was carried out for three snow cover regions in China separately (Fig. 6).

[revised manuscript text omitted]

**6.** The authors observed higher errors for shallow snow depth, but the manuscript lacks any discussion on the contributions from the underlying ground layer to the passive microwave brightness temperature in case of shallow snow depth. The authors have simply added some references. A discussion is required in the manuscript on the sensitivity of snowpack thickness and stratigraphy towards the passive microwave brightness temperature.

**Response 6:** We redesigned the methodology in this study. The new RF product presented lower errors under shallow snow cover conditions (Table 1). We have discussed this finding in Section 4.3. Please refer to the response to "Specific comment 5" above.

The microwave emission model of layered snowpack (MEMLS) was applied to simulate the $T_B$ with varying snow parameters (Mätzler et al., 1999; Löwe et al., 2015; Pan et al., 2015). Fig. 7 shows the sensitivity of snow depth to $T_B$ at 36 GHz for various snow density and snow grain size. Generally, the snow density (< 100 kg/m$^3$) and snow grain size (correlation length < 0.2 mm) are small for shallow snow cover (< 5 cm). The passive microwave signals are insensitive to the shallow snow cover. Moreover, the snow cover is patchy under shallow snow conditions, challenging the relationship between satellite $T_B$ and snow depth.

[Figure]

Figure 7. The sensitivity of snowpack stratigraphy to the passive microwave brightness temperature simulated with the MEMLS model.

**7.** Page 12, L25-27: Does this mean 3-10 samples in (3x25)x(3x25) sq. km area? This is not clear to me. I think the authors are referring to measurements from field campaigns or

weather stations as samples. In this case, the number of samples is very small per the averaging window. Please provide references for this.

**Response 7:** Thank you for your comments. We apologize that the description of this part was not clear. We have redesigned the paper and removed the pixel-based method according to other reviewers' comments.

**8.** Figures 9a and 9b. There are very few samples used for validation in these figures. Further, these samples are discontinuous (Figure 9a) and therefore, this should not be used as the basis for ascertaining the performance of the proposed method, since due to the distribution of the points, it is expected that the fit will provide better results.

The authors may perform other significance tests such as Nemar's test, but the fact remains that the validation data is not really comprehensive. The data shown in Figure 9b is much better for assessment, as it is continuous. But why only 10 points? Earlier it was shown that several ground stations exist in the area. I suggest the authors also use data from other years in their validation scheme, as the results shown at present are not convincing. Why is the modeled snow depth showing very less sensitivity between 20-40cm (nearly constant) and again afterward? This is an issue that requires investigation.

**Response 8:** Thank you for your constructive comments. We used independent ground truth observations from 1987 to 2018 to validate the RF product. Fig. 3 shows the error bars and scatterplots. The "o" marker is the mean snow depth computed at each corresponding ground truth bin, while upper and lower colorful bars indicate one standard deviation from the mean. There are almost 280,000 samples. Please refer to the response to "Specific comment 3" above.

**9.** In section 4.5, the selection of sample size for training and testing is reversed. Since the MEMLS requires auxiliary information, which is seldom available, the training samples should be much less than the validation samples. This validation strategy is not convincing. From the discrepancy in the training and testing samples, it is already expected that the model accuracy would be high.

**Response 9:** We appreciate your suggestions. The aim of this part work is to demonstrate that more prior snow information can improve the performance of the RF model. Reviewer #4 suggested we should omit this part and return to the combination in a future publication. Thus, combining the snow forward model with the ML method will be the focus of our future work.

**Minor issues:**

**1.** Page 02, L7: " the Himalayas during: : :.". The Himalayan ranges are very long and are shared by several countries. Please specify which Himalayan ranges the authors are referring to here. I do not agree with the statement that mean snow depth is maximum in Xinjiang for the entire Himalayan range. Please provide references for this.

**Response 1:** We apologize for the confusion. Three snow cover areas are shown in Fig.1 (Please refer to the response to "Specific comment 3" above). The trend analysis of snow depth was conducted based on the ground truth observations, RF dataset and WESTDC

product during the period 1987-2018. To illustrate the different changing patterns, the trends in northern XJ, NE and QTP were analyzed.

[Figure]

Figure 8. The trend analysis of snow depth based on (a) station observations, (b) RF estimates, and (d) WESTDC product in three stable snow cover areas in China. The correlation is statistically significant at the 0.05 level.

We rewrote the sentence as follows: "On a temporal scale, the ground truth snow depth presented a significant increasing trend from 1987 to 2018, especially in NE. However, the RF and WESTDC products displayed no significant changing trends except in QTP. The WESTDC product presented a significant decreasing trend in QTP, with a correlation coefficient of -0.55, whereas there were no significant trends for ground truth observations and the RF product" (Page 1, Line 26-29 in the revised manuscript).

**2.** Page 02: L8-11: These are documented facts in literature for several other locations, however. Thus, the authors should strictly restrict their inferences to their own findings and not speculate. Thus, here the sentence should be specific to the study area in the manuscript.

**Response 2:** We appreciate your suggestions. Three snow cover areas in China are shown Fig. 1. The time series of mean snow depth in three stable snow cover areas over China is shown in Fig. 8. Fig. 8a shows that the mean snow depth in northern XJ is the largest among the three regions, and the pattern in NE is highly consistent with the overall trend in China. Comparing the ground truth data and RF product (Fig. 8a *vs.* 8b) shows that there are similar patterns in terms of the magnitude of snow depth in the three snow cover areas.

**3.** Page 02, L11-13: The sentence "In conclusion: : :." is not clear. Please rephrase.
**Response 3:** We consider this sentence to be unnecessary and have removed it.

**4.** Page 02, L24: "mean snow density". I believe the authors are here referring to mean stratigraphic snow density". Please correct this.
**Response 4:** Thank you for your comments. Reviewer #4 thought this paper should focus on snow depth and not snow water equivalent. Thus, we removed this description and rewrote the sentence as follows: "Snow depth is a crucial parameter for climate studies, hydrological applications and weather forecasts (Foster et al., 2011; Takala et al., 2017; Tedesco et al., 2016; Safavi et al., 2017)" (Page 2, Line 4-6, in the revised manuscript).

**5.** Page 03, L17-18: "however, these: : :". Is there any evidence that the RTM based

methods are computationally more expensive than machine learning-based methods. In my opinion, both depend on the selection of the parameters. For example, an RF with substantial input and a high number of trees may be as expensive computationally. If there is no documented evidence on this, please remove this statement.
**Response 5:** We deleted the sentence in the revised manuscript.

**6.** Page 11, L 11-13: Please correct the range as 200-350 kg/m3 and provide a reference, for example- Meløysund, Vivian, Bernt Leira, Karl V. Høiseth, and Kim R. Lisø. 2007. "Predicting snow density using meteorological data." Meteorological Applications14 (4): 413–23. doi:10.1002/met.40.
**Response 6:** We appreciate the reviewer's help and suggestions. We read the reference carefully. It is a good paper and very useful for us. We corrected the range and cited the reference in the revised manuscript (Page 11, Page 5-7).

**7.** Page 17, L20: "The snowpack is set ..". This should be the snowpack is assumed to comprise a single layer indicating a semi-infinite medium. This is a common assumption in electromagnetic modeling of the snowpack. Please add references to this.
**Response 7:** We removed this part. Please refer to the response to "Specific comment 9" above.

8. Figure 1: This needs to be revised. Firstly, the authors use 3 areas for their study which have not been shown on the large map. Secondly, the two pixels mentioned previously should be shown at a higher resolution. Third, write in captions what the color bar represents, is it elevation? Finally, the pixels shown should also have a lat-long grid and scale bar.
**Response 8:** We appreciate the reviewer's comments and suggestions. We redesigned the map (Fig. 1). Because of the paucity of samples from the field sampling campaign, we omitted these data and added more station observations (1987 to 2018) as a new validation dataset.

**9.** Figure 7: Why is the number of points and their locations changing in the maps showing stations. I believe this should remain fixed irrespective of the month. If there is no snow at some of the stations which have been omitted, these should be shown with either a different symbol or a color.
**Response 9:** As you pointed out, the number of available station observations is not fixed during the snow winter season. In the revised manuscript, we have deleted this statement.

**10.** Figure 8: The images are distorted. It appears as if they were stretched manually to fit some size.
**Response 10:** Thank you for your comments. The pixel-based algorithm was omitted in the revised manuscript. Please refer to the response to "Specific comment 5" above.

**11.** Figure 9/Table 4 and several other instances: The R2 and R, i.e. the determination coefficient and the correlation coefficient, respectively, are two different parameters

and have been used interchangeably with similar symbols in the manuscript, which makes it difficult to judge the accuracy of the results.

**Response 11:** We apologize that we did not describe this consistently. We corrected it in the revised manuscript.

---

## Author Comment (AC2) · 5 Jan 2020

**Response to Reviewer Comments by Review #2 on "Real-Time Snow Depth Estimation and Historical Data Reconstruction Over China Based on a Random Forest Machine Learning Approach" by Jianwei Yang et al.**

Thank you for your letter and the comments concerning our manuscript. Those comments have been very helpful for revising and improving our paper as well as providing guidance for our research. We have studied the comments carefully and have made corrections, which we hope meet with approval. We provide responses in blue below.

Review #2

Aim of the manuscript

[1] The aim of the manuscript is (a) to test random forests in estimating snow depth in a remote sensing application and (b) to reconstruct historical snow depth in China in the period 1987–2018 (see page 5, lines 10–14).

[2] The procedure of the manuscript is presented in Figure 3.

Recommendation: Major revisions are needed

**General evaluation**

**1.** The procedure followed in the manuscript is complicated, while I think that some steps are unnecessary and a more straightforward approach to the problem would achieve comparable (or even better results).

**Response 1:** Other reviewers (Reviewers #3 and #4) gave similar comments. Thus, we redesigned the methodology in this study to improve this manuscript. The results demonstrate that certain predictor variables are unnecessary. There are four major revisions in the new manuscript.

1) Revision 1: scientific validation dataset

One of the major issues of the original manuscript was that the validation data are not independent temporally and spatially. Thus, in the revised manuscript, available stations were randomly divided into two roughly equal-sized parts by Matlab software (Fig. 1). The snow depth observations from training stations (342 sites) together with satellite $T_B$ and other auxiliary data can be used to train the RF model. The measurements from validation stations (341 sites), as spatially independent data, can be applied to validate the fitted RF algorithm and the reconstructed snow depth product. Fig. 2 shows the histograms of snow depth observations from training and validation stations during the period 2012-2018. Ninety percent of the samples range from 1 cm to 25 cm. The maximum values of the snow depth extend to approximately 50 cm. However, the number of such cases is small and is therefore not evident in Fig. 2.

[Figure]

Figure 1. Spatial distribution of the weather stations and land cover types in the study area. There are three stable snow cover areas in China: Northeast China (NE), northern Xinjiang (XJ) and the Qinghai-Tibetan Plateau (QTP).

[Figure]

Figure 2. Histograms of snow depth observations from (a) training and (b) validation stations. The average values (black dashed lines) are equal to 10.5 cm and 9.8 cm, respectively.

2) Revision 2: four selection rules of predictor variables

The procedure described in the original manuscript is complicated. Based on the correlations between the predictor variables and the variable importance metrics (Fig. 3), we designed four schemes of predictor variables to train the RF model in the revised manuscript. The scheme one was the simplest and its predictor variables included satellite observations at 19 GHz and 37 GHz only (Table 1). The scheme four was the most complicated. We first demonstrated whether certain predictor variables are necessary and whether their inclusion affects the RF model.

[Figure]

Figure 3. Correlations between the predictor variables (left) and the ranking of variable importance (right). The importance of variables, referred to as Mean Decrease Accuracy (MDA) in the RF model, is obtained by averaging the difference in out-of-bag error estimation before and after the permutation over all trees. The larger the MDA, the greater the importance of the variable is.

Table 1. A detailed description of the input predictor variables based on four selection rules of the training sample.

| Name | Predictor Variables | Target | Note |
|------|---------------------|--------|------|
| RF1 | $T_{B19V}$, $T_{B37V}$ | | land cover types: |
| RF2 | $T_{B19V}$, $T_{B37V}$, Latitude, Longitude | snow depth | grassland, cropland, |
| RF3 | $T_{B19V}$, $T_{B37V}$, Latitude, Longitude, Elevation | | bareland, |
| RF4 | $T_{B19V}$, $T_{B37V}$, Latitude, Longitude, Elevation, Land cover fraction | | shurbland, forest |

3) Revision 3: validation of the fitted RF algorithms

We conducted three tests to verify the fitted RF algorithms (Table 2). The same training samples (same algorithms) were used for the three tests but with different validation datasets. In Test1, the validation data were from out-of-bag (OOB) samples. Generally, approximately two-thirds of the samples (in-bag samples) were used to train the trees and the remaining one-third (OOB samples) were used to estimate how well the fitted RF algorithm performed. This preliminary assessment generally provides a simple way to adjust the parameters of the RF model. However, we should use the OOB errors with caution because its samples are not independent at temporal and spatial scales. In Test2, we applied temporally independent reference data during the period 2015-2018 to assess the accuracy of the temporal prediction of fitted algorithms. In Test3, a spatially independent dataset from validation stations during the period 2015-2018 was used to assess the accuracy of spatio-temporal prediction.

Fig. 4 indicates that the accuracy of RF model is greatly influenced by geographic location, elevation, and land cover fractions. However, the redundant predictor variables (if highly correlated) slightly affect the RF model. The fitted RF algorithms perform better at the

temporal scale than that at the spatial scale, with unbiased RMSEs of ~4.4 cm and ~7.3 cm, respectively.

Table 2. Summary of three tests of the fitted RF algorithms in Table 1.

| Name | Test1 (OOB) | | Test2 (temporal subset) | | Test3 (spatio-temporal subset) | |
|---|---|---|---|---|---|---|
| training | training stations | 2012-2014 | training stations | 2012-2014 | training stations | 2012-2014 |
| | samples | 28602 | samples | 28602 | samples | 28602 |
| validation | training stations | 2012-2014 | training stations | 2015-2018 | validation stations | 2015-2018 |
| | samples | 14301 | samples | 34684 | samples | 25879 |

[Figure]

Figure 4. The color-density scatterplots of the estimated snow depth with four fitted RF algorithms and the ground truth snow depth. The four trained RF algorithms (RF1, RF2, RF3, RF4) were evaluated with three validation datasets (Test1, Test2, Test3).

4) Revision 4: validation of the reconstructed snow depth product

Finally, we directly used the fitted RF2 algorithm to retrieve a consistent 32-year daily snow depth dataset from 1987 to 2018. This product was evaluated against the independent station observations during the period 1987-2018. The mean unbiased RMSE and bias were 7.1 cm and -0.05 cm, respectively, outperforming the former snow depth dataset (8.4 cm and -1.20 cm) from the Environmental and Ecological Science Data Center for West China (WESTDC).

[Figure]

Figure 5. Scatterplots of the estimated snow depth and the ground truth observation for (a) RF and (b) WESTDC products.

To determine the interannual variability in the uncertainty, the time series of assessment indexes, including the unbiased RMSE, bias and correlation coefficient, are shown in Fig. 6. The results show that the RF estimates outperform the WESTDC product with respect to unbiased RMSE and correlation coefficient from season to season.

[Figure]

Figure 6. Time series of (a) unbiased RMSE (unRMSE), (b) correlation coefficient (corr.coe) and (c) bias for RF and WESTDC products. The colorful dashed lines represent mean values of assessment indexes.

**2.** Regarding the algorithmic part of the manuscript, I have some recommendations to justify certain choices of the manuscript and highlight some advantages and drawbacks of random forests (regarding most minor comments on the algorithmic part, e.g. parameters of random forests, variable importance, number of predictor variables and more, as well as why one should use random forests instead of another algorithm, please consider reading the random forests review by Tyralis et al. 2019a for more details)..

**Response 2:** We appreciate the reviewer's help and suggestions. We conducted a test to justify whether certain steps are necessary. Please refer to the response to "General evaluation 1" above.

We read the reference carefully. It is a good paper and was very useful for us. We have rewritten the introduction to the RF model in Section 2.2.1.

"RF is an ensemble ML algorithm proposed by Breiman in 2001. It combines several randomized decision trees and aggregates their predictions by averaging in regression (Biau and Scornet, 2016). Generally, approximately two-thirds of the samples (in-bag samples) are used to train the trees and the remaining one-third (out-of-bag samples, OOB) are used to estimate how well the fitted RF algorithm performs. Few user-defined parameters are generally required to optimize the algorithm, such as the number of trees in the ensemble (*n*tree) and the number of random variables at each node (*m*try). The *n*tree is set equal to 1000 in the present study since the gain in the predictive performance of the

algorithm would be small with the addition of more trees (Probst and Boulesteix, 2018). The default value of *m*try is determined by the number of input prediction variables, usually 1/3 for regression tasks (Biau and Scornet, 2016). The RF regression is insensitive to the quality of training samples and to overfitting due to the large number of decision trees produced by randomly selecting a subset of training samples and a subset of variables for splitting at each tree node (Maxwell et al., 2018). In addition, RF provides an assessment of the relative importance of predictor variables, which have proven to be useful for evaluating the relative contribution of input variables (Tyralis et al., 2019b). Furthermore, the RF model can rapidly trained and is easy to use. In this paper, a randomForest R package (Version 4.6-14) is used for regression (Liaw and Wiener 2002; Breiman et al. 2018)" (Page 4, Line 20-30 in the revised manuscript).

We also highlignted the drawbacks of RF model in Senction 4.1.
"The RF technique is already used to generate temporal and spatial predictions. Generally, the RF model cannot extrapolate outside the training range (Hengl et al., 2018). Fig. 6 and Table 4 indicate that the spatial predictions of fitted RF algorithms are more biased than are the temporal predictions. Thus, the transferability of a fitted RF algorithm to other areas is in question. Several studies (Prasad, Iverson & Liaw, 2006; Hengl et al., 2017; Vaysse & Lagacherie, 2015; Nussbaum et al., 2018) have proven that RF is a promising technique for spatial prediction; however, these studies aim at spatial prediction of properties that are relatively static over the observational period, e.g., soil types and soil properties.
What makes the Earth system interesting is that it is not static but dynamic (especially concerning snow parameters). Generally, snow depth increases at the beginning of winter and then decreases in spring due to melting. Moreover, snow cover has different spatial patterns in various regions, such as generally deep snow in high-latitude and high-elevation areas. In China, there are five climatological snow classes following the classification by Sturm et al. (1995). Each snow class is defined by an ensemble of snow stratigraphic characteristics, including snow density, grain size, and crystal morphology, which influences the snowpack's microwave signature (Sturm et al., 2010). These dynamic properties of snow will lead to many cases in which the same satellite $T_B$ corresponds to different snow depths, while the same snow depth is associated with various $T_B$ observations, rendering the fitted RF algorithm suboptimal. Using ML techniques in combination with snow forward models (physical modeling) has the potential to overcome many limitations that have hindered a more widespread adoption of ML approaches" (Page 9, Line 20-30 in the revised manuscript).

**3.** Furthermore, I think that the manuscript is wordy at some Sections, for instance explanation of Figures.

**Response 3:** We agree with the reviewer's opinion. We revised all the sections thoroughly to make it more precise.

**4.** Perhaps the reconstructed dataset could be made available online increasing the value of the manuscript.

**Response 4:** We agree with the reviewer's opinion. The reconstructed dataset from 1987 to 2018 is now available and we will upload the data later.

**Major comments:**

**1.** Page 8, line 10 – page 9, line 25: In general, I think that the procedure described here is complicated, while some steps may be unnecessary. In particular:

a. Random forests are fitted using 15 predictor variables in the period 2014–2015 (page 8, lines 11, 12) and then they are validated in the period 2012–2013. I do not understand the scope of this validation, considering that parameters of the algorithm have been defined earlier.

**Response 1:** Thank you for your comments. We have revised the manuscript. Please refer to the response to "General evaluation 1" above.

**2.** Random forests are used to predict snow depth in the period 2012–2018. Then a linear model is trained in the predictions of the period 2012-2018 using two predictor variables. The trained linear model is used to predict snow depth in the period 1987-2018.
In my opinion it would be more straightforward to train random forests in the period 2014-2015 using two predictor variables and then predict in the period 1987-2018. Another straightforward option would be to train a linear model in the period 2014-2015 and then predict in the period 1987-2018.

**Response 2:** Thank you for your constructive comments. In the revised manuscript, we directly used the fitted RF algorithm to retrieve a consistent 32-year daily snow depth dataset from 1987 to 2018. Please refer to the response to "General evaluation 1" above.

**3.** Instead, following the two-stage procedure of the manuscript, a dataset, obtained by some predictions, is used to train a new model. In these procedures uncertainties are introduced (since the dataset obtained by random forests is an approximation of the true snow depth) which are transferred to the second stage prediction. I understand that this approach gives a rich dataset to do the second stage training, however I think that the induced uncertainties are not compensated by the bigger dataset. Perhaps the manuscript could justify this approach by performing some comparisons between the one and the two-stage approaches in the period 2012-2018 or just completely use the straightforward approach.

**Response 3:** Other reviewers gave similar useful and constructive comments. Thus, we directly used the fitted RF algorithm to retrieve a consistent 32-year daily snow depth dataset from 1987 to 2018 in the revised manuscript and omitted the pixel-based algorithm.

**4.** Perhaps the approximation of equation (2) is suboptimal because it is based on data before 2008, while it does not include the intercept parameter. Given the big magnitude of the dataset, it is surprising that a one-parameter linear model (equation 2) would be preferable to the two-parameter model of equation (1).

**Response 4:** According to reviewers' suggestions, we directly used the trained RF model to retrieve long-term snow depth product, leaving out the pixel-based algorithm. Please refer to the response to "General evaluation 1" above.

**Minor comments**

**1.** Page 2, lines 15 – 20: A proper assumption for applying random forests is stationarity. Furthermore, random forests do not predict outside the range of the training sample. Therefore, the assumption of global warming is not compatible with random forests.
**Response 1:** We agree with the reviewer's opinion. We deleted this sentence.

**2.** Page 6, line 1: SSMIS provides data in the period 2006-present according to Table 1.
**Response 2:** Yes, SSMIS provides data from 2006 to the present and SSM/I from 1987 to 2008 (Table 3). We changed the sentence to the following: "The series of the Special Sensor Microwave/Imager (SSM/I) and Special Sensor Microwave Imager Sounder (SSMIS) instruments has provided continuous $T_B$ measurements at 19.35, 23.235, 37, 85.5 and 91.655 GHz since July 1987" (Page 3, Line 18-20, in the revised manuscript).
Table 3. Summary of the main passive microwave remote sensing sensors.

| Sensor | SSM/I | | | SSMIS |
|---|---|---|---|---|
| Satellite | DMSP-F08 | DMSP-F11 | DMSP-F13 | DMSP-F17 |
| On Orbit time | 1987-1991 | 1991-1995 | 1995-2008 | 2006-present |
| Passing Time | A: 06:20
D: 18:20 | A: 17:17
D: 05:17 | A: 17:58
D: 05:58 | A: 17:31
D: 05:31 |
| Frequency & footprint (GHz) : (km × km) | | 19.35: 45×68
23.235: 40×60
37: 24×36
85.5: 11×16 | | 19.35: 42×70
23.235: 42×70
37: 28×44
91.655: 13×15 |

**3.** Page 7, lines 16 – 17: Random forests parameters are more than two.
**Response 3:** Thank you for your comments. We changed the sentence to the following: "Few user-defined parameters are generally required to optimize the algorithm, such as the number of trees in the ensemble (*n*tree) and the number of random variables at each node (*m*try)" (Page 4, Line 23-24, in the revised manuscript).

**4.** Page 7, lines 21 – 27: In general the default values (in the software implementation) of random forests parameters are good.
**Response 4:** We agree with the reviewer's opinion. In this study, we used the default values of parameters.

**5.** Page 7, lines 21 – 27: In general it is suggested to use as high number of trees as computationally feasible. However, indeed the number of 500 trees is high enough in most applications.
**Response 5:** We agree with the reviewer's opinion. Please refer to the response to "Minor Comment 4" above.

**6.** Page 7, line 27 – page 8, line 2: In general the larger the dataset, the better the predictive ability of a regression algorithm.

**Response 6:** We agree with the reviewer's opinion. Fig. 7 suggests that the accuracy of the SVM estimation is related to the training data size (Xiao et al., 2018).

[Figure]

Figure 7. Trend of R (correlation coefficient), MAE (mean absolute error) and RMSE (root mean squared error) with increasing training sample size. K represents one thousand (from Xiao et al., 2018).

In our study, we also analyzed the performances of the RF model with increasing training sample size. The results revealed that the accuracy of RF estimation is insensitive to the training data size (Fig. 8). One of the advantages of the RF model is that it can effectively handle small sample sizes (Biau and Scornet et al., 2016).

[Figure]

Figure 8. Trends of (a) unbiased RMSE, (b) bias and (c) correlation coefficient with increasing training sample size.

**7.** Page 10, lines 8–12: By increasing the size of training sample one would expect that the performance of predictive algorithm would increase.

**Response 7:** Thank you for your comments. Please refer to the response to "Minor Comment 6" above.

**8.** Page 11, lines 4, 5: Which linear model?

**Response 8:** Thank you for your comments. We have changed the sentence to the following: "The reconstructed product was also compared with the static linear-fitting algorithm developed by fitting 19 and 37 GHz with the snow depth measurements with a constant empirical coefficient over China (Che et al., 2008). The daily snow depth data were obtained from the Environmental and Ecological Science Data Center for West China (http://westdc.westgis.ac.cn) (hereafter, WESTDC product)" (Page 6, Line 17-20, in the revised manuscript).

**9.** Page 11, lines 22–24: The comparison between random forests and the linear model is unfair considering that the latter uses less predictor variables.

**Response 9:** Thank you for your comments. We studied whether the machine learning method can overcome the limitations of empirical algorithms. Yang et al. (2019) validated five empirical algorithms and found that this linear model outperformed four other snow depth estimation methods in China. Thus, in this study, we directly compared the estimates of the RF and linear models. We removed this comparison and conducted a more comprehensive analysis of the reconstructed snow depth product.

[1] Yang, J., Jiang, L., Wu, S., Wang, G., Wang, J., and Liu, X.: Development of a Snow Depth Estimation Algorithm over China for the FY-3D/MWRI, Remote Sensing, 11, 977, 10.3390/rs11080977, 2019.

**10.** Page 12, lines 25–27: This procedure is not clear.

**Response 10:** We apologize that the description of this procedure was not specific and clear. We omitted this procedure in the revised manuscript according to the reviewers' suggestions. Please refer to the response to "General evaluation 1" above.

**11.** Page 13, lines 3, 4: I do not understand why assigning values to the slope and intercept.

**Response 11:** We apologize that the description was not clear. If there are fewer than three available measurements in a pixel during the winter seasons for the 2012-2018 period, the regression coefficients (slope and intercept) can not be calculated. But the snow cover detection method maybe classify this pixel into snow. In such case, we have to assign values to the slope (0.66) and intercept (0) according to the linear model.

We omitted this procedure in the revised manuscript according to the reviewers' suggestions. Please refer to the response to "General evaluation 1" above.

**12.** Page 16, lines 8–11: It is not clear which period was used to compute variable importance.

**Response 12:** Thank you for your comment. We added the period in the revised manuscript (Page 4, Line 8-9).

**13.** Page 16, lines 24–28: Perhaps the information added by the longitude and latitude predictor variables is already included in the remaining predictor variables (see e.g. a similar application in Tyralis et al. 2019b). In the latter study, the predictive performance was examined by comparing models with and without longitude and latitude, and the effect of coordinates was found insignificant. Perhaps, computing variable importance and predicting performance would give some explanations on the value of the remaining predictor variables and make the model less dependent on the proximity of nearby stations.

**Response 13:** We agree with your opinion. Fig. 3 shows that the latitude is highly correlated with the brightness temperature. Thus, latitude has a very slight influence on the predictive performance. However, longitude is poorly correlated with the brightness temperature. Moreover, Fig. 3 indicates that the longitude is more important than latitude to snow depth. We read the reference carefully and cited it as follows: "In addition, RF provides an assessment of the relative importance of predictor variables, which have

proven to be useful for evaluating the relative contribution of input variables (Tyralis et al., 2019b)" (Page 4, Line 29-30, in the revised manuscript).

**14.** Page 18, lines 1–3: In general one would expect that using more predictor variables related to the dependent variable of interest would improve the trained model. Furthermore, redundant predictor variables slightly affect random forests.

**Response 14:** We agree with the reviewer's opinion. Our results also demonstrate that redundant predictor variables slightly affect random forests.

**15.** Figure 6: Figures should be numbered and respective explanations should be added in the caption.

**Response 15:** We corrected it.

**16.** Regarding the implementation of random forests, some of their disadvantages and their impact in the results of the study can be discussed (see a list of disadvantages in Tyralis et al. 2019a), e.g. they do not extrapolate outside the training range, variable importance metrics are not always reliable, as they are affected by high correlations and interactions, and more.

**Response 16:** These comments are very useful for improving our paper. We read the reference paper carefully and disscussd the limitaions of the RF model in Section 4.1.

"The RF technique is already used to generate temporal and spatial predictions. Generally, the RF model cannot extrapolate outside the training range (Hengl et al., 2018). Fig. 6 and Table 4 indicate that the spatial predictions of fitted RF algorithms are more biased than are the temporal predictions. Thus, the transferability of a fitted RF algorithm to other areas is in question. Several studies (Prasad, Iverson & Liaw, 2006; Hengl et al., 2017; Vaysse & Lagacherie, 2015; Nussbaum et al., 2018) have proven that RF is a promising technique for spatial prediction; however, these studies aim at spatial prediction of properties that are relatively static over the observational period, e.g., soil types and soil properties.

What makes the Earth system interesting is that it is not static but dynamic (especially concerning snow parameters). Generally, snow depth increases at the beginning of winter and then decreases in spring due to melting. Moreover, snow cover has different spatial patterns in various regions, such as generally deep snow in high-latitude and high-elevation areas. In China, there are five climatological snow classes according to Sturm et al. (1995). Each snow class is defined by an ensemble of snow stratigraphic characteristics, including snow density, grain size, and crystal morphology, which influences the snowpack's microwave signature (Sturm et al., 2010). These dynamic properties of snow will lead to many cases in which the same satellite $T_B$ corresponds to different snow depths, while the same snow depth is associated with various $T_B$ observations, rendering the fitted RF algorithm suboptimal. Using ML techniques in combination with snow forward models (physical modeling) has the potential to overcome many limitations that have hindered a more widespread adoption of ML approaches" (Page 9, Line 22-30, in the revised manuscript).

**17.** Implemented software, software packages, libraries etc used in the study for computations and visualizations should be cited in the references list to credit software developers.

**Response 17:** Thank you for your suggestion. We added the information on the RF model (https://cran.r-project.org/web/packages/randomForest): "In this paper, a randomForest R package (Version 4.6-14) is used for regression (Liaw and Wiener 2002; Breiman et al. 2018)" (Page 5, Line 1-2, in the revised manuscript).

**Language**

**1.** Page 4, line 8: Perhaps regression instead of prediction would be more accurate.

**Response 1:** We agree with your opinion. We changed "prediction" to "regression" in the revised manuscript.

---

## Author Comment (AC3) · 5 Jan 2020

**Response to Reviewer Comments by Tomasz Berezowski on "Real-Time Snow Depth Estimation and Historical Data Reconstruction Over China Based on a Random Forest Machine Learning Approach" by Jianwei Yang et al**

Thank you for your letter and the comments concerning our manuscript. Those comments have been very helpful for revising and improving our paper as well as providing guidance for our research. We have studied the comments carefully and have made corrections, which we hope meet with approval. We provide responses in blue below.

Review #3

**General Comments:** The manuscripts aims to reconstruct the historical snow data set and to develop a real time snow depth estimation. I qualify this manuscript somewhere between major revision and rejection. The major revision is because the MS has some serious issues in methods, validation and some of the statements are not supported by the result. On the other hand the historical snow data set is an interesting product (if properly validated). The rejection is due to lack of novelty in this study: Authors use well established methods in a standard way and what they obtain is a product that has a similar RMSE as a former product available for China.

**Response:** Thank you for your comments. We revised the manuscript carefully and thoroughly. According to yours and other reviewers' suggestions, we redesigned the methodology and conducted the comparisons between the complicated and simple methods to demonstrate which procedure is more effective for snow depth estimation, also improving novelty of the study. The primary objectives of this study are to assess the feasibility of the RF model in estimating snow depth, to determine whether the inclusion of auxiliary information (geolocation, elevation and land cover fraction) contributes to the improvement of RF, and eventually to develop a time series (1987 to 2018) of snow depth data in China and analyze the trends in annual mean snow depth. To complete the feasibility study of the RF model, we designed four RF algorithms trained with different combinations of predictor variables and validated them using temporally and spatially independent reference data. To our knowledge, this type of assessment of RF algorithm performance has not been made to date over China. The reconstructed snow depth dataset is now available and we will upload it later. There are four major revisions in this study.

**1) Revision 1: scientific validation dataset**
One of major issues in the original manuscript was the validation data are not temporally and spatially independent. Thus, in the revised manuscript, available stations in China were randomly divided into two roughly equal-sized parts by Matlab software (Fig. 1). The snow depth observations from training stations (342 sites) together with satellite $T_B$ and other auxiliary data can be used to train the RF model. The measurements from validation stations (341 sites), as spatially independent data, can be applied to validate the fitted RF algorithm and the reconstructed snow depth product. Fig. 2 shows the histograms of snow depth observations from training and validation stations during the period 2012-2018. Ninety percent of the samples range from 1 cm to 25 cm. The maximum values of the snow

depth extend to approximately 50 cm. However, the number of such cases is small and is therefore not evident in Fig. 2.

[Figure]

Figure 1. Spatial distribution of the weather stations and land cover types in the study area. There are three stable snow cover areas in China: Northeast China (NE), northern Xinjiang (XJ) and the Qinghai-Tibetan Plateau (QTP).

[Figure]

Figure 2. Histograms of snow depth observations from (a) training and (b) validation stations. The average values (black dashed lines) are equal to 10.5 cm and 9.8 cm, respectively.

**2) Revision 2: four selection rules of predictor variables**

The procedure described in the original manuscript was complicated. Based on the correlations between the predictor variables and the variable importance metrics (Fig. 3), we designed four schemes of predictor variables to train the RF model in the revised manuscript. The scheme one was the simplest and its predictor variables included satellite observations at 19 GHz and 37 GHz only (Table 1). The scheme four was the most complicated. We first demonstrated whether certain predictor variables are necessary and whether their inclusion affects the RF model.

[Figure]

Figure 3. Correlations between the predictor variables (left) and the ranking of variable importance (right). The importance of variables, referred to as Mean Decrease Accuracy (MDA) in RF model, is obtained by averaging the difference in out-of-bag error estimation before and after the permutation over all trees. The larger the MDA, the greater the importance of the variable is.

Table 1. A detailed description of the input predictor variables based on four selection rules of training sample.

| Name | Predictor Variables | Target | Note |
|------|---------------------|--------|------|
| RF1 | $T_{B19V}$, $T_{B37V}$ | | land cover types: |
| RF2 | $T_{B19V}$, $T_{B37V}$, Latitude, Longitude | snow depth | grassland, cropland, |
| RF3 | $T_{B19V}$, $T_{B37V}$, Latitude, Longitude, Elevation | | bareland, |
| RF4 | $T_{B19V}$, $T_{B37V}$, Latitude, Longitude, Elevation, Land cover fraction | | shurbland, forest |

**3) Revision 3: validation of the fitted RF algorithms**

We conducted three tests to verify the fitted RF algorithms (Table 2). The same training samples (same algorithms) were used for three tests but with different validation datasets. In Test1, the validation data are from out-of-bag (OOB) samples. Generally, in the RF model, approximately two-thirds of the samples (in-bag samples) are used to train the trees and the remaining one-third (OOB samples) are used to estimate how well the fitted RF algorithm performs. This preliminary assessment offers a simple way to adjust the parameters of the RF model. However, we should use the OOB errors with caution because its samples are not independent at temporal and spatial scales. In Test2, we applied temporally independent reference data during the period 2015-2018 to assess the accuracy of the temporal prediction of fitted algorithms. In Test3, a spatially independent dataset from validation stations during the period 2015-2018 was used to assess the accuracy of spatio-temporal prediction.

Fig. 4 indicates that the accuracy of RF model is greatly influenced by geographic location, elevation, and land cover fractions. However, the redundant predictor variables (if highly correlated) slightly affect the RF model (Fig. 3). The fitted RF algorithms perform better at

the temporal scale than that at the spatial scale, with unbiased RMSEs of ~4.4 cm and ~7.3 cm, respectively.

Table 2. Summary of three tests of the fitted RF algorithms in Table 1.

| Name | Test1 (OOB) | | Test2 (temporal subset) | | Test3 (spatio-temporal subset) | |
|---|---|---|---|---|---|---|
| training | training stations | 2012-2014 | training stations | 2012-2014 | training stations | 2012-2014 |
| | samples | 28602 | samples | 28602 | samples | 28602 |
| validation | training stations | 2012-2014 | training stations | 2015-2018 | validation stations | 2015-2018 |
| | samples | 14301 | samples | 34684 | samples | 25879 |

[Figure]

Figure 4. The color-density scatterplots of the estimated snow depth with four fitted RF algorithms and the ground truth snow depth. The four trained RF algorithms (RF1, RF2, RF3, RF4) were evaluated with three validation datasets (Test1, Test2, Test3).

**4) Revision 4: validation of the reconstructed snow depth product**

Finally, we directly used the fitted RF2 algorithm to retrieve a consistent 32-year daily snow depth dataset from 1987 to 2018. The product was evaluated against the independent station observations during the period 1987-2018. The mean unbiased RMSE and bias were 7.1 cm and -0.05 cm, respectively, outperforming the former snow depth dataset (8.4 cm and -1.20 cm) from the Environmental and Ecological Science Data Center for West China (WESTDC).

[Figure]

Figure 5. Scatterplots of the estimated snow depth and the ground truth observation for (a) RF and (b) WESTDC products.

To determine the interannual variability in the uncertainty, the time series of assessment indexes, including the unbiased RMSE, bias and correlation coefficient, are shown in Fig. 6. The results show that the RF estimates outperform the WESTDC product with respect to unbiased RMSE and correlation coefficient from season to season.

[Figure]

Figure 6. Time series of (a) unbiased RMSE (unRMSE), (b) correlation coefficient (corr.coe) and (c) bias for RF and WESTDC products. The colorful dashed lines represent mean values of assessment indexes.

The assessment of snow depth product was performed in three snow cover areas in China. We selected 20 cm as a threshold to assess the performances in deep (> 20 cm) and shallow (≤ 20 cm) snow cover. Table 3 displays the comparison between RF estimates and WESTDC product in the three snow cover areas. Both products present notable underestimation of deep snow cover, with the biases of -34.1 cm and -33.8 cm in QTP for the RF and WESTDC products, respectively. The biases are -10.4 cm and -8.9 cm in NE and northern XJ for RF product, respectively, whereas they are -11.8 cm and -13.2 cm for WESTDC data. For shallow snow cover, the RF product is superior to the WESTDC estimates in QTP, with unbiased RMSEs of 3.4 cm (RF) and 5.6 cm (WESTDC). Furthermore, the WESTDC product presents an overestimation in QTP, with a bias of 4.0 cm that is much higher than RF's 0.6 cm. The unbiased RMSEs of the RF product are 5.4 cm and 6.1 cm in NE and northern XJ for shallow snow cover, respectively, lower than the WESTDC's values of 6.5 cm and 7.4 cm.

In the Discussion, we list the potential errors of the reconstrued snow depth (Page 10, Line 18-28 and Page 11, Line 1-13, in the revised manuscript).

[Figure]

Figure 7. The validation of RF and WESTDC snow depth products in the three stable snow cover areas in China with respect to (a) the unbiased RMSE, (b) bias and correlation coefficient.

Table 3. Comparison between RF estimates and WESTDC product in three stable snow cover areas for deep (> 20 cm) and shallow (≤ 20 cm) snow cover.

| RF product | | | | | | |
|---|---|---|---|---|---|---|
| Regions | QTP | | NE | | northern XJ | |
| SnowDepth (cm) | <= 20 | > 20 | <= 20 | > 20 | <= 20 | > 20 |
| corr.coe | 0.30 | 0.06 | 0.49 | 0.17 | 0.48 | 0.31 |
| bias (cm) | 0.59 | -34.12 | 1.79 | -10.38 | 2.52 | -8.85 |
| unRMSE (cm) | 3.43 | 20.70 | 5.36 | 7.00 | 6.12 | 9.62 |
| Samples | 15503 (96.4%) | 583 (3.6%) | 151939 (87.3%) | 22168 (12.7%) | 32468 (69.8%) | 14051 (30.2%) |

| WESTDC product | | | | | | |
|---|---|---|---|---|---|---|
| Regions | QTP | | NE | | northern XJ | |
| SnowDepth (cm) | <= 20 | > 20 | <= 20 | > 20 | <= 20 | > 20 |
| corr.coe | 0.16 | -0.18 | 0.37 | 0.03 | 0.34 | 0.16 |
| bias (cm) | 4.02 | -33.78 | 0.47 | -11.75 | -0.39 | -13.22 |
| unRMSE (cm) | 5.60 | 21.62 | 6.47 | 9.10 | 7.35 | 11.30 |
| Samples | 15503 (96.4%) | 583 (3.6%) | 151939 (87.3%) | 22168 (12.7%) | 32468 (69.8%) | 14051 (30.2%) |

**Major issues:**

**1.** I agree with the Anonymous Reviewer (point 9b), who pointed that this complicated methodology of using RF to produce more data is probably unnecessary and that it should be tested whether this step is necessary and whether it does increase uncertainty to the product or not.

**Response 1:** Thank you for your comments. We tested four RF algorithms trained with different predictor variables (Fig. 4). The results showed that the accuracy of RF model is greatly influenced by geographic location, elevation, and land cover fractions. However, we also found redundant predictor variables due to high correlation. The elevation variable is highly correlated (correlations higher than 0.9) with geographic location (Fig. 3). Additionally, the correlation between longitude and land cover type (e.g., grassland, cropland, forest and bareland) is significant. Thus, land cover type and elevation are not

necessary. We directly used a simple RF algorithm to retrieve a consistent 32-year daily snow depth dataset from 1987 to 2018. Please refer to the response to "General Comments" above.

**2.** The most important issue is that the validation of the RF and the pixel based snow depths is not fair. This is because stations used for validation are only a temporal subsample of the training station set. The spatial sub-sampling was not conducted, i.e., stations for all geographic locations were used for training and validation. This is a very important problem, because latitude and longitude are the third and fourth important predictors in the model, nearly as important as the Tb. The RF model cannot know values of this predictors already during training, because the validation does not make sense. Therefore, the errors reported in this study are very optimistic (underestimated) and should be recalculated using 50% of the stations (not the data, i.e. spatial not temporal subset) which were not used to train the RF model. The same 50% subset should be used to validate the pixel-based method.

**Response 2:** Thank you for your constructive comments. Other reviewers gave similar comments. In the revised manuscript, available stations were randomly divided into two roughly equal-sized parts by Matlab software (Fig. 1). The snow depth observations from training stations (342 sites) together with satellite $T_B$ and other auxiliary data can be used to train the RF model. The measurements from validation stations (341 sites), as spatially independent data, can be applied to validate the fitted RF algorithm and the reconstructed snow depth product. Please refer to the response to "General Comments" above.

**3.** The pixel based SD product effectively fails to model snow above 20cm depth (Figure9). This is a serious limitation and it should be explained very deeply in the discussion: (1) why this happens, (2) what is the true applicability of the product given the RMSE is 5cm.

**Response 3:** Thank you for your comments. We discussed this in Section 4.3.
"Fig. 7 indicates that the RF model does not fully solve the overestimation and underestimation problems. For deep snow (> 20 cm), the biases are up to -8.9 cm and -10.4 cm in NE and northern XJ, respectively. Deep snow conditions account for roughly 10% of all training samples (Fig. 2). The estimates for deep snow cover in the QTP exhibit a large bias of -34.1 mm. Fig. 6 also illustrates that the fitted RF algorithms have no predictive ability for extremely deep snow conditions, especially in QTP. We checked the training data and found that the extreme high snow depth data (> 60 cm) occurred in QTP. However, the number of such cases is very small. In addition, the station measurements are point values while the satellite grids have a spatial resolution of 25 km × 25 km. Thus, the representativeness of these data is questionable. Snow depth estimation in the mountains remains a challenge (Lettenmaier et al., 2015; Dozier et al., 2016; Dahri et al., 2018). Numerous studies have been conducted on the snow cover over the QTP and have indicated that the snow cover in the Himalayas is higher than elsewhere, ranging from 80% to 100% during the winter (Basang et al., 2017; Hao et al., 2018). Additionally, Dai et al. (2018) showed that deep snow (greater than 20 cm) was mainly distributed in the Himalayas, Pamir, and Southeastern Mountains. Thus, the RF product produced in this paper has poor performance in QTP for deep snow cover."

**4.** The methods are very difficult to follow, I noticed that the other Reviewers managed to understand them better than me, but still, I am not completely sure how the study was conducted. This entire chapter should be rewritten, simplified and better structured. Often different words are used in the same context to name the same things, what makes understanding of this paper even more difficult (see attachment for some examples). The results and discussion sections are very poorly written: methods, results and discussion are mixed in each of these sections (see attachment for some examples).

**Response 4:** Thank you for your comments. We revised the manuscript carefully and thoroughly to make paper structure clearer. Additionally, a thorough revision of the manuscript was completed by a native speaker.

**5.** Authors should also justify better why this is a real-time approach. Is there an operational implementation of this algorithm?

**Response 5:** We removed the word 'real-time' in the revised manuscript.

**6.** Eventually, Authors claim that ML in RS is a very novel research problem, e.g. "Machine learning (ML) is a common method used in many research fields, and its early application in remote sensing is promising". The applications of ML in RS are not early, they are in RS since decades, either for regression (as in this study) or classification. The use of RF for regression, cannot be understand as a novelty, because it simply is not. Authors should better explain in which aspects the MS is novel.

**Response 6:** We apologize for the ambiguous description. We rewrote this paragraph as follows: "Over the last two decades, RF has been one of the most successful ML algorithms for practical applications due to its proven accuracy, stability, speed of processing and ease of use (Rodriguez-Galiano et al., 2012; Belgiu et al., 2016; Maxwell et al., 2018; Bair et al., 2018; Qu et al., 2019; Reichstein et al., 2019, Tyralis et sl., 2019a)" (Page 3, Line 2-5, in the revised manuscript).

In Section 2.2, we listed some advantages of the RF model. (Page 4, Line 19-30, in the revised manuscript).

We agree with your opinion that machine learning method is not novel in remote sensing and have rewritten the sentence. It now reads, "The primary objectives of this study are to assess the feasibility of the RF model in estimating snow depth, to determine whether the inclusion of auxiliary information (geolocation, elevation and land cover fraction) contributes to the improvement of RF, and eventually to develop a time series (1987 to 2018) of snow depth data in China and analyze the trends in annual mean snow depth. To complete the feasibility study of the RF model, we designed four RF algorithms trained with different combinations of predictor variables and validated them using independent reference data temporally and spatially. To our knowledge, this type of assessment of RF algorithm performance has not been made to date over China" (Page 3, Line 7-12, in the revised manuscript).

**Minor issues: (from hand-written comments)**

**1.** Page 1, line 20, the applications of ML in RS are not early, please remove early.

**Response 1:** Word 'early' removed.

**2.** Page 1, Line 22, from 1987-2018.

**Response 2:** We changed the sentence to 'during the period 1987-2018.'

**3.** Page 1, Line 23, 'the advanced microwave scanning radiometer'. The first letter should be capitalized.

**Response 3:** We selected SSM/I and SSMIS data as satellite observations and thus deleted this description.

**4.** Page 2, Line 23, this paper is about snow depth, not SWE.

**Response 4:** Thank you for your comments. We rewrote it as "Snow depth is a crucial parameter for climate studies, hydrological applications and weather forecasts (Foster et al., 2011; Takala et al., 2017; Tedesco et al., 2016; Safavi et al., 2017)."

**5.** Page 4, Line 8, not prediction, but regression.

**Response 5:** We changed 'prediction' to 'regression.'

**6.** Page 4, Line 24, 25*25km$^2$ ? ?

**Response 6:** It is 25 km x 25 km. We selected SSM/I and SSMIS data as satellite observations to retrieve snow depth in the revised manuscript and thus deleted this description.

**7.** Page 6, Line 7, cold overpass ? ?

**Response 7:** Thank you for your comments. We rewrote this sentence as 'To avoid the influence of wet snow, only ascending (F08) and descending (F11, F13 and F17) overpass data were used (Table 1).'

Table 1. Summary of the main passive microwave remote sensing sensors.

| Sensor | SSM/I | | | SSMIS |
|---|---|---|---|---|
| Satellite | DMSP-F08 | DMSP-F11 | DMSP-F13 | DMSP-F17 |
| On Orbit time | 1987-1991 | 1991-1995 | 1995-2008 | 2006-present |
| Passing Time | A: 06:20 D: 18:20 | A: 17:17 D: 05:17 | A: 17:58 D: 05:58 | A: 17:31 D: 05:31 |
| Frequency & footprint (GHz) : (km × km) | | 19.35: 45×68 23.235: 40×60 37: 24×36 85.5: 11×16 | | 19.35: 42×70 23.235: 42×70 37: 28×44 91.655: 13×15 |

**8.** Page 6, Line 13-15, station data is daily? What is harsh climate?

**Response 8:** Thank you for your comments. We rewrote these sentences as 'The weather station daily data in China from 1987 to 2018 were provided by the National Meteorological Information Centre, China Meteorology Administration (CMA, http://data.cma.cn/en)' and

'The sites are not distributed homogeneously, and few are located in inaccessible regions with extreme climates and complex terrain conditions, e.g., the western part of QTP.'

**9.** Page 6, Line 22, snow depth can be over 70 cm

**Response 9:** Thank you for your comments. Fig. 2 showed the histograms of snow depth observations from training and testing stations. Ninety percent of the samples range from 1 cm to 25 cm. The maximum values of the snow depth extend to approximately 50 cm. However, the number of such cases is small and therefore not evident. In the revised manuscript, we maintained these data.

**10.** Page 7, Line 15-19, the description is not clear.

**Response 10:** Thank you for your comments. We rewrote this paragraph in Section 2.2.1.
'2.2.1 Random forest
RF is an ensemble ML algorithm proposed by Breiman in 2001. It combines several randomized decision trees and aggregates their predictions by averaging in regression (Biau and Scornet, 2016). Generally, approximately two-thirds of the samples (in-bag samples) are used to train the trees and the remaining one-third (out-of-bag samples, OOB) are used to estimate how well the fitted RF algorithm performs. Few user-defined parameters are generally required to optimize the algorithm, such as the number of trees in the ensemble ($n$tree) and the number of random variables at each node ($m$try). The $n$tree is set equal to 1000 in the present study since the gain in the predictive performance of the algorithm would be small with the addition of more trees (Probst and Boulesteix, 2018). The default value of $m$try is determined by the number of input prediction variables, usually 1/3 for regression tasks (Biau and Scornet, 2016). The RF regression is insensitive to the quality of training samples and to overfitting due to the large number of decision trees produced by randomly selecting a subset of training samples and a subset of variables for splitting at each tree node (Maxwell et al., 2018). In addition, RF provides an assessment of the relative importance of predictor variables, which have proven to be useful for evaluating the relative contribution of input variables (Tyralis et al., 2019b). Furthermore, the RF model can rapidly trained and is easy to use. In this paper, a randomForest R package (Version 4.6-14) is used for regression (Liaw and Wiener 2002; Breiman et al. 2018)."

**11.** Page 7, Line 27-28, why you asking questions here. Page 8, Line 3, 80000 pairs? Not clear

**Response 11:** We deleted the questions and rewrote this paragraph in Section 2.2.2.
'(2) Training sample size
One of the advantages of the RF model is that it can effectively handle small sample sizes (Biau and Scornet et al., 2016). A test was conducted to demonstrate the insensitivity of the RF model to the training sample size. The input predictor variables include geographic location and $T_B$ (Table 2, RF2). The flowchart of the test process is shown in Fig. 4. To ensure a sufficient number of samples, 80,000 records from 1987 to 2004 were used to test the required size of the training samples and a two-year stand-alone dataset from (2005-2006) was applied to assess the performance. During this process, the number of

samples selected randomly was from 5000 to 80,000 (step, 5000). We consider three evaluating indicators (the unbiased root mean square error (RMSE), bias and correlation coefficient) to illustrate the stability of the RF model to the training sample size."

**12.** Page 8, Line 4-8, what is this paragraph about? What is 'stability'? in respect to what?

**Response 12:** Thank you for your comments. We tested the sensitivity of the RF model to the training sample size. We rewrote this paragraph. Please refer to the response to "Minor Comment 11" above.

**13.** Page 8, Line 15-26, this is ambiguous. Which radiation is scattered by snow? Which radiation the snow is transparent? What is the snow of these radiations? Perhaps some of the radiation is radiated by snow itself, not scattered…

**Response 13:** Thank you for your comments. Most passive microwave snow depth retrieval algorithms exploit the negative spectral gradient between measurements at 19 GHz and 37 GHz. We rewrote this paragraph in Section 2.2.2.
'All available channels on the SSM/I and SSMIS are listed in Table 1. The 23 GHz channel is sensitive to water vapor and not surface scattering, which introduces uncertainty to the estimation process (Ji et al., 2017). The 85 (91) GHz channel is seriously influenced by the atmosphere (Kelly et al., 2009; Xue et al., 2017). Typically, the lower frequency (19 GHz) is used to provide a background $T_B$ against which the higher frequency (37 GHz) scattering-sensitive channels are used to retrieve snow depth.'

**14.** Page 9, Line 2-4, this sentence should move to the introduction section.

**Response 14:** We left out the pixel-based method in this paper due to RF's limitations.

**15.** Page 9, Line 6-7, 19GHz is always 18GHz.

**Response 15:** Thank you for your comments. We used the same symbol in the manuscript. 'In this study, the difference between 19.35 (36.5) GHz and 18.7 (37) GHz was ignored (hereafter referred as 19 GHz and 37 GHz, respectively).'

**16.** Page 9, Line 24-25, seasons, should be season or months. Isn't wet snow likely in November?

**Response 16:** We changed the word 'seasons' to 'season.' Although a snow cover detection method (Grody et al., 1996) was used to filter out wet snow conditions, wet snow is still possible in November.

**17.** Page 10, Line 1-3, some repetition, not clear.

**Response 17:** We modified the sentence to "The sensitivity of the RF model to the training sample size was conducted to confirm the appropriate number of training samples."

**18.** Page 10, Line 5, the term 'represents' is changed to 'presents'. RMSE ranges.., not RMSEs range…

**Response 18:** Thank you for your comments. We rewrote this sentence as 'Fig. 4a presents unbiased RMSE ranges from 5.1 cm to 5.5 cm.'

**19.** Page 10, Line 10, what is the optimal number you chosen here?

**Response 19:** According to the sensitivity analysis, the number of training samples has less influence on the prediction accuracy of the RF model. In our study, we selected all available samples (28602) from training stations (Fig. 1) during the period 2012-2014 to train the RF models.

**20.** Page 10, Line 11, this statement is not supported by the results.

**Response 20:** One of the advantages of the RF model is that it can effectively handle small sample sizes (Biau and Scornet et al., 2016). Our results also indicated that the performance of RF model is insensitive to the training sample size.

**21.** Page 10, Line 16-18, please move this sentence to the method section.

**Response 21:** We moved it to Section 2.2.2.

**22.** Page 10, Line 23, this is discussion, not result.

**Response 22:** It was moved to Section 4.3.

**23.** Page 11, Line 2-3, how the relative error was calculated?

**Response 23:** $RPE=abs(bias*100/SD_{ground})$.

**24.** Page 11, Line 6-8, is method, not result.

**Response 24:** Moved.

**25.** Page 11, Line 11-13, the reference?

**Response 25:** We added the reference and moved this sentence to the discussion section. " Second, the large diurnal temperature range tends to subject the snowpack to frequent freeze-thaw cycles and leads to rapid snow grain (~2 mm) and snow density (200-350 $kg/m^3$) growth and consequently a high TB difference (Meløysund et al., 2007; Durand et al., 2008; Yang et al., 2015; Dai et al., 2017)."

**26.** Page 11, Line 16-19, aren't only cold/night orbits data used?

**Response 26:** In this study, a snow cover detection method is used to filter out wet snow cover; however, there are still misclassification errors, especially at the end of winter (Liu et al., 2018).
Liu, X., Jiang, L., Wu, S., Hao, S., Wang, G., and Yang, J.: Assessment of Methods for Passive Microwave Snow Cover Mapping Using FY-3C/MWRI Data in China, Remote Sensing, 10, 524-539, 10.3390/rs10040524, 2018.

**27.** Page 11, Line 25-30, this is how to judge base on the maps?

**Response 27:** We moved this sentence to the discussion.

**28.** Page 12, Line 12-16, mixing results and discussion!

**Response 28:** We moved this sentence to Section 4.3.

**29.** Page 12, Line 25-27, move to method section!

**Response 29:** In the revised manuscript, we left out the pixel-based method and thus deleted this sentence.

**30.** Page 13, Line 3-4, where and why?

**Response 30:** We deleted this sentence because the pixel-based method was left out in the revised manuscript.

**31.** Page 13, Line 12-13, this sentence should be "To evaluate the long-term…."

**Response 31:** We corrected this sentence.

**32.** Page 13, Line 23, where is the comparison?

**Response 32:** We rewrote it as "The overall accuracy of the RF product is higher than the WESTDC estimates, with unbiased RMSEs of 7.1 cm and 8.5 cm, respectively (Fig. 7a and 7b)."

**33.** Page 15, Line 3-13, move to results section.

**Response 33:** Done.

**34.** Page 15, Line 17-22, only cold/night orbits data were used in winter season, how to explain it?

**Response 34:** Please refer to the response to "Minor Comment 26" above.

**35.** Page 15, Line 22, It is result, not discussion.

**Response 35:** Moved.

**36.** Page 16, Line 2, "H-pol" is "in horizontal polariton".

**Response 36:** Corrected.

**37.** Page 16, Line 8-15, not clear explanation. Not 'predictor importance' but 'predictor variable importance'.

**Response 37:** We modified the sentence to "The importance of predictor variables, referred to as Mean Decrease Accuracy (MDA) in the RF model, is obtained by averaging the difference in out-of-bag error estimation before and after the permutation over all trees. The larger the MDA, the greater the importance of the variable is" (Page 19, Line 6-9, in the revised manuscript).

**38.** Page 16, Line 12, remove the 'by far', 'more dependent on station data' is changed to 'geographically dependent'.

**Response 38:** Done.

**39.** Page 16, Line 17-27, the result does not support this because DEM was not a predictor variable in this paper. If DEM is better than lat/lon, why not use DEM?

**Response 39:** We redesigned the procedure and included the DEM as one of the predictor variables (Table 1). Fig. 3 indicates that DEM is highly correlated with the geolocation (lat/lon).

**40.** Page 17, Line 2, Significantly? Statistical test conducted?

**Response 40:** It means that there is a notable accuracy difference for different land cover types. We deleted the word 'significantly.'

**41.** Page 17, Line 3, what if land cover changes over time?

**Response 41:** This is a wonderful question. In this study, we assume the land cover type does not change. We can study this in future work.

**42.** Page 17, Line 15-29, These sentences belong to method section.

**Response 42:** The aim of this part is to demonstrate that more prior snow information can improve the performance of the RF model. According to Reviewer #4's suggestion, we omitted this and will present it in a future publication.

**43.** Page 18, Line 4-6, This part is discussion.

**Response 43:** We moved it to Section 4.1.

**44.** Page 18, Line 8, where is this method?

**Response 44:** The method is the pixel-based algorithm. We omitted this part.

**45.** Page 18, Line 11, past or present

**Response 45:** We revised the manuscript carefully and thoroughly to make the tense correct.

**46.** Page 18, Line 15, than the former…

**Response 46:** word 'former' added.

**47.** Page 18, Line 16-20, is this really a conclusion? Page 18, Line 21, This is not a conclusion, but summary. What is the conclusion here? I do not find…

**Response 47:** We rewrote the conclusion (Page 11, Line 14-28, Page 12, Line 1-16, in the revised manuscript).
"The present study analyzed the application of the RF model to snow depth estimation at temporal and spatial scales. Temporally and spatially independent datasets were applied

to verify the fitted RF algorithms. The results suggested that the accuracy of fitted RF algorithms was greatly influenced by auxiliary data, especially the geographic location. However, the inclusion of strongly correlated predictor variables (elevation and land cover fraction) did not further improve the RF estimates. Therefore, in some cases, a few representative predictor variables should be selected. Due to naive extrapolation outside the training range, the transferability of a fitted RF algorithm at the temporal scale was better that that in spatial terms, e.g., with unbiased RMSEs of 4.5 cm and 7.2 cm for the RF2 algorithm, respectively.

In this study, the fitted RF2 algorithm was used to retrieve a consistent 32-year daily snow depth dataset from 1987 to 2018. Then, an evaluation was carried out using independent reference data from the validation stations during the period 1987-2018. The overall unbiased RMSE and bias were 7.1 cm and -0.05 cm, respectively, outperforming the WESTDC product (8.4 cm and -1.20 cm). In QTP, the unbiased RMSE and bias of RF estimates for shallow (≤ 20 cm) snow cover were 3.4 cm and 0.59 cm, respectively, much lower than WESTDC's 5.6 cm and 4.02 cm. In NE and northern XJ, RF estimates were superior to the WESTDC product but still presented an underestimation for deep snow (> 20 cm), with biases of -10.4 cm and -8.9 cm, respectively.

Three long-term (1987-2018) datasets, including ground truth observations, RF estimates and WESTDC product, were applied to analyze the trends of snow depth variation in China. The results suggested that there existed different trends among the three datasets. The overall trend of snow depth in China presented a significant increasing based on the ground truth observations, with a correlation coefficient of 0.57. Moreover, the trend in NE was highly consistent with the overall trend in China, with a correlation coefficient of 0.64. Neither the WESTDC nor the RF product presented significant trends except in QTP. The WESTDC product showed a significant decreasing trend in QTP, with a correlation coefficient of -0.55, whereas there were no significant trends for ground truth observations and the RF product.

As discussed in Section 4, our reconstructed snow depth estimates are still challenged by several problems, e.g., underestimation for deep snow. Additional prior knowledge of snow cover, such as snow cover fraction, snow density, and snow grain size, is necessary to improve the RF model. Combining the snow forward model with the ML method will be the focus of future work. Furthermore, the mass balance approaches, e.g., the Parallel Energy Balance model, will be used to improve the snow depth retrievals in high-altitude areas. In addition, although our results indicate that the RF method is a promising potential tool for snow depth estimation, there are a few pitfalls such as the risk of naive extrapolation and poor transferability in spatial terms limiting its application in spatio-temporal dynamics. It is in addressing these shortcomings that the techniques of deep learning promise breakthroughs. We are attempting to operate the Deep Neural Networks (DNN) model to overcome the limitations of traditional ML approaches."

---

## Author Comment (AC4) · 5 Jan 2020

**Response to Reviewer Comments by Nir Krakauer on "Real-Time Snow Depth Estimation and Historical Data Reconstruction Over China Based on a Random Forest Machine Learning Approach" by Jianwei Yang et al.**

Thank you for your letter and the comments concerning our manuscript. Those comments have been very helpful for revising and improving our paper as well as providing guidance for our research. We have studied the comments carefully and have made corrections, which we hope meet with approval. We provide responses in blue below.

Review #4

**General Comments:** The basic theme of this manuscript, the application of random forest (RF) to provide an empirical transfer model from remotely sensed radiances to snow depth, has merit, given that physically based transfer models are subject to limitations. However, some of the modeling choices appear questionable and should be better justified or simplified. The RF modeling described in Section 2.3 has the following main components: (1) Using SSMI data from 1987-2004 for training and from 2005-2006 for validation, in order to evaluate the number of training samples required for good accuracy. (2) Using AMSR2 data from 2014-2015 for training and from 2012-2013 for validation. Snow depth estimated by this model is then used to generate an approximate spatially varying relationship between 2 SSMI channel radiances and snow depth. The resulting simple SSMI-based formula is used to reconstruct estimated snow depth for 1987-2018, which is validated for 2017-2018.

**Specific comments:**

**1.** Approach (2) appears unnecessarily complicated. If the goal is to establish a product

for 1987-2018, where only SSMI inputs are available for the entire period, it is more logical to train an RF model directly with SSMI inputs (as done in (1) – not with AMSR2 inputs) fitted to station data (not reconstructed data). If the authors want to retain their more complicated approach, they should compare it to the simpler one to demonstrate that it actually has superior accuracy.

**Response 1:** We agree with the reviewer's opinion, and these suggestions are very constructive. Other reviewers gave us similar comments. Thus, we directly selected SSM/I and SSMIS data as satellite observations in the revised manuscript.

The procedure described in the original manuscript was complicated. Based on the correlations between the predictor variables and the variable importance metrics (Fig. 1), we designed four schemes of predictor variables to train the RF model in the revised manuscript. The scheme one was the simplest and its predictor variables included satellite observations at 19 GHz and 37 GHz only (Table 1). The scheme four was the most complicated. We first demonstrated whether certain predictor variables are necessary and whether their inclusion affects the RF model.

[Figure]

Figure 1. Correlations between the predictor variables (left) and the ranking of variable importance (right). The importance of variables, referred to as Mean Decrease Accuracy (MDA) in RF model, is obtained by averaging the difference in out-of-bag error estimation before and after the permutation over all trees. The larger the MDA, the greater the importance of the variable is.

Table 1. A detailed description of the input predictor variables based on four selection rules of training sample.

| Name | Predictor Variables | Target | Note |
|------|---------------------|--------|------|
| RF1 | $T_{B19V}$, $T_{B37V}$ | | land cover types: |
| RF2 | $T_{B19V}$, $T_{B37V}$, Latitude, Longitude | snow depth | grassland, cropland, |
| RF3 | $T_{B19V}$, $T_{B37V}$, Latitude, Longitude, Elevation | | bareland, |
| RF4 | $T_{B19V}$, $T_{B37V}$, Latitude, Longitude, Elevation, Land cover fraction | | shurbland, forest |

Then, we conducted three tests to verify the fitted RF algorithms (Table 1). The same training samples (same algorithms) were used for the three tests but with different validation datasets. In Test1, the validation data are from out-of-bag (OOB) samples. Generally, in the RF model, approximately two-thirds of the samples (in-bag samples) are used to train the trees and the remaining one-third (OOB samples) are used to estimate how well the fitted RF algorithm performs. This preliminary assessment offers a simple way to adjust the parameters of the RF model. However, we should use the OOB errors with caution because its samples are not independent at temporal and spatial scales. In Test2, we applied temporally independent reference data during the period 2015-2018 to assess the accuracy of temporal prediction of fitted algorithms. In Test3, a spatially independent dataset from validation stations during the period 2015-2018 was used to assess the accuracy of spatio-temporal prediction.

Fig. 2 indicates that the accuracy of RF model is greatly influenced by geographic location, elevation, and land cover fractions. However, the redundant predictor variables (if highly correlated) slightly affect the RF model. The fitted RF algorithms perform better at the temporal scale than that at the spatial scale, with unbiased RMSEs of ~4.4 cm and ~7.3 cm, respectively.

**Table 2. Summary of three tests to the fitted RF algorithms in Table 1.**

| Name | Test1 (OOB) | | Test2 (temporal subset) | | Test3 (spatio-temporal subset) | |
|---|---|---|---|---|---|---|
| training | training stations | 2012-2014 | training stations | 2012-2014 | training stations | 2012-2014 |
| | samples | 28602 | samples | 28602 | samples | 28602 |
| validation | training stations | 2012-2014 | training stations | 2015-2018 | validation stations | 2015-2018 |
| | samples | 14301 | samples | 34684 | samples | 25879 |

[Figure]

Figure 2. The color-density scatterplots of the estimated snow depth with four fitted RF algorithms and the ground truth snow depth. The four trained RF algorithms (RF1, RF2, RF3, RF4) were evaluated with three validation datasets (Test1, Test2, Test3).

Finally, we directly used the fitted RF2 algorithm to retrieve a consistent 32-year daily snow depth dataset. It was evaluated against the independent ground truth measurements from the validation stations (Fig. 6) during the period 1987-2018. The mean unbiased RMSE and bias were 7.1 cm and -0.05 cm, respectively, outperforming the former snow depth

dataset (8.4 cm and -1.20 cm) from the Environmental and Ecological Science Data Center for West China (WESTDC).

[Figure]

Figure 3. Scatterplots of the estimated snow depth and the ground truth observation for (a) RF and (b) WESTDC products.

To determine the interannual variability in the uncertainty, the time series of assessment indexes, including the unbiased RMSE, bias and correlation coefficient, are shown in Fig. 4. The results show that the RF estimates outperform the WESTDC product with respect to unbiased RMSE and correlation coefficient from season to season.

[Figure]

Figure 4. Time series of (a) unbiased RMSE (unRMSE), (b) correlation coefficient (corr.coe) and (c) bias for RF and WESTDC products. The colorful dashed lines represent mean values of assessment indexes.

The assessment of snow depth product was also performed in three snow cover areas in China for shallow (≤ 20 cm) and deep snow cover (> 20 cm).

[Figure]

Figure 5. The validation of RF and WESTDC snow depth products in three stable snow cover areas in China with respect to (a) the unbiased RMSE, (b) bias and correlation coefficient.

Table 3. Comparison between RF estimates and WESTDC product in three stable snow cover areas for deep (> 20 cm) and shallow (≤ 20 cm) snow cover.

| RF product | | | | | | |
|---|---|---|---|---|---|
| Regions | QTP | | NE | | northern XJ | |
| SnowDepth (cm) | <= 20 | > 20 | <= 20 | > 20 | <= 20 | > 20 |
| corr.coe | 0.30 | 0.06 | 0.49 | 0.17 | 0.48 | 0.31 |
| bias (cm) | 0.59 | -34.12 | 1.79 | -10.38 | 2.52 | -8.85 |
| unRMSE (cm) | 3.43 | 20.70 | 5.36 | 7.00 | 6.12 | 9.62 |
| Samples | 15503 (96.4%) | 583 (3.6%) | 151939 (87.3%) | 22168 (12.7%) | 32468 (69.8%) | 14051 (30.2%) |

| WESTDC product | | | | | | |
|---|---|---|---|---|---|
| Regions | QTP | | NE | | northern XJ | |
| SnowDepth (cm) | <= 20 | > 20 | <= 20 | > 20 | <= 20 | > 20 |
| corr.coe | 0.16 | -0.18 | 0.37 | 0.03 | 0.34 | 0.16 |
| bias (cm) | 4.02 | -33.78 | 0.47 | -11.75 | -0.39 | -13.22 |
| unRMSE (cm) | 5.60 | 21.62 | 6.47 | 9.10 | 7.35 | 11.30 |
| Samples | 15503 (96.4%) | 583 (3.6%) | 151939 (87.3%) | 22168 (12.7%) | 32468 (69.8%) | 14051 (30.2%) |

**2.** There is another way to tackle the problem of different microwave satellite sensors being available over different portions of the 1987-2018 period, which the authors may also want to consider. This would involve combining estimates from multiple fitted RF models, one for each satellite sensor available for part of the time period, which would potentially more fully use the partly-independent information from multiple satellite sources, which may each have different wavelength ranges, overpass times, and other sensor characteristics.

**Response 2:** These suggestions are very constructive. However, as a change from the original manuscript, we resorted to using only SSM/I and SSMIS data as satellite observations in this study. As shown in Table 4 below, the characteristics of these sensors are sufficiently similar to assume that an algorithm defined for one sensor can be applicable

to the next. We have rewritten the introduction of satellite data in Section 2.1: "The SSM/I and SSMIS sensors are suitable for producing a long-term consistent snow depth dataset due to their similar configurations and intersensor calibrations (Armstrong et al., 1994)" (Page 3, Line 21-23, in the revised manuscript).

Table 4. Summary of the main passive microwave remote sensing sensors.

| Sensor | SSM/I | | | SSMIS |
|---|---|---|---|---|
| Satellite | DMSP-F08 | DMSP-F11 | DMSP-F13 | DMSP-F17 |
| On Orbit time | 1987-1991 | 1991-1995 | 1995-2008 | 2006-present |
| Passing Time | A: 06:20 | A: 17:17 | A: 17:58 | A: 17:31 |
| | D: 18:20 | D: 05:17 | D: 05:58 | D: 05:31 |
| Frequency & footprint (GHz) : (km × km) | | 19.35: 45×68 | | 19.35: 42×70 |
| | | 23.235: 40×60 | | 23.235: 42×70 |
| | | 37: 24×36 | | 37: 28×44 |
| | | 85.5: 11×16 | | 91.655: 13×15 |

**3.** Another issue is the training/validation station data split. As one of the other reviewers points out, in order to better estimate the error at ungauged sites, it makes more sense to not use some stations at all for training and retain them for validation, instead of validating with data for the same stations but different years.

**Response 3:** Thank you for your comments. One of the major issues of this study is that the validation data are not temporally and spatially independent. Thus, available stations in China were randomly divided into two roughly equal-sized parts by Matlab software (Fig. 6). The snow depth observations from training stations (342 sites) together with satellite $T_B$ and other auxiliary data can be used to train the RF model. The measurements from validation stations (341 sites), as spatially independent data, can be applied to validate the fitted RF algorithm and the reconstructed snow depth product. Fig. 7 shows the histograms of snow depth observations from training and validation stations during the period 2012-2018. Ninety percent of the samples range from 1 cm to 25 cm. The maximum values of the snow depth extend to approximately 50 cm. However, the number of such cases is small and is therefore not evident in Fig. 7.

[Figure]

Figure 6. Spatial distribution of the weather stations and land cover types in the study area. There are three stable snow cover areas in China: Northeast China (NE), northern Xinjiang (XJ) and the Qinghai-Tibetan Plateau (QTP).

[Figure]

Figure 7. Histograms of snow depth observations from (a) training and (b) validation stations. The average values (black dashed lines) are equal to 10.5 cm and 9.8 cm, respectively.

**4.** There is no comparison presented between the RF method and physically based transfer models or existing satellite or reanalysis snow products over China. This work would be stronger if the authors can conduct such a comparison and show whether RF in fact leads to improvements in snow estimation beyond existing approaches.

**Response 4:** Thank you for your comments. The linear-fitting method was developed based on SSM/I observations and station snow depth data by Che et al (2008). The daily snow depth data were obtained from the Environmental and Ecological Science Data Center for West China (http://westdc.westgis.ac.cn) (hereafter, WESTDC product). Yang et al. (2019) demonstrated that the WESTDC product outperforms four other snow depth datasets in China. Thus, in this study, we directly compared the RF estimates with the WESTDC product.

We also show that an overall improvement of 15.4 % in China is achieved compared to the WESTDC product (Fig. 3). In QTP, the unbiased RMSE and bias of RF estimates for shallow (≤ 20 cm) snow cover were 3.4 cm and 0.59 cm, respectively, much lower than WESTDC's 5.6 cm and 4.02 cm (Table 3). Please refer to the response to "Specific comment 1" above.

[1] Yang, J., Jiang, L., Wu, S., Wang, G., Wang, J., and Liu, X.: Development of a Snow Depth Estimation Algorithm over China for the FY-3D/MWRI, Remote Sensing, 11, 977, 10.3390/rs11080977, 2019.

**5.** Section 4.5 discusses the performance of an RF model under an ensemble of simulated weather conditions and microwave radiances. It is not clear what this section adds to the stronger results of the earlier section, which are based on real satellite and snow data. The authors should consider omitting it, and returning to these considerations in a future publication.

**Response 5:** We agree and deleted it.

**6.** Also, the authors should discuss the difference between snow depth and snow water equivalent (SWE). To my understanding, SWE is more relevant for hydrologic applications, and may be more directly measured by the microwave retrievals.

**Response 6:** We agree with the reviewer's opinion. Snow water equivalent (SWE), describing the amount of water stored in a snowpack, is a key variable for hydrological applications. Generally, a reasonable 'global' snow density (240 kg/m$^3$) is used to transfer snow depth to SWE (Takala et al., 2011).
In our study, we used the RF algorithm to retrieve snow depth rather than SWE because that station observations include only snow depth data.
Generally, snow density presents a variation in space and time. Thus, a relation to SWE through a fixed snow density is unreasonable. In the future, the temporospatial distribution of snow density in China will be mapped based on the reanalysis data from ERA5-land to improve SWE estimation. We are now assessing the ERA5 data using ground truth observations.

Takala, M., Luojus, K., Pulliainen, J., Lemmetyinen, J., Juha-Petri, K., Koskinen, J., and Bojkov, B., 2011. Estimating northern hemisphere snow water equivalent for climate research through assimilation of space-borne radiometer data and ground-based measurements. Remote Sensing of Environment. 115, 3517-3529.

**7.** On a related note, the authors note that snow measurements in high mountain areas are sparse, so that remote sensing based snow estimates cannot be validated. This could be partly overcome using a mass balance approach based on, for example, spring and summer streamflow measurements, which would give SWE (and hence, making assumptions about density, also snow depth) on a watershed scale (which in some cases might even be comparable with the satellite spatial resolution scale). See, e.g., Dahri et al.

(2018) "Adjustment of measurement errors to reconcile precipitation distribution in the high-altitude Indus basin" and related work.

**Response 7:** We appreciate your constructive suggestions. We are considering a snow depletion curve, e.g., Parallel Energy Balance Model, to improve the snow depth retrievals in high-altitude areas. We read the reference carefully and cited it in the revised manuscript. "Snow depth estimation in the mountains remains a challenge (Lettenmaier et al., 2015; Dozier et al., 2016; Dahri et al., 2018)" (Page 10, Line 25-26).

---

## Author Comment (AC5) · 5 Jan 2020

Dear editor,

First, we would like to thank the four referees and the editor for dedicating their time to our manuscript and providing us with positive and constructive comments.

Our major revisions include the following points:

- **Scientific validation dataset:**
  One of major issues of the original study was that the validation data were not temporally and spatially independent from the training data. Thus, available stations were randomly divided into two roughly equally sized parts: training stations and validation stations. The snow depth observations from training stations (342 sites) together with satellite $T_B$ and other auxiliary data can be used to train the RF model. The measurements from validation stations (341 sites), as spatially independent data, can be applied to validate the fitted RF algorithm and reconstructed snow depth product.

- **Four combinations of predictor variables:**
  The procedure described in the original manuscript was complicated due to so many predictor variables. Based on the correlations between the predictor variables and the variable importance metrics, we designed four schemes of predictor variables to train the RF model in the revised manuscript. The scheme one was the simplest and its predictor variables included satellite observations at 19 GHz and 37 GHz only. The scheme four was the most complicated. The predictor variables were satellites observations, latitude, longitude, elevation and land cover fraction. These four combinations of predictor variables, together with snow depth measurements, trained the four RF algorithms. We validated these four fitted RF algorithms to determine whether certain predictor variables are necessary and whether their inclusion affects the RF model.

- **Validation of the fitted RF algorithms:**
  The fitted RF algorithm was validated using temporally independent data in the original manuscript. To assess the feasibility of RF model in estimating snow depth, we conducted three tests to verify the fitted RF algorithms in the revised manuscript. The same training samples (same algorithms) were used for three tests but with different validation datasets. In Test1, the validation data were from out-of-bag (OOB) samples. Generally, approximately two thirds of the samples (in-bag samples) were used to train the trees and the remaining one-third (OOB samples) were used to estimate how well the fitted RF algorithm performed. This preliminary assessment generally provides a simple way to adjust the parameters of the RF model. In Test2, we applied temporally independent reference data during the period 2015-2018 to assess the accuracy of the temporal prediction of fitted algorithms. In Test 3, a spatially independent dataset from validation stations during the period 2015-2018 was used to assess the accuracy of spatio-temporal prediction.
  According to the validation of the fitted RF algorithms, we found many redundant inputs due to highly correlated predictor variables. Thus, we used a straightforward fitted RF algorithm (trained with $T_B$ and geolocation) to retrieve a consistent 32-year daily snow depth dataset from 1987 to 2018.

- **Validation of the reconstructed snow depth product:**

This product was evaluated against the independent station observations during the period 1987-2018. We also compared the performances of snow depth product in three snow cover areas over China.

- **Trends analysis of snow depth:**

  Three long-term (1987-2018) datasets, including ground truth observations, RF estimates and former snow depth product in China, were applied to analyze the trends of snow depth variation in China using the Mann-Kendall test and slope method.

- **Available long-term snow depth dataset:**

  The reconstructed dataset from 1987 to 2018 is now available and we will upload the data later.

- **Rewritten, simplified and better structured:**

  We revised the manuscript carefully and thoroughly to clarify the structure and content of the paper. We rewrote the results and discussion sections and split them. Additionally, a thorough revision of the manuscript was completed by a native speaker.

We have studied the comments carefully and have made corrections, which we hope will meet with approval.

We thank you for giving us the chance to revise the manuscript, and we look forward to hearing from you.

Jianwei Yang and Lingmei Jiang, on behalf of the authors

---

## Author Comment (AC6) · 5 Jan 2020

We thank you for your correction of review. Please find our responses (Responses to RC3) attached as a supplement.

---

## Referee Report (RR1)

**Review of the manuscript Yang et al., "Real-Time Snow Depth Estimation and Historical Data Reconstruction Over China Based on a Random Forest Machine Learning Approach."**

The authors have made significant improvements in the revised manuscript. However, some issues, as mentioned previously, remain. I am not convinced by some of the responses by the authors. These issues are as follows.

1. My primary concern was the selection of the RF algorithm for snow depth estimation. With the broader availability of deep neural networks for similar applications, I am not convinced by the utilization of RF for snow depth estimation. Therefore, the authors should either demonstrate that such advanced methods do not provide a significant improvement in accuracy over RF algorithm or re-do the experiments with these methods. Without this analysis or these experiments, we cannot ascertain the potential of the proposed method, particularly in practical applications.

   The experiment with ANN by the authors is appreciated. However, this is not at all convincing, since, we already know from the literature that for medium resolution or coarse resolution imagery, SVM or RF outperforms the ANN.

2. The authors have now discussed the impact of the diurnal changes in the snowpack geophysical properties, including the microstructure, as mentioned in my previous comment. However, this discussion is not comprehensive. Due to the absence of any experiments, this cannot be correlated to the errors observed. Thus, the analysis and the discussion part in this manuscript is still lacking.

   For example, the authors have considered in their RF algorithms inputs such as the latitude and the longitude. But why not any snowpack specific parameters. These can be obtained from the ERA5 data and used as inputs to their RF models. The authors then should check the discrepancy in the RMSE based on these parameters.

   Another analysis that I find missing is seasonal variation in accuracy, which is very important. The readers should know the difference in the performance of the algorithm in the off-peak snow seasons, i.e., during the early winter and the melt season.

3. Please mention the upscaling method for 1-km to 25-km LULC map. Additionally, I also mention which latitude and longitudes were used as inputs. I hope these were geographic ones and not projected ones.

4. I find that the majority of the discussion is based on the comparison between the results from the RF method and the WESTDC. While I believe there should be a better balance in the discussion for comparison with the in-situ measurements.

5. Another analysis to the manuscript that the authors should add is for the selection of trees. Presently, the authors have fixed this to 1000. However, is there any merit in using this large number? What is the performance for other smaller number of trees, for example, 500, 250, etc. Is the performance significant enough to use such a relatively large number of trees?

6. An important parameter in the discussion is the R-square that is missing?

7. In Table 5, the results are contradictory to the explanations given by the authors. The authors have mentioned that for shallow snow depth, the PMW could be insensitive. However, Table 5 shows the contradictory, i.e., rather some sensitivity in case of shallow snow depths as compared to deeper snow where there is nearly no correlation. I believe the inconsistency of the results requires a thorough analysis.

*Minor issues*

- Please check Figure 4 for correctness. Is the validation dataset correct? This is very confusing from the authors response.
- Some of the colors used by authors are very poor. Figure 6 the turquoise text, Figure 9, the orange legends and lines.
- Although I meant with my comment to include a discussion, the MEMLS simulation experiment is appreciated. However, the parameters used in the MEMLS simulation are not correct in the response by the authors. How often do we have a snowpack representing fresh snow characteristics? This is very seldom. Thus, there is no point showing an experiment with a snow density of 100 kg/cubic m corresponding to fresh snow. Instead, the authors should have selected something between 200-350 km/cubic m for the simulation, which corresponds to old snow.

---

## Author Response (AR2)

**Response to Reviewer Comments by Divyesh Varade on "Real-Time Snow Depth Estimation and Historical Data Reconstruction Over China Based on a Random Forest Machine Learning Approach" by Jianwei Yang et al.**

Thank you for your further comments concerning our manuscript. Those comments are very helpful for revising and improving our paper as well as providing important guidance for our future research. We have studied the comments carefully and have made corrections, which we hope meet with approval. The detailed corrections and the point by point responses to your comments are listed below:

Review #1

**General Comments:**

**1.** My primary concern was the selection of the RF algorithm for snow depth estimation. With the broader availability of deep neural networks for similar applications, I am not convinced by the utilization of RF for snow depth estimation. Therefore, the authors should either demonstrate that such advanced methods do not provide a significant improvement in accuracy over RF algorithm or re-do the experiments with these methods. Without this analysis or these experiments, we cannot ascertain the potential of the proposed method, particularly in practical applications.

The experiment with ANN by the authors is appreciated. However, this is not at all convincing, since, we already know from the literature that for medium resolution or coarse resolution imagery, SVM or RF outperforms the ANN.

**Response 1:** Thank you for your comments. The basic assumption for the widely used empirical or semiempirical inversion algorithms is that TB at K-band (~19 GHz) is weakly affected by the snow on the ground but the Ka-band (~37 GHz) is more sensitive to the snowpack (Chang et al., 1987). However, such empirical and semiempirical retrieval approaches are not robust due to the non-linear relationship between snow depth and TB gradients caused by the confounding effects of snow microstructure, the impact of snowpack stratigraphy, the forest cover, and the saturation for snow conditions exceeding a given threshold (Durand and Margulis, 2007). Machine learning (ML) techniques are powerful tools for snow depth estimation due to their ability to handle non-linearity. There are lots of ML methods (e.g., ANN, SVR) applicated into snow depth estimation. Actually, ANNs with many layers are examples of deep learning algorithms (Reichstein et al., 2019). We conducted a comparison of SVM, ANN and RF models for snow depth estimation over China (Fig.1). The results showed that RF still outperforms others. Although SVM performs better than ANN, it takes much more time to train the model. Moreover, to our knowledge, the performance of the RF model in spatiotemporal snow depth estimation has not been assessed to date. Thus, the main aim of this study was to assess the feasibility of the RF model in estimating snow depth, rather than determining which ML technique is most accurate.

We agree with your opinion that deep learning is an advanced technique. As we discussed in the revised manuscript, it is in addressing the shortcomings of traditional machine learning methods that the techniques of deep learning promise breakthroughs. However, any machine learning model has certain advantages. For example, RF regression is insensitive to the quality and size of training samples and tends to process quickly and be easy to use (Biau and Scornet, 2016; Probst and Boulesteix, 2018; Maxwell et al., 2018). The deep learning approach generally requires high-dimensional predictor variables and large sample numbers to train the model. In our view, prior snow knowledge is the most important variable in the ML approach, and the absence of snow parameters (e.g., grain size) does not improve its predictive ability. The reviewer provides a promising approach for snow depth estimation, and our next step will focus on coupling physical process models with the data-driven deep learning technique (Please refer to the response to "General comments 2" below).

[Figure]

Figure 1. Comparison of (a) RF, (b) ANN, and (c) SVM with respect to snow depth estimation in China.

**2.** The authors have now discussed the impact of the diurnal changes in the snowpack geophysical properties, including the microstructure, as mentioned in my previous comment. However, this discussion is not comprehensive. Due to the absence of any experiments, this cannot be correlated to the errors observed. Thus, the analysis and the discussion part in this manuscript is still lacking.

For example, the authors have considered in their RF algorithms inputs such as the latitude and the longitude. But why not any snowpack specific parameters. These can be obtained from the ERA5 data and used as inputs to their RF models. The authors then should check the discrepancy in the RMSE based on these parameters.

Another analysis that I find missing is seasonal variation in accuracy, which is very important. The readers should know the difference in the performance of the algorithm in the off-peak snow seasons, i.e., during the early winter and the melt season.

**Response 2:** Thank you for your comments. We agree with your opinion that integrating a priori knowledge of snowpack into the machine learning models can improve snow depth estimates. We evaluated the ERA5-land hourly snow density product (https://cds.climate.copernicus.eu/) with station measurements. Many stations in China measured (observation time 8:00-9:00 a.m.) snow depth and snow pressure every five days during the period 2014-2017; therefore, we collected many in situ snow density observations as the validation dataset. We used this dataset to evaluate ERA5-land hourly snow density data at 8:00 a.m. Fig.2 shows that ERA5 retrievals are poorly correlated with station measurements. Thus, we dismissed this strategy. Actually, we are attempting to combine physical snow evolution models of snowpack with machine learning techniques. Those physical snow evolution models, such as the Snow Thermal Model (SNTHERM) (Jordan, 1991), SNOWPACK (Lehning et al., 2002a, b), and Crocus (Brun et al., 1989; Vionnet et al., 2012), can be used to simulate snow parameters (e.g., snow grain size, snow density) relatively accurately using time-series meteorological and snowpack measurements. ERA5-land have provided all these forcing data for physical snow evolution models since 1981. It will take some time to conduct this work because such physical models are driven by surface meteorological hourly data (precipitation, air temperature, wind, downward solar, and longwave radiation) and require topography, soil, and vegetation information. Thus, we will show the results in a future publication.

In addition, we discussed this in Section 4.1. "Snow physical snow evolution models, e.g., the Snow Thermal Model (SNTHERM) (Jordan, 1991), SNOWPACK (Lehning et al., 2002a, b), and Crocus (Brun et al., 1989; Vionnet et al., 2012) can be used to simulate snow parameters (e.g., grain size, density) relatively accurately. Thus, integrating a priori knowledge of snowpack into ML techniques has the potential to overcome many limitations that have hindered a more widespread adoption of ML approaches." (Page 10, Lines 15-18 in the revised manuscript).

[Figure]

Figure 2. Validation of the ERA5-land snow density product with in situ observations in China.

According to the reviewer's suggestion, monthly performances of both RF and WESTDC products were analyzed and compared. Fig.3 shows that unRMSEs of both products present increasing trends from the beginning to end of the snow season. This is due to seasonal evolution of snowpack, which leads to uncertainty in snow depth estimates. WESTDC estimates tend to be underestimated in November, December and March, while the RF product is superior to the WESTDC data. The correlation coefficient in January is the highest among snow season months, which is attributable to stable snow cover.

We added the following text in the revised manuscript: "Snow depth estimates with PMW data are usually challenged by the snow metamorphism (e.g., snow grain size). In particular, the large diurnal temperature range in the late snow season leads to rapid snow grain growth (Dai et al., 2012). Fig. 9 presents the monthly performances of both RF and WESTDC products. The RF estimates outperform the WESTDC product in terms of correlation, overall bias and unbiased RMSE. WESTDC estimates tend to be underestimated in November, December and March, while the RF product is superior to the WESTDC data. Due to the influence of the seasonal evolution of snowpack, the unbiased RMSEs of both products present increasing trends from November to March during snow seasons. The correlation coefficient in January is the highest among snow season months, which is attributed to stable snow cover." (Page 8, Lines 7-13 in the revised manuscript).

[Figure]

Figure 3. Monthly performances of (a) RF and (b) WESTDC snow depth products.

**3.** Please mention the upscaling method for 1-km to 25-km LULC map. Additionally, I also mention which latitude and longitudes were used as inputs. I hope these were geographic ones and not projected ones.

**Response 3:** The 1-km land use/land cover (LULC) data were provided by the Data Center for Resources and Environmental Sciences, Chinese Academy of Sciences (http://www.resdc.cn/). In a 25-km pixel, there are a total of 625 1-km grids. For example, the number of grassland 1-km grids within a 25-km pixel is *n*, and the fraction is calculated as:

$$\text{fraction} = \frac{n}{625} * 100\%$$

The latitude and longitude used in this study were geographic coordinates, retrieved with an official procedure (https://nsidc.org/data/ease/tools) based on the row and column of EASE-GRID data.

**4.** I find that the majority of the discussion is based on the comparison between the results from the RF method and the WESTDC. While I believe there should be a better balance in the discussion for comparison with the in-situ measurements.

**Response 4:** Both the RF and WESTDC products were evaluated using in situ measurements. In Section 3, we presented all the validation results. In Section 4, we discussed the disadvantage (poor performance at the spatial scale) of the RF model, the unnecessary predictor variables (if highly correlated) for the RF model, and the potential errors (overestimation and underestimation) of the RF product. All these discussions were based on the assessment results in Section 3.

**5.** Another analysis to the manuscript that the authors should add is for the selection of trees. Presently, the authors have fixed this to 1000. However, is there any merit in using this large number? What is the performance for other smaller number of trees, for example, 500, 250, etc. Is the performance significant enough to use such a relatively large number of trees?

**Response 5:** The default value for *ntree* is 500 in the RandomForest package. One of the merits of the RF model is rapid training. Moreover, it can avoid overfitting due to the large number of decision trees. Thus, we set *ntree* to 1000.

**6.** An important parameter in the discussion is the R-square that is missing?

**Response 6:** Thank you for your comments. In general, the correlation of in situ measurements and PMW retrievals is not significant (low R-square) due to the non-linear relationship between snow depth and satellite TB gradients. Thus, we dismissed this index in the discussion.

**7.** Table 5, the results are contradictory to the explanations given by the authors. The authors have mentioned that for shallow snow depth, the PMW could be insensitive. However, Table 5 shows the contradictory, i.e., rather some sensitivity in case of shallow snow depths as compared to deeper snow where there is nearly no correlation. I believe the inconsistency of the results requires a thorough analysis.

**Response 7:** PMW signals are insensitive to shallow fresh snow (< 5 cm) due to low snow density and snow grain size (Chang et al., 1987; Foster et al., 2005). However, they tend to overestimate snow depth for shallow old snow in the late snow season due to the seasonal evolution of snowpack. This is one of the reasons for the overestimation of shallow snow cover (Table 5). Please refer to the response to "Minor issues 3" below.

Due to snow metamorphism and the 'saturation effect', the general tendency for the correlation between snow depth and brightness temperature gradient ($TBD_{19V\&37V}$) is poor for deep (> 20 cm) snow conditions (Fig. 4).

[Figure]

Figure 4. Correlation between brightness temperature gradient and in situ snow depth measurements.

**Minor issues:**

**1.** Please check Figure 4 for correctness. Is the validation dataset correct? This is very confusing from the authors response.

**Response 1:** We apologize for the confusion. Training datasets for RF models were selected randomly from station records during the period 1987-2004. We applied the same validation dataset (2005-2006) to evaluate the well-trained RF models (Fig. 5).

We rewrote this part as follows: "To ensure a sufficient number of samples, all station records (approximately 100,000 samples) from 1987 to 2006 were used to analyze the sensitivity of the RF model to the training sample size. A total of 5000 to 80,000 (with a step of 5000) samples were selected randomly from data during the period 1987-2004 and used to train the RF models, and a two-year stand-alone dataset from 2005 to 2006 was applied to assess the performance of the well-trained model" (Page 6, Lines 1-4 in the revised manuscript).

[Figure]

Figure 5. The test process flowchart for the sensitivity of the RF model to the training sample size.

**2.** Some of the colors used by authors are very poor. Figure 6 the turquoise text, Figure 9, the orange legends and lines.

**Response 2:** Corrected.

![Scatter plots: 4 rows (RF1-RF4) by 3 columns (Test1-Test3) of Estimated SD vs Ground truth SD density scatterplots]

| Panel | corr.coe | RMSE | unbiasedRMSE | bias | samples |
|---|---|---|---|---|---|
| RF1-Test1 | 0.72 | 6.4 cm | 6.4 cm | -0.01 cm | 14301 |
| RF1-Test2 | 0.77 | 5.4 cm | 5.4 cm | 0.12 cm | 34684 |
| RF1-Test3 | 0.57 | 8 cm | 7.9 cm | -0.76 cm | 25879 |
| RF2-Test1 | 0.9 | 4 cm | 4 cm | 0.07 cm | 14301 |
| RF2-Test2 | 0.85 | 4.5 cm | 4.5 cm | 0.27 cm | 34684 |
| RF2-Test3 | 0.66 | 7.3 cm | 7.2 cm | -0.97 cm | 25879 |
| RF3-Test1 | 0.9 | 3.9 cm | 3.9 cm | 0.08 cm | 14301 |
| RF3-Test2 | 0.85 | 4.5 cm | 4.5 cm | 0.24 cm | 34684 |
| RF3-Test3 | 0.66 | 7.3 cm | 7.3 cm | -0.83 cm | 25879 |
| RF4-Test1 | 0.91 | 3.9 cm | 3.9 cm | 0.03 cm | 14301 |
| RF4-Test2 | 0.85 | 4.4 cm | 4.4 cm | 0.21 cm | 34684 |
| RF4-Test3 | 0.65 | 7.3 cm | 7.3 cm | -0.41 cm | 25879 |

Figure 6. The color-density scatterplots of the estimated snow depth with four fitted RF algorithms and the ground truth snow depth. The four trained RF algorithms (RF1, RF2, RF3, RF4) were evaluated with three validation datasets (Test1, Test2, Test3).

[Figure]

Figure 7. The validation of the RF and WESTDC snow depth products in three stable snow cover areas over China with respect to (a) the unbiased RMSE and (b) bias and the correlation coefficient.

**3.** Although I meant with my comment to include a discussion, the MEMLS simulation experiment is appreciated. However, the parameters used in the MEMLS simulation are not correct in the response by the authors. How often do we have a snowpack representing fresh snow characteristics? This is very seldom. Thus, there is no point showing an experiment with a snow density of 100 kg/cubic m corresponding to fresh snow. Instead, the authors should have selected something between 200-350 km/cubic m for the simulation, which corresponds to old snow.

**Response 3:** Thank you for your constructive comments. The RF product presents an overestimation for shallow snowpack. One of the reasons is the strong scattering effect caused by snow metamorphism, especially for old shallow snow. Table 1 shows that the accuracy in the late snow season is poor. This is due to the seasonal snow evolution.

We rewrote this part as follows: "Table 5 indicates that there is overestimation in NE and northern XJ for shallow snow cover, which may be due to the following reasons. First, the PMW signals are insensitive to thin snow cover (< 5cm), especially for fresh snow with low snow density and snow grain size, which generally results in underestimation (Foster et al., 2005). In contrast, it tends to overestimate snow depth for shallow old snow in the late snow season due to the seasonal evolution of snowpack. For example, the large diurnal temperature range in the late snow season tends to subject the snowpack to frequent freeze-thaw cycles and leads to rapid snow grain (~2 mm) and snow density (200-350 kg/m³) growth and consequently a high $T_B$ difference (Meløysund et al., 2007; Durand et al., 2008; Yang et al., 2015; Dai et al., 2017). Thus, the overall bias and unbiased RMSE for shallow snowpacks (< 10 cm) present increasing trends from November to March in NE and northern XJ" (Page 11, Lines 14-21 in the revised manuscript).

Table 1. Summary of monthly performances of the RF product in NE and northern XJ.

| | NE | | | | |
| --- | --- | --- | --- | --- | --- |
| Month | November | December | January | February | March |
| corr.coe | 0.32 | 0.41 | 0.40 | 0.23 | 0.08 |
| bias (cm) | 2.33 | 2.19 | 2.93 | 4.74 | 7.97 |
| unRMSE (cm) | 3.66 | 3.69 | 4.16 | 5.24 | 6.16 |

| northern XJ | | | | | |
|---|---|---|---|---|---|
| Month | November | December | January | February | March |
| corr.coe | 0.20 | 0.27 | 0.40 | 0.20 | 0.08 |
| bias (cm) | 3.68 | 3.35 | 2.97 | 5.65 | 10.60 |
| unRMSE (cm) | 4.49 | 4.77 | 4.61 | 6.83 | 7.09 |

The microwave emission model of layered snowpack (MEMLS) was applied to simulate the $T_B$ with varying snow parameters. Generally, the snow density (< 100 kg/m$^3$) and snow grain size (correlation length < 0.2 mm) are small for shallow, new snow cover (< 5 cm), while these values are 200-350 kg/m$^3$ and 1-2 mm for snow density and grain size, respectively. Fig. 8 shows the sensitivity of snow depth to $T_B$ at 36 GHz for fresh and old snowpacks. The passive microwave signals are insensitive to the new snow cover. In contrast, there is notable attenuation to microwave radiation at the 36.5-GHz channel for old snow; therefore, snow depth is overestimated.

[Figure]

Figure 8. The sensitivity of snowpack stratigraphy to the passive microwave brightness temperature simulated with the MEMLS model. Snow grain size is generally 4-5 times greater than the correlation length in the MEMLS model (Mätzler, 2002). Top: fresh snow; Bottom: old snow.

**Response to Reviewer Comments by Review #2 on "Real-Time Snow Depth Estimation and Historical Data Reconstruction Over China Based on a Random Forest Machine Learning Approach" by Jianwei Yang et al.**

Many thanks for the very positive comments on the revised version of the manuscript. We have studied the comments carefully and have made corrections, and responses are provided in blue font below.

Referee #2

The manuscript has been improved considerably. The authors had to address multiple conflicting reviewers' comments and in my opinion they did so fine. I have some minor comments remaining to be addressed:

**General evaluation**

**1.** Perhaps the manuscript emphasizes the use of random forests, however given the limited comparison with other algorithms in both the original and revised manuscript, the manuscript should emphasize the construction of the new dataset.
Based on this, the dataset should be uploaded online and made available to the readers of the Journal.

**Response 1:** We appreciate the reviewer's suggestions. Accurate retrieval of snow depth from passive microwave (PM) measurements has been challenging due to the confounding effects of snow microstructure, the impact of snowpack stratigraphy, the forest cover, the liquid water in the snowpack, the coarse spatial resolution, and the saturation effect (Durand and Margulis, 2007). Thus, the relationship between snow depth and satellite observation is clearly non-linear for all frequencies. Machine learning (ML) techniques are powerful tools for establishing the non-linear relations between independent variables and the target variable. Random forest approach is one of the most successful ML algorithms due to its ability to handle non-linearity (Rodriguez-Galiano et al., 2012; Belgiu et al., 2016; Maxwell et al., 2018; Bair et al., 2018). In addition, the assessment of RF algorithm performance in snow depth estimation has not been made to date for China.
We would like to share this dataset online. Actually, we have already prepared the reconstructed dataset from 1987 to 2018. We also wrote a descriptive document to introduce this dataset. Snow depth data are publicly available at the following link: https://figshare.com/s/60c2e9db228130e6768b.

**2.** Furthermore, the discussion in Section 4.1 should be modified to incorporate elements of stationarity of the earth's system that allow the use of random forests. Stationarity should not be confused with the concept of a constant system and is not directly related to the concept of dynamic system (which is used in Section 4.1).

**Response 2:** We agree with your opinion. We have changed the sentence to "however, these studies aim to obtain spatial predictions of elements of stationarity in the Earth system, e.g., soil types and soil properties" in the revised manuscript.

**3.** Lastly, while the revised manuscript includes an extended presentation of the properties of the implemented algorithms, it is has not been made explicit that the properties of the constructed dataset largely depend on the implemented algorithm (albeit this can be deduced by the manuscript), while reporting RMSE (and other metrics) values does not fully address issues related to the properties of the dataset.

**Response 3:** Thank you for your comments. The widely used inversion algorithms for snow depth estimation are based on empirical or semiempirical relationships between snow depth and brightness temperature (TB) gradients. The basic assumption for these algorithms is that TB at K-band (~19 GHz) is weakly affected by the snow on the ground but the Ka-band (~37 GHz) is more sensitive to the snowpack. However, accurate retrieval of snow depth from PM data has been challenging due to many factors, e.g., the confounding effects of snow microstructure, the impact of snowpack stratigraphy, the forest cover, and the saturation for snow conditions exceeding a given threshold. Thus, such empirical and semiempirical retrieval approaches are not robust due to the non-linear relationship between snow depth and TB gradients. Random forest approach is one of the most powerful ML tools to handle non-linearity relationship between snow depth and satellite observation, which makes retrievals improved.

Meanwhile, we have explicitly described the properties of the implemented RF model in the introduction.

**Procedure code:** a version 4.6-14 randomForest R package;

**Model parameters:** two user-defined parameters: $mtry = 1000$; $mtry = 2$;

**Predictor variables:** satellite vertical polarization brightness temperature observations at 19 GHz and 37 GHz ($TB_{19V}$, $TB_{37V}$), Latitude, Longitude;

**Target Variable:** snow depth;

**Training samples:** a total of 28602 during the period 2012-2014;

**Validation dataset:** spatially independent reference data during the period 1987-2018.

**Dear Editor, dear reviewers,**

**I would like to thank the two reviewers for their positive and constructive comments on our paper. Below, we list the changes in the revised manuscript.**

**List of changes (page and line numbers refer to the revised manuscript):**

• **p.6, lines 1-4:** Rewrote the description of the flowchart as: "All station records (approximately 100,000 samples) from 1987 to 2006 were used to analyze the sensitivity of the RF model to the training sample size. A total of 5,000 to 80,000 (with a step size of 5,000) samples were selected randomly from the data during the period 1987-2004 and were used respectively to train the RF models, and a two-year stand-alone dataset from 2005 to 2006 was applied to assess the performances of the well-trained models."

• **p.8, lines 7-14:** Added the analysis of monthly performances of the RF product constructed in this study: "Snow depth estimates with PMW data are usually challenged by the snow metamorphism (e.g., snow grain size). In particular, the large diurnal temperature range in the late snow season leads to a rapid snow grain growth (Dai et al., 2012). Fig. 9 presents the monthly performances of both RF and WESTDC products. The RF estimates outperform the WESTDC product in terms of correlation, overall bias and unbiased RMSE. WESTDC estimates tend to be underestimated in November, December and March, while the RF product is superior to the WESTDC data. Due to the influence of the seasonal evolution of snowpack, the unbiased RMSEs of both products present increasing trends from November to March during the snow seasons. The correlation coefficient in January is the highest among snow season months, which is attributed to stable snow cover."

• **p.10, lines 6-8:** Replaced "static" with "stationarity" and "not static but dynamic" with "non-stationary".

• **p.10, lines 15-19:** Added: "Physical snow evolution models, e.g., the Snow Thermal Model (SNTHERM) (Jordan, 1991), SNOWPACK (Lehning et al., 2002a, b), and Crocus (Brun et al., 1989; Vionnet et al., 2012), can be used to simulate snow parameters (e.g., grain size, density) relatively accurately. Thus, integrating a priori knowledge of snowpack into ML techniques has the potential to overcome many limitations that have hindered a more widespread adoption of ML approaches."

• **p.11, lines 15-21:** Rewrote: "First, the PMW signals are insensitive to thin snow cover (< 5cm), especially for fresh snow with low snow density and snow grain size, which generally results in underestimation (Foster et al., 2005). In contrast, it tends to overestimate snow depth for shallow old snow in the late snow season due to the seasonal evolution of snowpack. For example, the large diurnal temperature range in the late snow season tends to subject the snowpack to frequent freeze-thaw cycles and leads to rapid snow grain (~2 mm) and snow density (200-350 kg/m$^3$) growth and consequently a high $T_B$ difference (Meløysund et al., 2007; Durand et al., 2008; Yang et al., 2015; Dai et al., 2017). Thus, the overall bias and unbiased RMSE for shallow snowpacks (< 10 cm) present increasing trends from November to March in NE and northern XJ (Table 6)."

• **p.12, line 24:** Added: "Combining the physical snow evolution model (e.g., SNOWPACK) with the ML method will be the focus of future work."

• **p.13, line 33:** Added: "Brun, E., Martin, E., Simon, V., Gendre, C., and Coleou, C.: An Energy and Mass Model of Snow Cover Suitable for Operational Avalanche Forecasting, Journal of Glaciology, 35, 333–342, 10.1017/S0022143000009254, 1989."

• **p.15, line 18:** Added: "Jordan, R.E. 1991.: A One-Dimensional Temperature Model for a Snow Cover: Technical Documentation for SNTHERM.89; U.S. Army Cold Regions Research and Engineering Laboratory: Hanover, NH, USA."

• **p.15, lines 30-34:** Added: "Lehning, M., Bartelt, P., Brown, B., Fierz, C., Satyawali, P.: A physical SNOWPACK model for the Swiss avalanche warning part II. Snow microstructure, Cold Reg. Sci. Technol, 35(3), 147–167, 10.1016/S0165-232X(02)00073-3, 2002a. Lehning, M., Bartelt, P., Brown, B., Fierz, C.: A physical SNOWPACK model for the Swiss avalanche warning: Part III: meteorological forcing, thin layer formation and evaluation, Cold Reg. Sci. Technol, 35(3):169–184, 10.1016/S0165-232X(02)00072-1, 2002b."

• **p.17, line 37:** Added: "Vionnet, V., Brun, E., Morin, S., Boone, A., Faroux, S., Le Moigne, P., Martin, E., Willemet, J.-M.: The detailed snowpack scheme Crocus and its implementation in SURFEX v7.2, Geosci. Model Dev, 5, 773–791, 2012."

• **p.19, line 5:** Added: "Table 6"

• **p.22, line 6:** Redrawn: "Fig. 6"

• **p.24, line 2:** Redrawn: "Fig. 9"

• **p.24, line 6:** Added: "Fig. 10"

[revised manuscript text omitted]

---

## Author Response (AR3)

Dear editor:

Thank you for your extensive and detailed review of our paper. Below, we provide a detailed explanation of why the station data are not publicly accessible and list the changes in the revised manuscript.

**(1)** The snow course measurements provided by the Chinese Snow Survey (CSS) project were not used in the revised manuscript. Thus, we removed their description from the "Data availability" section.

**(2)** The dataset of daily snow depth measurements from stations used in this manuscript was obtained from the China Meteorological Administration (CMA). We are the data users, not the owners. Therefore, we are not allowed to share these station data. When we applied for permission to use the ground station data from the CMA, we signed an agreement of use for this dataset. If scientific researchers are interested in this dataset, it can be accessed through the submission of an application. The website is http://data.cma.cn/en.

**(3)** The WESTDC daily snow depth product was obtained from the Environmental and Ecological Science Data Center for West China (http://data.casnw.net/portal/).

**(4)** The daily snow depth product generated in this manuscript is available to the public (https://figshare.com/s/60c2e9db228130e6768b). The unzipped file contains daily snow depth across China from 1987 to 2018 in HDF format.

**(5)** The RF snow depth data have been deposited in a data repository (https://figshare.com/articles/RF_based_Longterm_SnowDepth_China_rar/11988027) and have been assigned a DOI (10.6084/m9.figshare.11988027).

[revised manuscript text omitted]

---

## Author Response (AR4)

Dear Editor Florent Dominé

Thanks for your understanding and your constructive suggestions on the data sharing policy and platform. We corrected the English website of station data application (http://data.cma.cn/en) in the revised manuscript. And I hope the data sharing policy would be more convenient and open to all over the world in the coming future. The Chinese scientists have being promoted the data sharing to all over the world, including providing English version website and open to all.

Best regards,

Lingmei Jiang, on behalf of the authors

[revised manuscript text omitted]